# LETTERS
# Cell cycle-specific phase separation regulated by protein charge blockiness

Hiroya Yamazaki[1,5], Masatoshi Takagi[2], Hidetaka Kosako[3], Tatsuya Hirano [4] and
Shige H. Yoshimura [1✉]

**Dynamic morphological changes of intracellular organelles are often regulated by protein phosphorylation or dephosphorylation[1-6]. Phosphorylation modulates stereospecific interactions among structured proteins, but how it controls molecular interactions among unstructured proteins and regulates their macroscopic behaviours remains unknown. Here we determined the cell cycle-specific behaviour of Ki-67, which localizes to the nucleoli during interphase and relocates to the chromosome periphery during mitosis. Mitotic hyperphosphorylation of disordered repeat domains of Ki-67 generates alternating charge blocks in these domains and increases their propensity for liquid–liquid phase separation (LLPS). A phosphomimetic sequence and the sequences with enhanced charge blockiness underwent strong LLPS in vitro and induced chromosome periphery formation in vivo. Conversely, mitotic hyperphosphorylation of NPM1 diminished a charge block and suppressed LLPS, resulting in nucleolar dissolution. Cell cycle-specific phase separation can be modulated via phosphorylation by enhancing or reducing the charge blockiness of disordered regions, rather than by attaching phosphate groups to specific sites.**

Numerous recent studies have reported the liquid-like behaviour of intracellular membraneless organelles, such as nucleoli, stress granules and processing bodies[7]. They are formed via promiscuous interactions among proteins and nucleic acids by liquid–liquid phase separation (LLPS), coacervation or condensation[8-11]. Reversible formation and dissolution of these organelles during the cell cycle and intracellular signalling plays critical roles in cellular responses and homoeostasis and is regulated by various post-translational modifications. Phosphorylation, one of the most common post-translational modifications occurring in cellular proteins, changes the structure, interactions and intracellular localization of substrate proteins and regulates several intracellular signalling pathways[12,13]. An increasing number of studies have reported phosphorylation-dependent regulation of LLPS and intracellular liquid-like organelles. Phosphorylation regulates not only protein-based phase separation in positive or negative manners[1-6] but also protein–nucleic acid coacervation[14]. A recent study showed that viral replication is regulated by phosphorylation-dependent LLPS of viral nucleocapsid protein[15].

In a protein with a rigid three-dimensional structure, the addition of a phosphate group induces a conformational change locally or allosterically, which in turn affects the interaction with its substrate and/or partner protein(s). However, how phosphorylation of intrinsically disordered regions (IDRs) of proteins regulates LLPS mechanistically remains elusive[16-19]. Recent studies using charged polymers and theoretical modelling demonstrated that a polyampholyte chain with segregated charged residues (charge blocks) exhibits stronger phase separation than the chain with the same number of charged residues randomly distributed[20-23]. Charge blocks also play important roles in phase separation in vivo[24-26], and shuffling of the charged residues along a polypeptide results in the dispersion of liquid-like organelles in the cell[24,25]. As many IDRs of cellular proteins are hyperphosphorylated during mitosis, for example, Ki-67, RIF1, INCENP and NPM1[16], the addition of multiple negatively charged groups may enhance or reduce such 'charge blockiness' of IDRs and affect the propensity for LLPS in the cell. However, direct evidence for this hypothesis is lacking.

Ki-67, a nucleolar phosphoprotein, plays a critical role in organizing the periphery of mitotic chromosomes, which are thought to have a liquid-like property. Ki-67 separates chromosomes from each other and prevents their coalescence during mitosis[27-32]. Human Ki-67 is composed of multiple domains, including an N-terminal PP1-binding domain, a central repeat domain (RD) composed of 16 repeats of an ~110-amino-acid unit, and a C-terminal chromatin-targeting domain (LR domain) for chromosome binding (Fig. 1a). Phosphoproteomic analyses demonstrated that Ki-67 is hyperphosphorylated by CDK1 and other mitotic kinases[33]. Our quantitative mass spectrometric analysis of mitotic phosphorylation identified more than 70 residues in the RD that are significantly phosphorylated upon entry into mitosis[16]. Comparison of the charge distributions between the mitotic (hyperphosphorylated) form and interphase (dephosphorylated) form revealed that mitotic phosphorylation converts the individual repeats into strong diblock ampholytes, in which a positive charge block is followed by a negative block (Fig. 1b). This tendency was identified in most of the repeats present in the RD (Extended Data Fig. 1a), suggesting that mitotic phosphorylation enhances the alternating charge blocks throughout the RD. In contrast, mitotic hyperphosphorylation of nucleophosmin (NPM1), an IDR-rich nucleolar protein that interacts with Ki-67 and plays a critical role in assembling nucleolar components in interphase cells[34], diminishes the alternating charge blocks that otherwise exist in the non-phosphorylated form (Extended Data Fig. 1b). Therefore, mitotic hyperphosphorylation may introduce negative charges to enhance or reduce the alternating charge blocks in the IDRs and modulate the propensity for LLPS (Fig. 1c). In this Letter, we tested the hypothesis that changes in charge blockiness, rather than the attachment of phosphate groups

[1]Graduate School of Biostudies, Kyoto University, Kyoto, Japan. [2]Cellular Dynamics Laboratory, RIKEN Cluster for Pioneering Research, Saitama, Japan. [3]Division of Cell Signaling, Fujii Memorial Institute of Medical Sciences, Tokushima University, Tokushima, Japan. [4]Chromosome Dynamics Laboratory, RIKEN Cluster for Pioneering Research, Saitama, Japan. [5]Present address: Graduate School of Science, The University of Tokyo, Tokyo, Japan. ✉e-mail: yoshimura@lif.kyoto-u.ac.jp

625

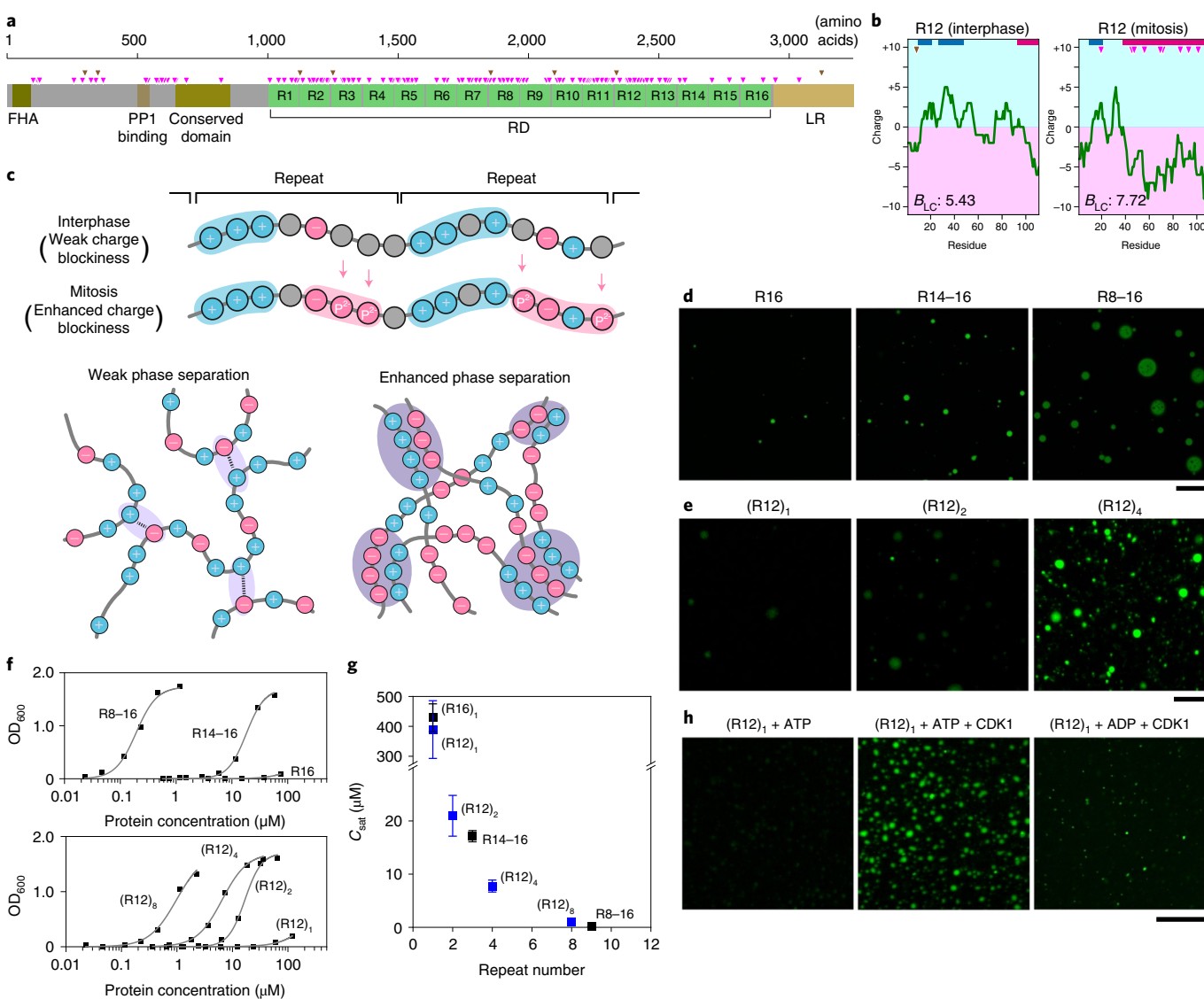

**Fig. 1 | Mitotic hyperphosphorylation of human Ki-67 enhances the charge blockiness of its RD and LLPS. a**, Primary structure of human Ki-67 (long isoform). Phosphorylation sites increased and decreased upon mitotic entry[16] are indicated by magenta triangles and brown triangles, respectively. FHA, forkhead-associated domain. **b**, Charge plot of R12 in interphase (left) and mitotic (right) forms (window size: 25 amino acids). The positions of phosphorylation are indicated by triangles. Positive and negative charge blocks (Methods) are depicted by blue and red bars, respectively. The $B_{LC}$ value, which quantifies the charge blockiness, is also shown. **c**, Illustration of how mitotic hyperphosphorylation enhances the charge blockiness and promotes phase separation. **d,e**, In vitro LLPS assay using ATTO488-labelled recombinant proteins. Fluorescence microscopic images of recombinant proteins carrying different RD stretches (40 μM R16, 5 μM R14–16 and 0.2 μM R8–16) (**d**) or increasing copy numbers of R12 (40 μM (R12)$_1$, 40 μM (R12)$_2$ and 20 μM (R12)$_4$) (**e**) are shown. Scale bar, 30 μm. **f,g**, The turbidity of the droplet solution was measured as the OD$_{600}$ and plotted against the protein concentration (**f**). One representative result is shown for each construct (out of three). $C_{sat}$, which represents the protein concentration giving a half-maximal OD value, was obtained by curve fitting and plotted against the repeat number (**g**). Error bars reflect standard deviation of the mean ($n = 3$ independent measurements). **h**, Wild-type R12 was incubated with ATP or ADP in the absence or presence of purified CDK1–cyclin B and subjected to LLPS assay. Scale bar, 50 μm. Source numerical data are available in source data.

at specific sites, play critical roles in phase separation of Ki-67 and NPM1 and in the morphological dynamics of the periphery of mitotic chromosomes and nucleoli.

Recombinant proteins of human Ki-67 RD formed liquid-like droplets in vitro in the presence of 100 mM NaCl and 15% polyethylene glycol. (Fig. 1d and Extended Data Fig. 2a,b). Droplet formation increased with the number of repeats (Fig. 1d and Extended Data Fig. 2c). Tandem homogeneous repeats of repeat 12 ((R12)$_1$, (R12)$_2$ and (R12)$_4$) showed a similar tendency (Fig. 1e and Extended Data Fig. 2d). A clear inverse correlation between the number of

repeats and the propensity of LLPS (quantified by the saturation concentration, $C_{sat}$[10]) was observed; $C_{sat}$ sharply decreased as the repeat number increased (Fig. 1f,g). The effect of phosphorylation on LLPS was examined. R12 contains nine mitotic phosphorylation sites (Fig. 1b), six of which harbour a consensus motif for CDK1[35]. In vitro phosphorylation of recombinant R12 by CDK1 increased droplet formation (Fig. 1h and Extended Data Fig. 2e).

Phosphomimetic mutations in nine mitotic phosphosites (Pm9) in R12 enhanced droplet formation (Fig. 2a,b and Extended Data Fig. 3a). Phosphomimetic mutations in another repeat (R7) also

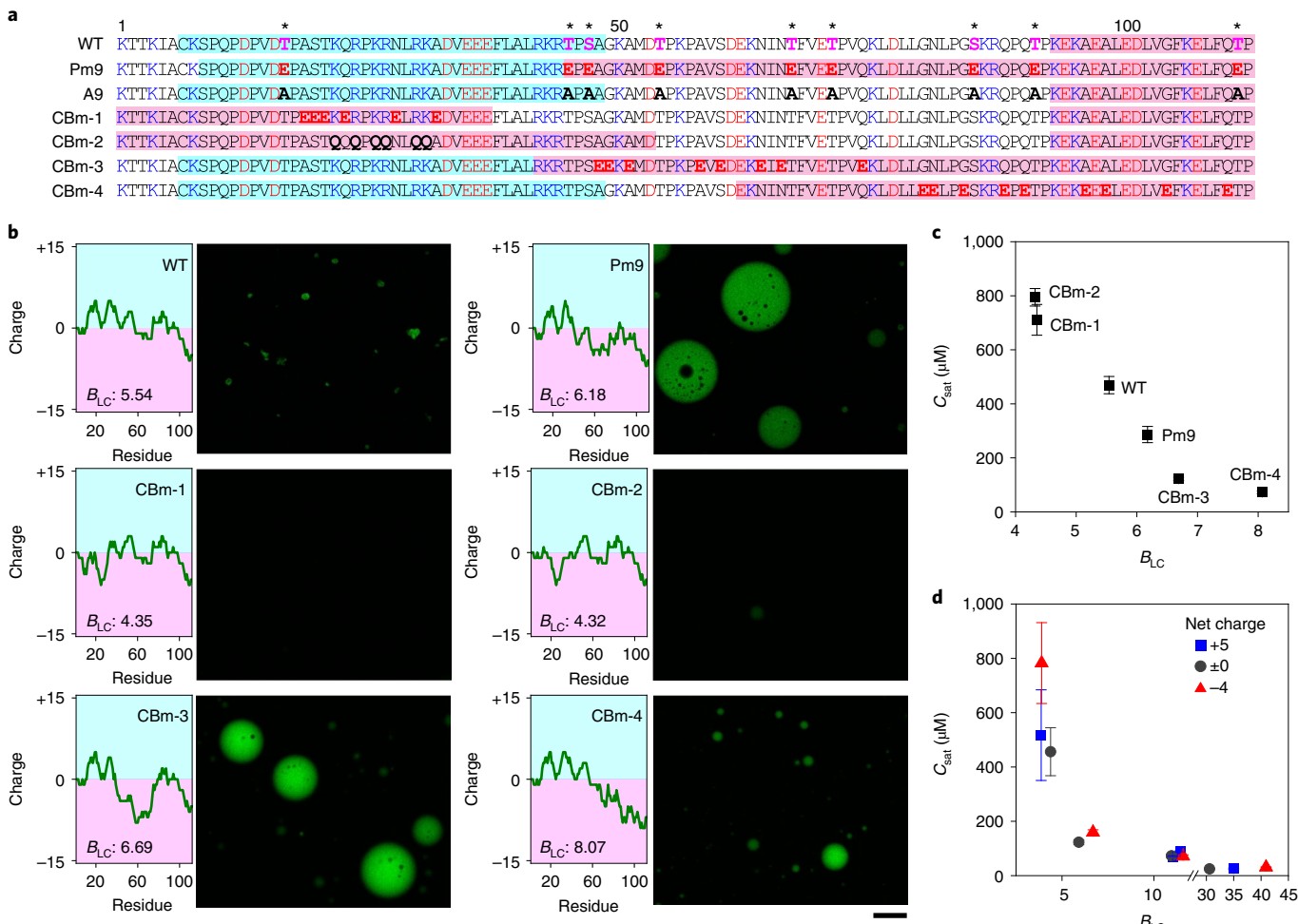

**Fig. 2 | Alternating charge blocks of RD are critical for LLPS. a**, Amino-acid sequence of R12 WT and mutants. Mitotic phosphosites are indicated by an asterisk. Positively and negatively charged amino acids are shown in blue and red, respectively. Mutated residues are shown in bold. Positive and negative charge blocks (Methods) are highlighted in cyan and red, respectively. **b**, LLPS assay of R12 fragments shown in **a**. Charge plot (window size: 25 amino acids), $B_{LC}$ values and fluorescence images of liquid droplets are shown. Scale bar, 30 μm. **c,d**, Correlation between $B_{LC}$ and $C_{sat}$. $C_{sat}$ values of R12 WT and mutants described in **a** were obtained by a turbidity assay and plotted against $B_{LC}$ values (**c**). The same analysis was performed with a series of charge block mutants carrying different net charges (−4, 0 or +5) (**d**). Amino-acid sequences and charge plots of individual mutants are shown in Extended Data Fig. 4b. Error bars reflect standard deviation of the mean ($n=3$ independent measurements). Source numerical data are available in source data.

enhanced LLPS (Extended Data Fig. 3b,c). Notably, the phosphomimetic mutations did not induce the formation of any secondary structures (Extended Data Fig. 3d). To demonstrate that alternating charge blocks are important and necessary for LLPS, we constructed a series of R12 mutants in which the charge distribution was modified by replacing the amino acids that are not involved in mitotic phosphorylation (charge-block mimetic mutant, CBm; Fig. 2a) and subjected these mutants to the LLPS assay (Fig. 2b and Extended Data Fig. 3a). Neutralization of the positive charge block at the amino-terminal region by substituting either the neutral amino acids with glutamic acid (E) residues (CBm-1) or K/R by Q (CBm-2) reduced the formation of liquid droplets (Fig. 2b). In contrast, replacement of neutral residues in the middle region with E residues, which mimics phosphorylated charge blocks (CBm-3), substantially promoted LLPS, as was observed in the phosphomimetic mutant (Pm9). A similar effect was observed when E residues were introduced in the carboxyl-terminal region (CBm-4) (Fig. 2b).

The relationship between charge blockiness and LLPS was investigated quantitatively. The extent of charge blockiness along the polypeptide was evaluated on the basis of either the blockiness of

like charges ($B_{LC}$) or degree of segregation ($D_{seg}$) (Methods). Plotting $C_{sat}$ against $B_{LC}$ and $D_{seg}$ revealed a clear inverse correlation (Fig. 2c and Extended Data Fig. 4a). To characterize the relationship further, a series of mutants (CBm-5–16) carrying different net charges (between −4 and +5) and/or charge blockiness (Extended Data Fig. 4b) were constructed and subjected to LLPS assay. $C_{sat}$ more closely correlated with charge blockiness ($B_{LC}$ and $D_{seg}$) (Fig. 2d and Extended Data Fig. 4c) than with the net charge (Extended Data Fig. 4d). Together, these results demonstrate that the existence of alternating charge blocks governs LLPS in vitro and indicate that neither the exact position of the charged residues nor a negative shift of the net charge is a critical determinant.

Next, we tested whether the Ki-67 RD could form a liquid phase on an artificial chromosome surface in vitro. Diethylaminoethyl (DEAE) beads were coated with double-stranded DNA and incubated with LR-fused R12 (LR is necessary for DNA binding). All these constructs bound to the DNA-coated beads; however, both the phosphomimetic mutant (Pm9) and the charge-block mimetic mutant (CBm-3) assembled on the beads stronger than the wild type (WT) (Fig. 3a,b). As the mutation affected neither the affinity

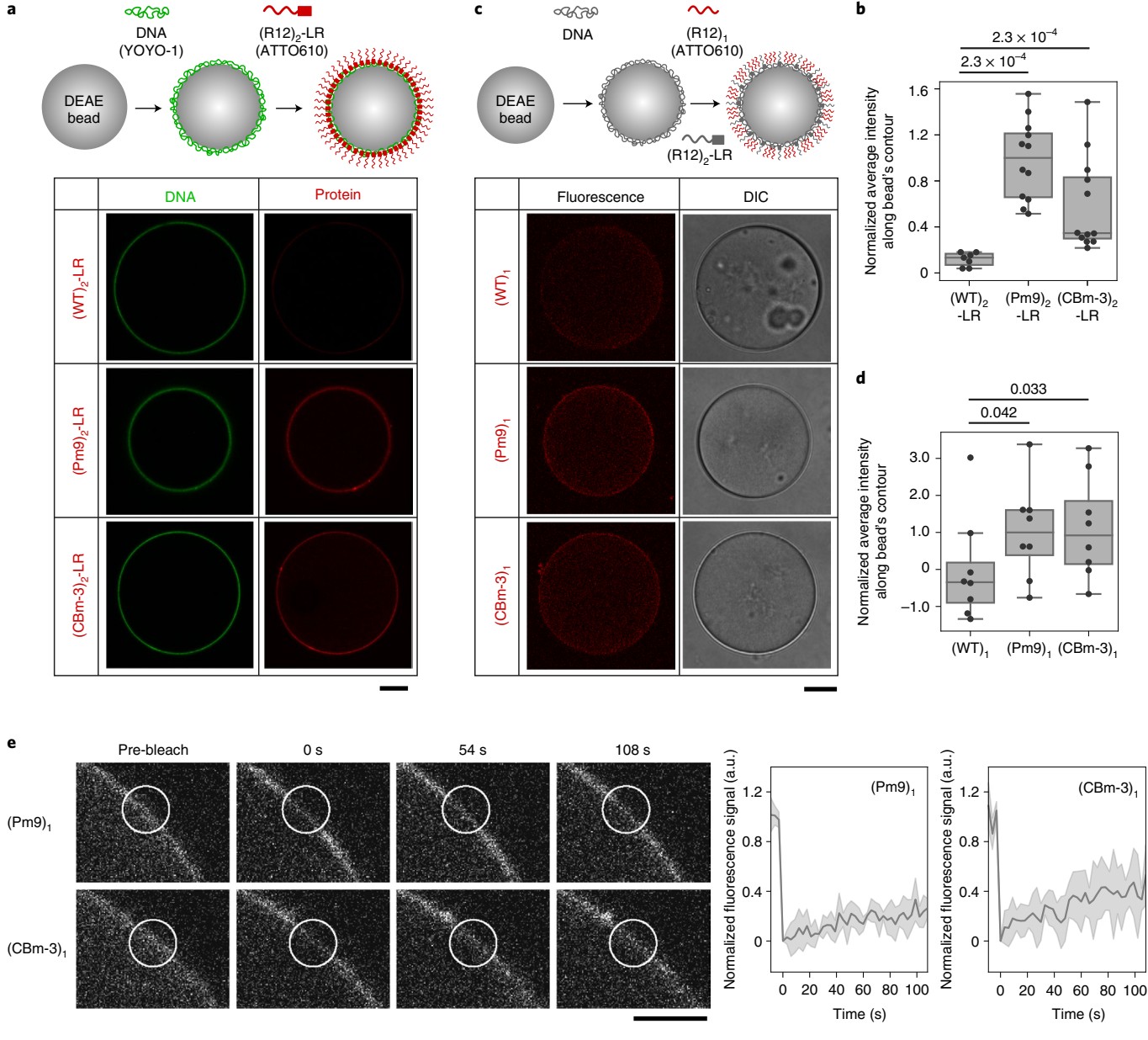

**Fig. 3 | Ki-67 RD forms a liquid phase on a DNA-coated bead. a,b,** λDNA-bound DEAE beads were incubated with purified R12 ((WT)₂, (Pm9)₂ and (CBm-3)₂) fused with LR. DNA was labelled with YOYO-1 (green) and LR-fused proteins were labelled with ATTO-610 (red). Fluorescence images are shown (**a**). The signal intensity of the protein layer along the bead's contour was measured and normalized to the median value of (Pm9)₂-LR (**b**) (n = 7 for (WT)₂-LR and (CBm-3)₂-LR) and n = 12 for (Pm9)₂-LR and (CBm-3)₂-LR). Box plot shows the minimum and maximum value by whiskers, 25% and 75% percentile by box boundaries and median by bar inside the box. Numbers above graph indicate P values by one-tailed Mann–Whitney U test. Scale bar, 15 μm.
**c–e,** Assembly and behaviour of LR-free R12 on a bead. λDNA-bound DEAE beads were incubated with purified LR-fused R12 ((WT)₂, (Pm9)₂ and (CBm-3)₂), together with ATTO610-labelled LR-free R12 ((WT)₁, (Pm9)₁ and (CBm-3)₁). Fluorescence and differential interference contrast (DIC) images are shown (**c**). Signal intensity of the protein layer along the bead's contour was measured and normalized to the median value of (Pm9)₁ (**d**) (n = 8). Box plot shows the minimum and maximum value by whiskers, 25% and 75% percentile by box boundaries and outlier by dot above upper whisker. Numbers above graph indicate P values by one-tailed Mann–Whitney U test. Representative fluorescence images acquired during the FRAP analysis are shown (**e**). The region surrounded by a circle was bleached and the fluorescence intensity in the area was measured and plotted with a solid line of mean and a shade of standard deviation (right panels) (n = 4 for (Pm9)₁ and n = 5 for (CBm-3)₁). Scale bar, 15 μm. Source numerical data are available in source data.

for DNA (Extended Data Fig. 5a) nor the efficiency of fluorescent labelling (Extended Data Fig. 5b), the protein layer observed in the Pm9 and CBm-3 sequences indicated stronger interactions among the RDs. To confirm this, fluorescently labelled LR-free (Pm9)₁ was added to the DNA beads together with non-labelled (Pm9)₂-LR. The labelled (Pm9)₁ was incorporated into the layer only in the presence of (Pm9)₂-LR (Extended Data Fig. 5c). The incorporation

of LR-free molecules was stronger in Pm9 and CBm-3 than in WT (Fig. 3c,d). The liquid-like property of the protein layer was further confirmed by fluorescence recovery after photobleaching (FRAP) analysis of the LR-free molecules in the layer (Fig. 3e and Extended Data Fig. 2b).

We investigated how phosphorylation of the Ki-67 RD and its LLPS-promoting activity were related to the formation of the

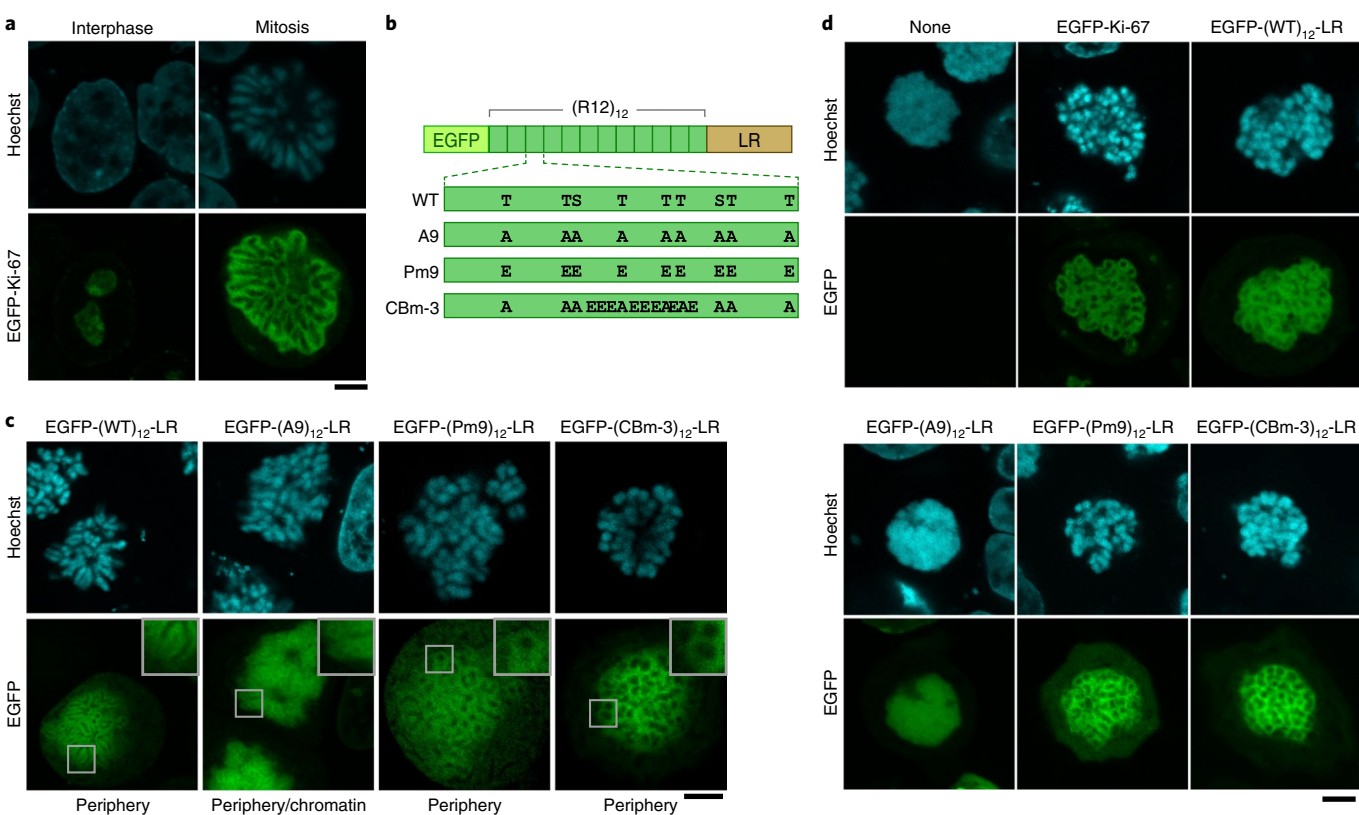

**Fig. 4 | Mitotic phosphorylation, phosphomimetic mutants and a charge-block mimetic mutant of R12 induce the formation of the mitotic chromosome periphery. a**, EGFP-tagged full-length human Ki-67 expressed in HeLa cells. DNA was stained with Hoechst 33342. Scale bar, 5 μm. **b**, Schematic illustration of EGFP- and LR-fused homogeneous R12 repeat constructs. The amino-acid sequence of a single repeat unit is shown in Fig. 2a. In the CBm-3 mutant, nine mitotic phosphosites were replaced with alanine to avoid mitotic phosphorylation. **c**, Localization of R12 repeat constructs in mitotic HeLa cells. DNA was stained with Hoechst 33342. Magnified images (square (3.4 × 3.4 μm)) are shown in the insets. Scale bar, 5 μm. **d**, Expression of R12 constructs in Ki-67-KO cells. DNA was stained with Hoechst 33342. Scale bar, 5 μm.

chromosome periphery in vivo. Enhanced green fluorescent protein (EGFP)-Ki-67 (full length) expressed in HeLa cells localized in the interphase nucleoli and at the mitotic chromosome periphery (Fig. 4a). The liquid-like behaviour of Ki-67 at the chromosome periphery was confirmed by treating the cells with ammonium acetate[36], as well as by FRAP analysis (Extended Data Fig. 6a–c). The homogeneous repeats of R12 were fused with LR (required for binding to chromosomes; Extended Data Fig. 6d) and expressed in HeLa cells (Fig. 4b). These homogeneous repeats localized at the mitotic chromosome periphery in a repeat-number-dependent manner (Extended Data Fig. 6e). Replacement of all nine mitotic phosphosites with non-phosphorylatable residues ((A9)₁₂-LR; Fig. 4b), which nearly completely abolished mitotic phosphorylation (Extended Data Fig. 7a), severely diminished the peripheral localization compared with that of the WT constructs containing the same number of repeats (Fig. 4c and Extended Data Figs. 6e and 7b). In contrast, the phosphomimetic mutant (Pm9)₁₂-LR, as well as a charge-block mimetic mutant showing similar LLPS in the in vitro droplet assay ((CBm-3)₁₂-LR) (Fig. 2b), localized at the chromosome periphery (Fig. 4c). These results indicate that the block-polyampholyte repeat is necessary and sufficient for localization at the mitotic chromosome periphery.

The ability to form a functional chromosome periphery was examined using Ki-67 knockout (KO) cells. In these cells, mitotic chromosomes coalesced, forming a large single mass of chromatin (Fig. 4d). The depletion caused a slight mitotic delay in some cell lines[37], although other cell lines showed proliferation similar to that of the WT counterpart[38]. The expression of full-length Ki-67

in KO cells nearly completely rescued the phenotype (Fig. 4d). Three-dimensional morphological analysis of the mitotic chromosomes indicated that chromosomes were more dispersed in the rescued cells than in non-rescued cells (Extended Data Fig. 7c). The recovery of the chromosome periphery was confirmed by the segregated localization of protein and DNA on mitotic chromosomes (Extended Data Fig. 7d,e). Notably, WT R12 ((WT)₁₂-LR), but not the non-phosphorylatable form ((A9)₁₂-LR), rescued the KO phenotype (Fig. 4d), although to a lower level than the full-length Ki-67 did (Extended Data Fig. 7c,d), probably because of its lower repeat number. Homogeneous repeat of R7 ((R7)₁₂-LR) not only localized at the chromosome periphery of mitotic HeLa cells, but also rescued the phenotype of Ki-67-KO cells (Extended Data Fig. 7c,d,f,g), suggesting that the number of repeats rather than specific amino-acid sequence of R12 is important for the formation of the chromosome periphery. Notably, not only the phosphomimetic mutant ((Pm9)₁₂-LR), but also the charge-block mimetic mutant ((CBm-3)₁₂-LR) of R12 rescued the KO phenotype (Fig. 4d and Extended Data Fig. 7c, d). The charge-block mimetic mutant carrying the same net charge as A7 but larger charge blockiness (CBm-7; net charge +5 and $B_{LC}$ 35) also rescued the KO phenotype (Extended Data Fig. 7e), indicating that charge blockiness, but not a negative shift of the net charge, is critical for the formation of the chromosome periphery. Together, these results demonstrate that the alternating charge blocks of the Ki-67 RD are necessary and sufficient for efficient LLPS in vitro and for forming the functional mitotic chromosome periphery in vivo.

Next, we investigated whether mitotic phosphorylation regulates the intracellular dynamics of other phosphoproteins by changing

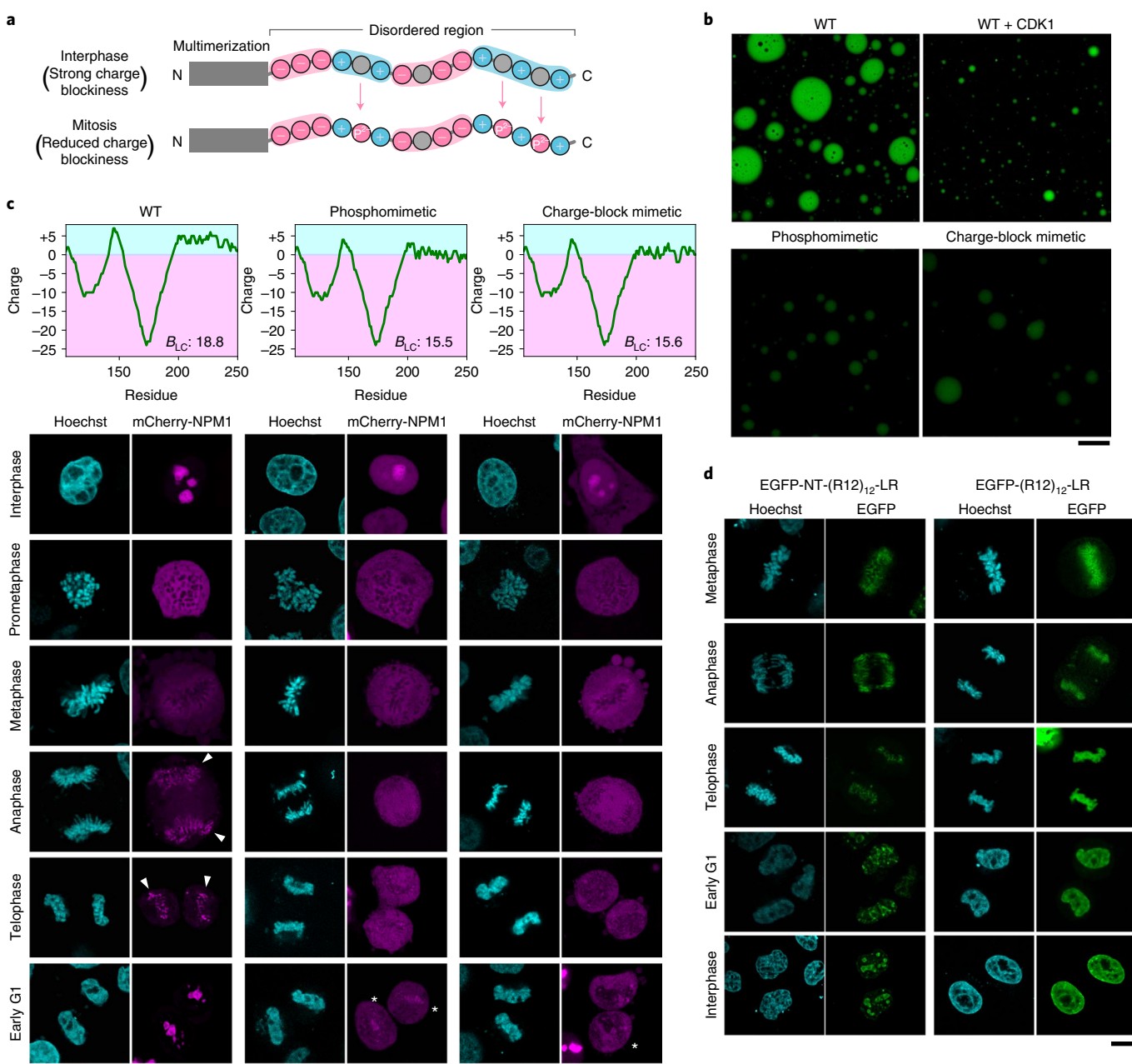

**Fig. 5 | Mitotic hyperphosphorylation reduces charge blockiness and the propensity for LLPS of the nucleolar phosphoprotein NPM1. a**, Schematic illustration of how hyperphosphorylation of the IDR of NPM1 reduces charge blockiness. **b**, In vitro LLPS assay of recombinant NPM1 IDR. WT, CDK1-treated WT, phosphomimetic mutant, and a charge-block mimetic mutant (amino-acid sequences are shown in Extended Data Fig. 8b). Scale bar, 50 μm. **c**, Mitotic localization of NPM1. mCherry-fused WT, phosphomimetic mutant and charge-block mimetic mutant of full-length NPM1 were expressed in HeLa cells. DNA was counterstained with Hoechst 33342. The charge plot (window size: 25 amino acids) and $B_{LC}$ value are also indicated. Scale bar, 10 μm. Arrowheads and asterisks indicate cells in which NPM1 started to re-assemble among WT- and mutant-expressing cells, respectively. **d**, Mitotic localization of LR domain-fused 12 tandem repeats of Ki-67 R12 with or without N-terminal domain (1-639) (EGFP-NT-(R12)$_{12}$-LR, EGFP-(R12)$_{12}$-LR). DNA was counterstained with Hoechst 33342. Scale bar, 10 μm.

the charge blockiness. NPM1 localizes in the GC region of the nucleoli and is heavily phosphorylated upon entry into mitosis[16]. Eleven mitosis-specific phosphorylation sites were identified in a long stretch of the IDR (Extended Data Fig. 1b). Comparison of the charge distributions revealed that the dephosphorylated (interphase) form has a strong block-polyampholytic charge distribution, whereas the hyperphosphorylated (mitotic) form loses positive charge blocks (Extended Data Fig. 1b). Therefore, mitotic hyperphosphorylation may reduce the propensity of NPM1 for LLPS, an effect opposite to

that observed for Ki-67 (Fig. 5a). Indeed, CDK1-treated NPM1-IDR and the phosphomimetic mutant of the 11 mitotic phospho-sites reduced the formation of liquid droplets in vitro (Fig. 5b and Extended Data Fig. 8a–c). Notably, a charge-block mimetic mutant of NPM1 showed reduced droplet formation (Fig. 5b and Extended Data Fig. 8b,c), demonstrating that mitotic phosphorylation suppresses LLPS of NPM1 by reducing its charge blockiness.

We found that the intracellular dynamics of NPM1 during mitosis were also modified by phosphomimetic and charge-block

mimetic mutations (Fig. 5c). NPM1 diffused from the nucleoli into the cytoplasm when the cell entered mitosis and localized mainly in the cytoplasm, with weak localization around the chromosome periphery. It re-appeared at the chromosome periphery in anaphase and eventually assembled into many small loci in telophase (arrowheads in Fig. 5c), finally fusing to form several nucleoli in early G1 phase. Phosphomimetic and charge-block mimetic mutants localized not only in the nucleoli, but also in the nucleoplasm and cytoplasm in interphase cells (Fig. 5c). Similar to the WT, the mutants diffused into the cytoplasm during prophase and metaphase but did not re-assemble at the chromosome periphery even in anaphase and telophase and remained in the cytoplasm throughout mitosis. Finally, they started to localize to the nucleoli in early G1 phase (asterisk in Fig. 5c). These results demonstrate a close correlation between the block-polyampholytic property of NPM1 and its intracellular dynamics during mitosis.

As Ki-67 directly or indirectly interacts with NPM1 via its N-terminal conserved domain (Extended Data Fig. 8d), we investigated how the opposing effects of mitotic phosphorylation on these proteins are integrated to determine their behaviour during mitosis. The homogeneous repeat construct of Ki-67 ($(R12)_{12}$-LR) localized exclusively in the nucleoplasm in interphase cells (interacting with the chromosome via LR), and addition of the N-terminal domain (1−639) (NT-$(R12)_{12}$-LR) directed it to the perinucleolar region (an interface between the nucleoli and nucleoplasm) (Fig. 5d), suggesting that Ki-67 bridges NPM1 and the chromosome. Ki-67 constructs were localized at the chromosome periphery in prophase and metaphase regardless of the presence of the N-terminal domain (Fig. 5d). In this period, the interaction between Ki-67 and NPM1 was severely abrogated (Extended Data Fig. 8d). When the interaction between Ki-67 and NPM1 recovered in anaphase and telophase, NT-$(R12)_{12}$-LR re-assembled with NPM1 and finally localized in the perinucleolar region, whereas $(R12)_{12}$-LR did not associate with NPM1 and was redistributed from the periphery to the entire chromosome until the end of mitosis (Fig. 5d), demonstrating that the perinucleolar localization of Ki-67 requires interaction with NPM1. Overall, these results suggest a reciprocal regulatory mechanism of the nucleoli and chromosome periphery during the cell cycle.

We demonstrated that mitotic hyperphosphorylation of the RD of Ki-67 enhanced its charge blockiness and promoted its LLPS to form the periphery of mitotic chromosomes. Notably, a mutant mimicking the mitotically phosphorylated charge blocks not only displayed strong LLPS in vitro (Fig. 2b,c), but also rescued deficiencies observed in Ki-67-KO cells (Fig. 4d). Thus, the occurrence of alternating charge blocks, rather than the exact position of negative charges, plays an important role in chromosome periphery formation via LLPS. This 'fuzzy' regulatory mechanism, regulated by charge blockiness, clearly contrasts the conventional 'tight' mechanism via which site-specific addition of a phosphate group modulates stereospecific interactions between structured proteins or domains. This mechanism is distinct from a previously reported mechanism of multiple phosphorylation, in which stepwise accumulation of multiple phosphate groups confers high cooperativity or ultrasensitivity in enzymatic activation (such as that for the CDK1 inhibitor Sic1 (ref. [39])). The proposed mechanism also explains why protein phosphorylation frequently occurs at multiple neighbouring residues located in IDRs[40–42].

Notably, the mode of cell cycle-dependent regulation is reversed in NPM1: mitotic hyperphosphorylation of NPM1 reduces, rather than enhances, its charge blockiness and suppresses its strong LLPS propensity, leading to dissolution of the nucleoli (Fig. 5). The opposing effects of mitotic phosphorylation of Ki-67 and NPM1 on their LLPS, together with their cell cycle-specific interaction (Extended Data Fig. 8d), underlie the morphological changes of nucleoli and the chromosome periphery during mitosis (Extended Data Fig. 8e). Our phosphoproteomic analyses demonstrated that mitotic

hyperphosphorylation changes the charge blockiness of IDRs in several other nucleolar proteins (Extended Data Fig. 9), suggesting that their LLPS is also regulated by charge blockiness-enhancing or charge blockiness-reducing phosphorylation. Notably, the average charge block size converges to ~30–40 amino acids, indicating that this charge block size is the most suitable for a polypeptide to undergo regulatable LLPS in an intracellular milieu.

In summary, the blockiness-enhancing or blockiness-reducing phosphorylation described here is distinct from previously reported phosphorylation-based regulatory mechanisms and may represent a general mechanism that regulates the behaviour of a broad spectrum of phosphoproteins and assembly and disassembly of intracellular membraneless organelles and structures.

## Online content

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

## Methods

**Materials.** All chemical reagents used in this study were purchased from Nacalai Tesque unless otherwise indicated.

**Protein sequence analysis.** The mitotic phosphosites in human Ki-67 and NPM1 were reported in our previous study[16]. The human Ki-67 and NPM1 sequences were obtained from the UniProt database (accession numbers P46013 and P06748, respectively). Charge distribution was calculated as the sum of the charges (Arg and Lys, +1; Glu and Asp, −1; phospho-Ser and phospho-Thr, −2) in the indicated window range. A charge block was designated when the area of the charge plot (window size: 35 amino acids) was larger than 20. Multiple sequence alignment was performed using Clustal Omega (https://www.ebi.ac.uk/Tools/msa/clustalo/). $D_{seg}$ and $B_{LC}$ were calculated with the equation described in Supplementary Note. We chose to use $B_{LC}$ over other charge patterning parameters such as sequence charge decoration[43] and $\kappa$ (ref. [44]), because $B_{LC}$ employs a larger block size, more appropriately capturing the longer charge tracts that we believe are important in Ki-67.

**DNA construction.** Complementary DNA of human Ki-67 (short isoform) was obtained as described previously[45]. Fragments of the WT Ki-67 RD, LR domain (amino acids 2,578–2,896) and N-terminal domain (amino acids 1–639 (corresponding to amino acids 1–135 and 496–999 of the long isoform)) as well as human NPM1 (amino acids 105–250) were amplified by PCR and subcloned into pET28a(+) (Novagen) for expression in *Escherichia coli* and/or pEGFP for mammalian expression. The nucleotide sequences of primers are presented in Supplementary Table 1. cDNA fragments encoding phosphomimetic mutants and charge-block mimetic mutants of Ki-67 and NPM1 were synthesized at Thermo Fisher Scientific. The amino-acid sequences of all mutants that were used in this study are presented in Supplementary Table 2. For bacterial expression, the codon usage was optimized for *E. coli* without changing the amino-acid sequence. To generate tandem repeats of Ki-67 R12 and R7, the DNA fragment encoding R12 was cleaved out from the expression vector with Xho I and Sal I digestion and ligated into the same expression construct digested with Sal I. Through these procedures, the number of R12 was increased up to 12. All of these homogeneous repeat constructs contained the linker sequence 'GHTEESVEDD' between each repeat unit.

**Cell culture, synchronization and transfection.** HeLa cells (ATCC, CCL-2.2) were cultured in Dulbecco's modified Eagle's medium (DMEM, Sigma-Aldrich) supplemented with 10% foetal bovine serum (FBS, Gibco) at 37 °C in the presence of 5% $CO_2$. The Ki-67 KO HCT116 cell line was described previously[46] and was cultured in high-glucose DMEM supplemented with 10% FBS and penicillin–streptomycin at 37 °C in the presence of 5% $CO_2$. Cells were transfected with the plasmids using PEI-MAX (Polysciences). To induce mitotic arrest, cells were treated with 0.2 μM nocodazole for 15 h. For microscopic observation, cells on a cover glass (Matsunami Glass) were fixed with 4% paraformaldehyde at room temperature for 15 min and mounted with Vectashield (Vector Laboratories) containing Hoechst 33342.

**Protein purification.** *E. coli* cells (BL21-CodonPlus(DE3)-RIL, Agilent Technologies) harbouring the expression vector for hexahistidine (His$_6$)-tagged Ki-67 fragments were cultured in Luria–Bertani medium. Protein expression was induced by adding 0.1–1.0 mM IPTG, and the cultures were further incubated at 20 °C or 37 °C overnight. The cells were collected by centrifugation (5,000*g*, 15 min, 20 °C) and stored at −80 °C until use. For purification under a denaturing condition, the cell pellet was subjected to two rounds of freeze–thaw cycles and was finally dissolved in urea-containing buffer (8 M urea, 10 mM Tris–HCl, 100 mM NaH$_2$PO$_4$, 5 mM 2-mercaptoethanol and 10 mM imidazole, pH 8.0) at 4 °C for overnight to 2 days. The lysate was centrifuged (10,000*g*, 4 °C, 30 min) and the supernatant was collected and mixed with Ni-NTA agarose beads (Qiagen). The beads were gently agitated at 4 °C for 1 h, washed with wash buffer (8 M urea, 10 mM Tris–HCl, 100 mM NaH$_2$PO$_4$, 10 mM 2-mercaptoethanol and 20 mM imidazole, pH 8.0), and His$_6$-tagged proteins were eluted with elution buffer (8 M urea, 10 mM Tris–HCl, 100 mM NaH$_2$PO$_4$, 5 mM 2-mercaptoethanol and 100, 300 or 500 mM imidazole, pH 8.0). Eluted proteins were sequentially dialysed at 4 °C against dialysis buffer 1 (0.1% (v/v) trifluoroacetic acid (TFA) and 2 mM 2-mercaptoethanol) for 3 h, dialysis buffer 2 (0.05% (v/v) TFA and 2 mM 2-mercaptoethanol) for 3 h or overnight and, finally, dialysis buffer 3 (0.05% (v/v) TFA) for 3 h. The purified proteins were lyophilized (FDU-2200, EYELA) and stored at 4 °C.

To purify His$_6$-tagged protein under a native condition, the cell pellet was dissolved in lysis buffer (50 mM Tris–HCl, 500 mM NaCl, 1 mM MgCl$_2$, 2 mM 2-mercaptoethanol, 1 mM phenylmethylsulfonyl fluoride, 20 mM imidazole, lysozyme and DNase I, pH 7.4) and subjected to three rounds of quick freeze–thaw cycles. The cell debris was removed by centrifugation (12,000*g*, 20 min, 4 °C) and the supernatant was mixed with Ni-NTA beads at 4 °C for 1 h. The beads were then washed with wash buffer (50 mM Tris–HCl, 500 mM NaCl, 2 mM 2-mercaptoethanol and 20 mM imidazole, pH 7.4) three times and eluted stepwise with increasing concentrations (50, 100, 200, 300, 400 and 500 mM) of imidazole in wash buffer. The eluted protein was dialysed against low-salt

buffer (50 mM Tris–HCl, 50 mM NaCl and 2 mM 2-mercaptoethanol, pH 7.4) at 4 °C for 3 h and then subjected to ion-exchange chromatography (Hi-Trap Q, GE Healthcare). The eluted fraction was collected, dialysed against assay buffer (50 mM HEPES and 100 mM NaCl, pH 7.4) at 4 °C, concentrated using Amicon centrifugal filters (Millipore) and stored at −80 °C in small aliquots. The amount of RNA contamination in the individual protein preparations was quantified using a fluorescent probe for RNA (QuantiFluor RNA, Promega) and was 0.1–0.7% (w/w).

**In vitro LLPS assay.** Lyophilized protein was dissolved into dissolving buffer (2 M guanidine hydrochloride, 100 mM Tris–HCl pH 8.0 and 10 mM HEPES) to a final concentration of 4 mM. For fluorescence microscopic observation, protein was incubated with 10 μM ATTO488-maleimide (ATTO-TEC) at room temperature for 1 h and then with 5 mM dithiothreitol at room temperature for 1 h or at 4 °C overnight. The labelled protein solution was diluted in droplet buffer (50 mM HEPES, 100 mM NaCl and 15% (w/v) PEG3350 (Sigma-Aldrich), pH 7.4) at a 1:100 ratio, incubated at room temperature for 30 min and transferred to a 96-well clear-bottom plate (Greiner Bio-One) for microscopic observation (FV3000, Olympus). The final concentration of protein was 40 μM unless otherwise indicated. For protein with multiple repeats, the final protein concentration is indicated in the figure legend. For the turbidity assay, protein in dissolving buffer was sequentially diluted with the same buffer, and then mixed with droplet buffer at 1:50. The mixture was incubated at room temperature for 10 min and transferred to a microcuvette. The optical density at 600 nm (OD$_{600}$) was measured using a V-630 spectrophotometer (JASCO). The $C_{sat}$ value was defined by the concentration at which the turbidity was at half-maximal value[10]. The data obtained were fitted using the equation (Supplementary Note) in OriginPro (v.9.8).

**In vitro phosphorylation by CDK1.** Protein purified under the native condition was incubated with CDK1–cyclin B1 (Abcam) in kinase buffer (50 mM HEPES–NaOH, 100 mM NaCl, 10 mM MgCl$_2$ and 50 μM ATP, pH 7.4) at room temperature for 30 min. Phosphorylation was confirmed by SDS–PAGE containing Phos-tag acrylamide (Fuji Film, Wako). Protein was then incubated with 10 μM ATTO488-maleimide for 1 h, mixed with droplet buffer at a 1:100 ratio and transferred to a 96-well clear-bottom plate for microscopic observation.

**DNA bead assay.** Lyophilized proteins were dissolved into dissolving buffer and labelled with ATTO610-maleimide (ATTO-TEC) as described above, if necessary. λDNA (Takara) (26.25 μg) was attached to 1.65 μl DEAE sepharose beads (DEAE Sepharose Fast Flow, GE Healthcare) and stained with YOYO-1 (Thermo Fisher Scientific) if necessary in bead buffer (50 mM HEPES and 100 mM NaCl, pH 7.4). The bead suspension was then incubated with protein (0.8 μM) in a 96-well clear-bottom plate (Greiner Bio-One) at room temperature for 2.5 h. To examine the incorporation of LR domain-free RD, the ATTO610-labelled protein was added to a final concentration of 40 μM after 2.5 h of incubation of the DNA beads with LR-fused repeat protein. For the FRAP assay, ATTO610 signal on the beads was bleached using 561-nm laser light and observed by time-lapse imaging (FV3000, Olympus).

**Microscopic observation, image processing and image analysis.** For the observation of fluorescence signals, a confocal laser-scanning microscope system (FV3000, Olympus) was used. For live-cell imaging, a stage chamber (Tokai Hit) was used to maintain the temperature and moisture and CO$_2$ levels. Phenol red-free DMEM supplemented with 10% FBS and 1 μg ml$^{-1}$ Hoechst 33342 was used to visualize chromosomes if necessary. The images obtained were processed and analysed using MetaMorph (Molecular Devices), Fiji[47] or Python 2 or 3 with add-on libraries (Numpy, Scipy, Pandas, Matplotlib, OpenCV and seaborn).

**Statistics and reproducibility.** Methods of statistical analysis and sample size are indicated in the figure legends.

No statistical methods were used to predetermine sample size. Sample sizes were estimated empirically on the basis of pilot experiments and previously performed experiments with similar setup to provide sufficient sample sizes for statistical analysis. No data were excluded from the analyses with the exception of the image analysis of droplet. For the quantification of droplet, ones with higher eccentricity than criterion (0.7) were excluded. For droplet assay, sample and measurement order was randomized. The area of microscopic observation was randomly determined. The investigators were not blinded to allocation during experiments and outcome assessment. Turbidity assay was performed three times, and microscopic observation, gel electrophoresis, western blotting and electrophoretic mobility shift assay were performed at least twice.

**Reporting Summary.** Further information on research design is available in the Nature Research Reporting Summary linked to this article.

## Data availability

Amino-acid sequences of human Ki-67 and human NPM1 can be obtained from UniProt database (accession number P46013 for Ki-67 and P06748 for NPM1). Source data are provided with this paper. All other data supporting the findings of this study are available from the corresponding author on reasonable request.

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

## Acknowledgements

This research was supported by Japan Agency for Medical Research and Development (AMED) under grants JP20fk0108144 and JP20wm0325009 (to S.H.Y.), the Joint Usage and Joint Research Programs, Institute of Advanced Medical Sciences, Tokushima University (to S.H.Y. and H.K.), a JSPS Grant-in-Aid for JSPS Fellows (17J09002 to H.Y.) and JSPS Grants-in-Aid for Scientific Research (20K06649 to M.T., and 18H05276 and 20H05938 to T.H.). We thank S. Dodo for technical assistance, and T. Sakaue and Y. Norizoe for useful discussions on phase separation.

## Author contributions

H.Y. and H.K. contributed to data acquisition, analysis and interpretation. M.T. and T.H. contributed to data interpretation. S.H.Y. conceptualized and designed the study, contributed to data acquisition, analysis and interpretation, and wrote the manuscript with input from all other authors.

## Competing interests

The authors declare no competing interests.

## Additional information

**Extended data** is available for this paper at https://doi.org/10.1038/s41556-022-00903-1.

**Correspondence and requests for materials** should be addressed to Shige H. Yoshimura.

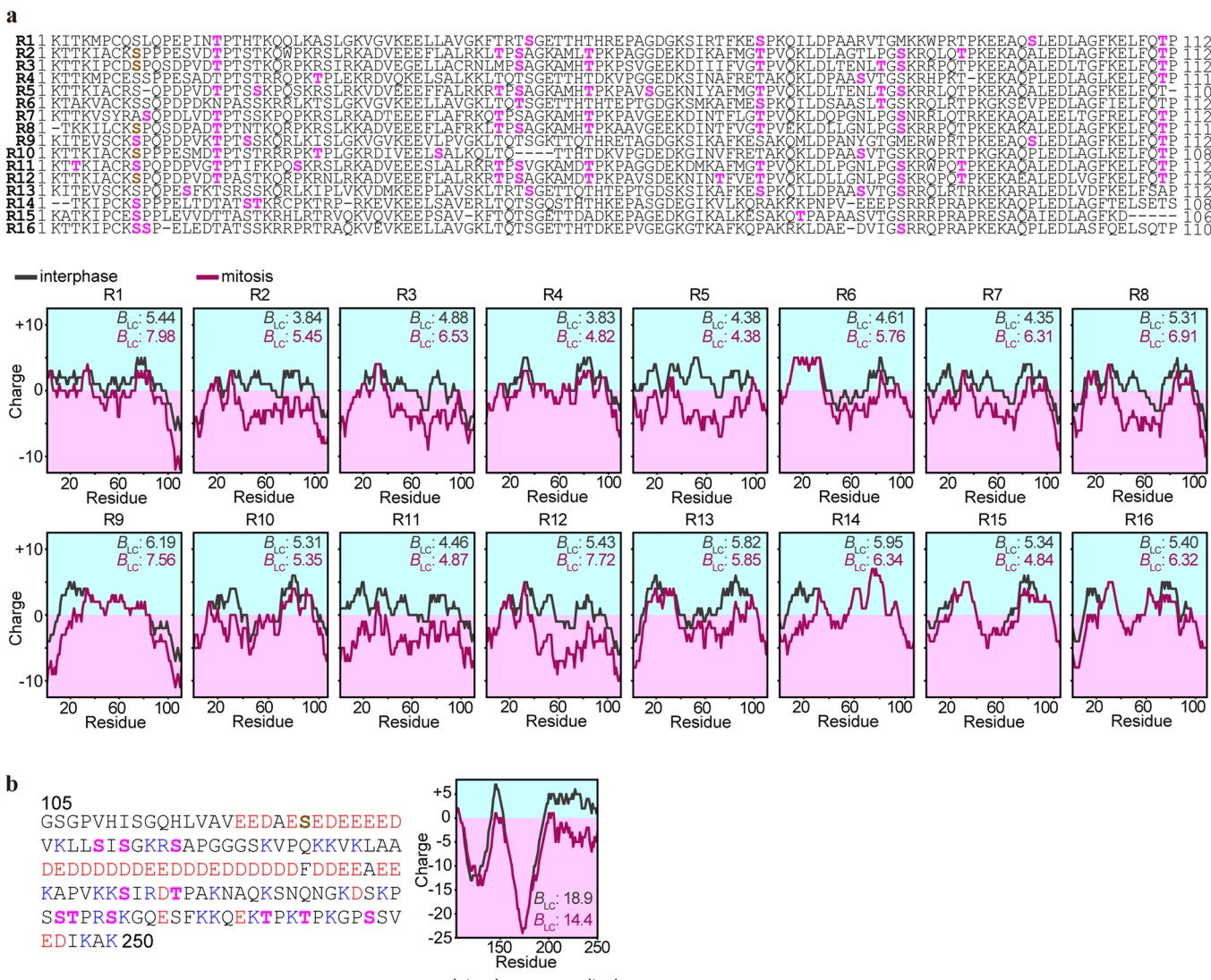

**Extended Data Fig. 1 | Amino acid sequences and charge plots of human Ki-67 and NPM1. a**, Amino-acid sequence alignment of the human Ki-67 repeat domain (RD). Residues in which the phosphorylated fraction increased or decreased during mitosis (quantified by tandem-mass-tag analysis[16]) are shown in magenta and brown, respectively. Charge distributions of individual repeats (R1–R16) in interphase and mitotic forms (window size: 25 amino acids) together with the $B_{LC}$ values (interphase: grey, mitotic: purple) are shown. Most repeats showed larger $B_{LC}$ values in mitotic forms. **b**, Amino-acid sequence of IDR of human NPM1 (amino acids 105–250). Residues in which the phosphorylated fraction increased or decreased during mitosis are shown in magenta and brown, respectively. A charge plot of interphase and mitotic forms (window size: 25 amino acids) and the $B_{LC}$ values (interphase: grey, mitotic: purple) are shown.

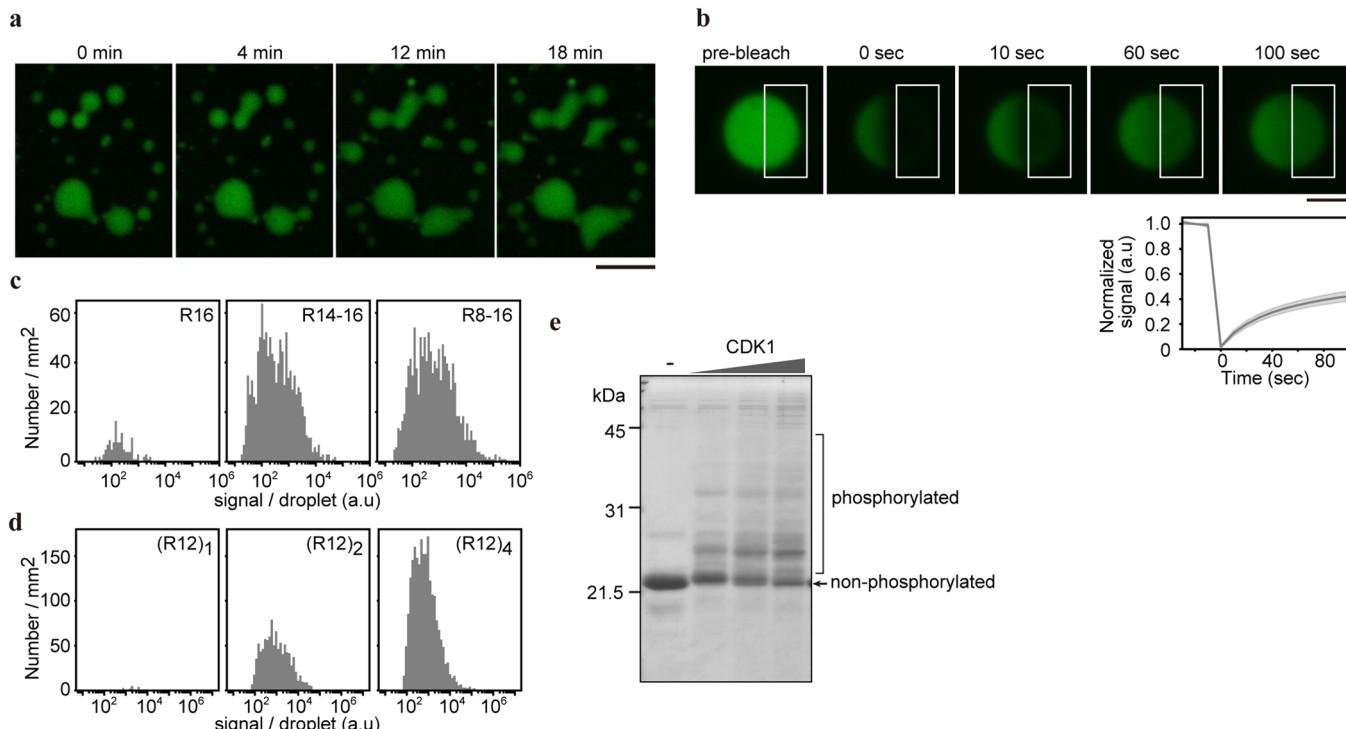

**Extended Data Fig. 2 | LLPS of Ki-67 RD. a**, Liquid-like behaviour of protein droplets formed by Ki-67 RD (15 μM R14-16). Bar, 20 μm. **b**, Fluorescence recovery after photobleaching (FRAP) analysis of a droplet of 40 μM R12. Time-lapse fluorescence images before and after photobleaching are shown. The area indicated by a rectangle was bleached using 488-nm laser light and the fluorescence intensity in the area was measured and plotted (mean ± SD, n = 13) (bottom panel). Bar, 2 μm. **c, d**, Statistical analysis of the fluorescence intensity of the protein droplets. Results from recombinant proteins carrying different RD stretches (c) and increasing copy number of R12 (d) are summarized. **e**, *In vitro* phosphorylation of R12 by CDK1. Purified His$_6$-tagged R12 (4 μg) was incubated with CDK1/cyclin B (0.1, 0.5, 1.0 μg), 1 mM ATP and 10 mM MgCl$_2$ at 25 °C for 60 min and then subjected to sodium dodecyl sulphate-polyacrylamide gel electrophoresis (SDS-PAGE) with a gel containing Phos-tag acrylamide (Fuji Film). Source numerical data and an unprocessed gel are available in Source Data.

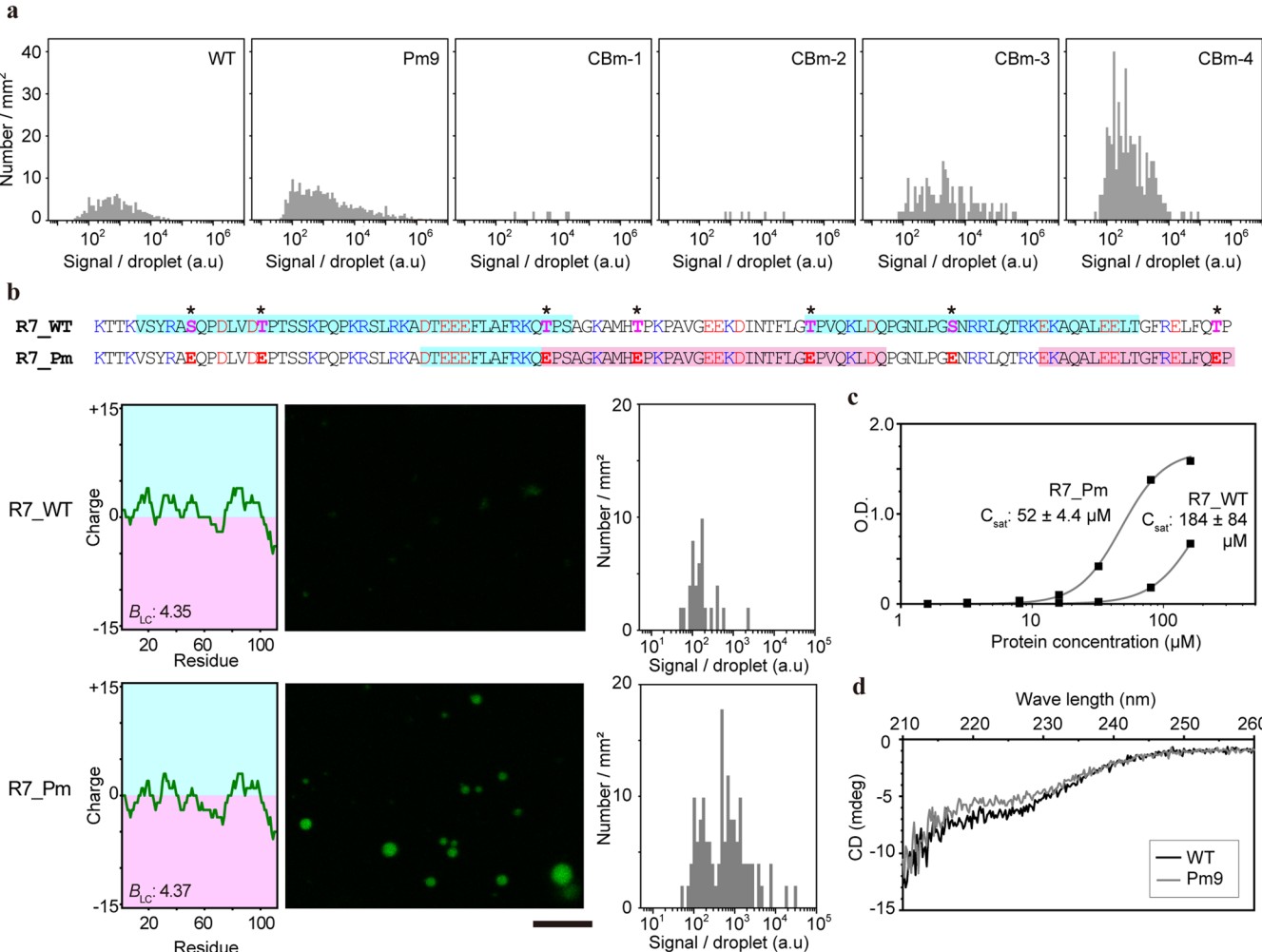

**Extended Data Fig. 3 | LLPS of Ki-67 RD mutants. a**, Statistical analysis of fluorescence intensity of the droplets shown in Fig. 2b. **b**, LLPS assay of R7. Amino-acid sequences of human Ki-67 R7 (wild-type (WT) and phosphomimetic mutant of mitotic phosphosites (Pm)). The mitotic phosphosite is denoted with an asterisk (*). Positive and negative charge blocks (see Methods) are highlighted in cyan and red, respectively. Charge plots (window size: 25 amino acids) with $B_{LC}$ values (left panels) are shown. Fluorescence images of an *in vitro* droplet assay and the statistical analysis of the fluorescence intensity are presented (middle and right panels). Bar, 30 μm. **c**, Turbidity assay of R7_WT and R7_Pm. One representative result is shown (out of three). The $C_{sat}$ values are presented as mean ± standard deviation from three independent experiments. **d**, Circular dichroism (CD) spectrum of purified R12 fragments. Purified R12 (WT and Pm9) were subjected to CD spectrum measurement in 50 mM HEPES, 100 mM NaCl (pH. 7.4) at 25 °C. Source numerical data are available in Source Data.

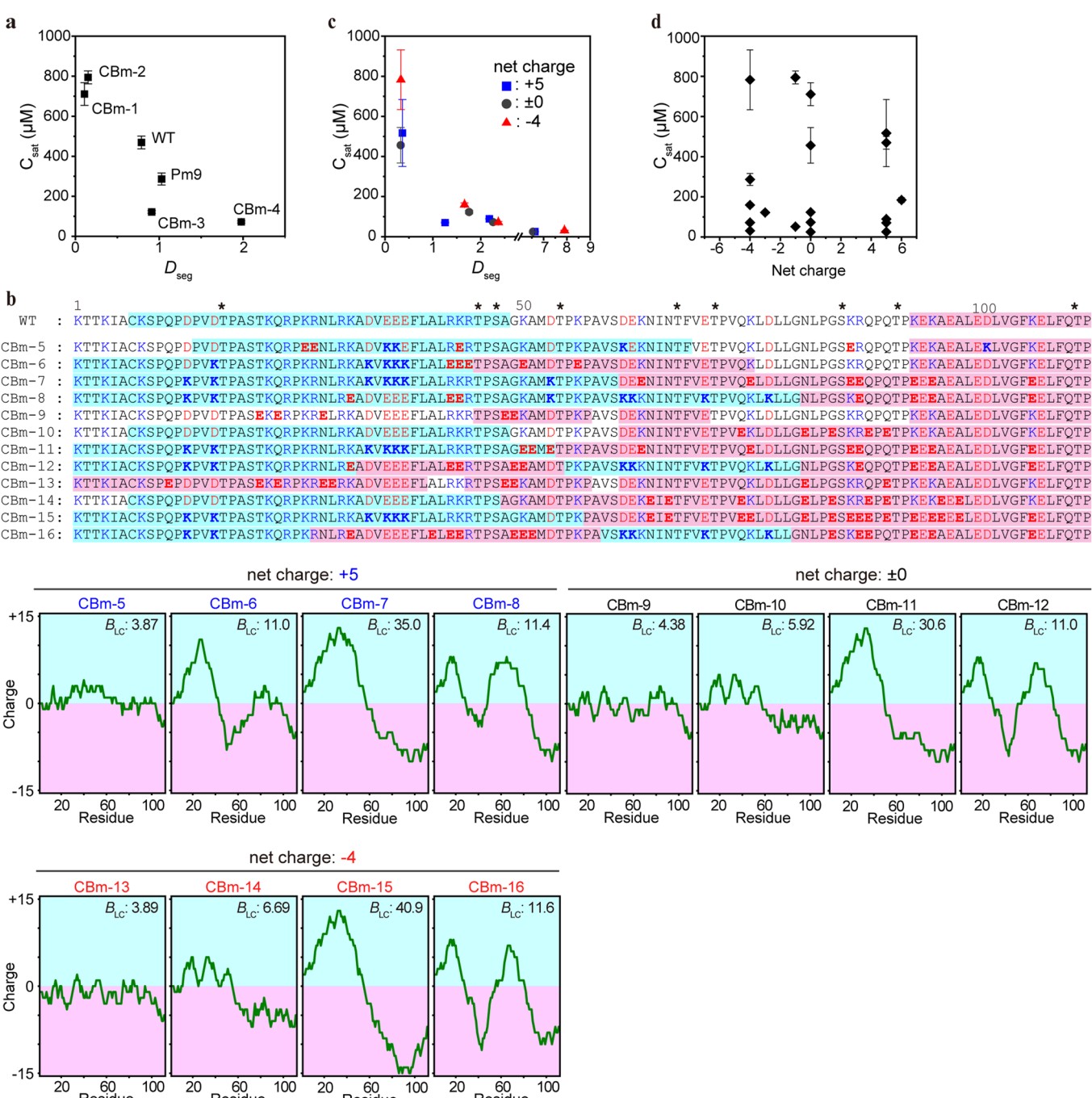

**Extended Data Fig. 4 | LLPS is governed by charge blockiness rather than net charge. a**, Relationship between $C_{sat}$ and $D_{seg}$. The $C_{sat}$ values of the mutants and WT of R12 were obtained by the turbidity assay described in Fig. 2c and plotted against $D_{seg}$. Error bars reflect standard deviation of the mean (n = 3 independent measurements). **b**, Amino-acid sequences of CBm-5–16 and charge plots (window size: 25 amino acids). The $B_{LC}$ values are presented. Positive and negative charge blocks (see Methods) are highlighted in the amino-acid sequence with cyan and red, respectively. The mitotic phosphosite is denoted with an asterisk (*). **c**, Relationship between $C_{sat}$ and $D_{seg}$ in the mutants shown in **b** (CBm-5 – CBm-16). The $C_{sat}$ values were obtained by the turbidity assay described in Fig. 2d and plotted against $D_{seg}$. Error bars reflect standard deviation of the mean (n = 3 independent measurements). **d**, Relationship between $C_{sat}$ and the net charge of the R12 (WT, Pm9, CBm1–16) and R7 (WT and Pm). The $C_{sat}$ values obtained by the turbidity assay described in Fig. 2c,d were plotted against the net charge. Error bars reflect standard deviation of the mean (n = 3 independent measurements). Source numerical data are available in Source Data.

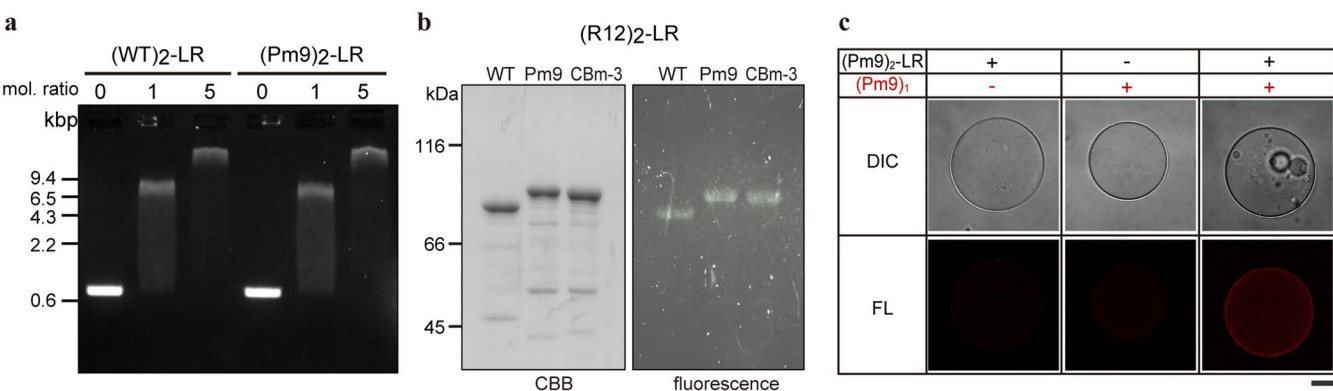

**Extended Data Fig. 5 | Ki-67 RD forms a liquid phase on a DNA-coated bead. a**, WT-LR and Pm9-LR bind to DNA with comparable affinities. Double-stranded linear DNA fragment (~650 bp, 2.5 μg) was incubated with 0, 0.4, or 2 μg of purified WT-LR or Pm9-LR in 50 mM HEPES, 100 mM NaCl (pH 7.4) at 37°C for 15 min and was then subjected to agarose gel electrophoresis. **b**, ATTO610-labelled (WT)$_2$-LR, (Pm9)$_2$-LR and (CBm-3)$_2$-LR were subjected to SDS-PAGE. The fluorescence image (right) and coomassie brilliant blue signal (left) are shown. **c**, λ DNA-bound DEAE beads were incubated with purified LR-fused phosphomimetic mutant of R12 (0.8 μM (Pm9)$_2$-LR), together with 40 μM LR-free (Pm9)$_1$ which was labelled with ATTO-610 and dissolved in droplet buffer (50 mM HEPES, 100 mM NaCl, 15% (w/v) PEG3350 (Sigma-Aldrich), pH 7.4). DIC and fluorescence images (red) are shown. Bar: 15 μm. Unprocessed gels are available in Source Data.

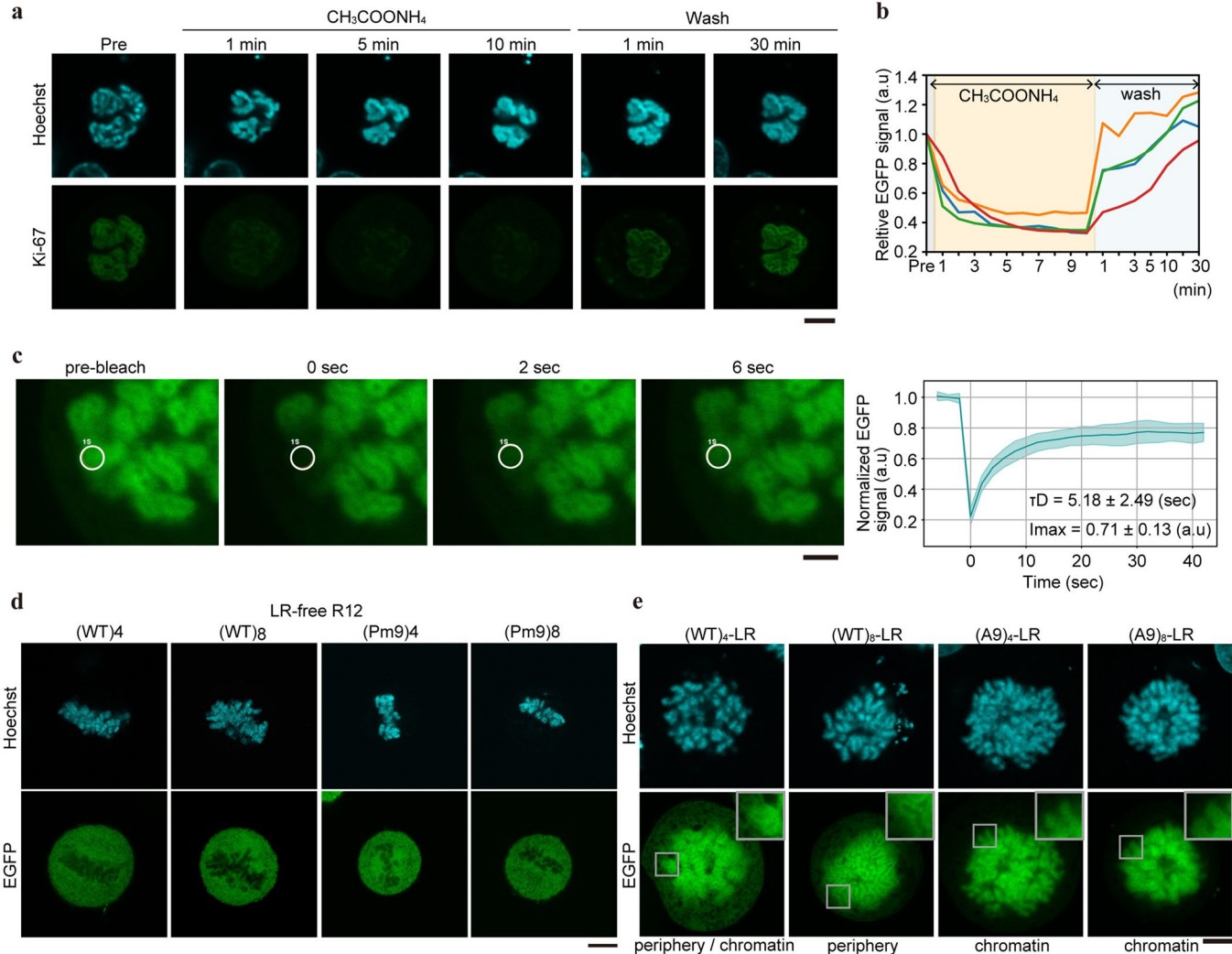

**Extended Data Fig. 6 | Mitotic phosphorylation of Ki-67 induces the formation of liquid phase at the mitotic chromosome periphery. a, b**, Ki-67 reversibly associates and dissociates from mitotic chromosomes upon ammonium acetate treatment. Ki-67-KO cells expressing EGFP-Ki-67 were treated with 100 mM ammonium acetate for 10 min and then returned to normal culture medium after washing with PBS. Time-lapse fluorescence images are shown in (a). Bar, 5 μm. The total EGFP signal intensity at the chromosomes was quantified and plotted along the time course (each line shows the data from an individual cell (n = 4)) (b). **c**, Ki-67 shows liquid-like behaviour on the mitotic chromosome periphery. FRAP analysis of EGFP-(WT)$_8$-LR expressed in HeLa cells. Time-lapse fluorescence images before and after bleaching are shown. The region surrounded by a circle was bleached using 488-nm laser light and the fluorescence intensity in the area was measured and plotted (mean ± SD, n = 38) (right panel). The fitting result from 38 cells is shown. Signal intensity was quantified using MetaMorph (Molecular Devices). Curve-fitting was performed using Python2 or 3 with accompanying libraries (Numpy, Scipy, Pandas, Matplotlib), using the equation described in Supplementary Note. Bar, 2 μm. **d**, Fluorescence images of mitotic HeLa cells expressing LR-free RDs. EGFP-fused WT R12 ((WT)$_4$, (WT)$_8$) and phosphomimetic mutants ((Pm9)$_4$, (Pm9)$_8$), were expressed in HeLa cells. Cells were fixed, stained with Hoechst33342, and observed by confocal fluorescence microscopy. Bar, 5 μm. **e**, Localization of R12 repeat (EGFP-(WT)$_4$-LR, EGFP-(WT)$_8$-LR, EGFP-(A9)$_4$-LR) and EGFP-(A9)$_8$-LR) in mitotic HeLa cells. DNA was stained with Hoechst33342. Magnified images (square (3.5×3.5 μm)) are shown in the insets. Bar, 5 μm. Source numerical data are available in Source Data.

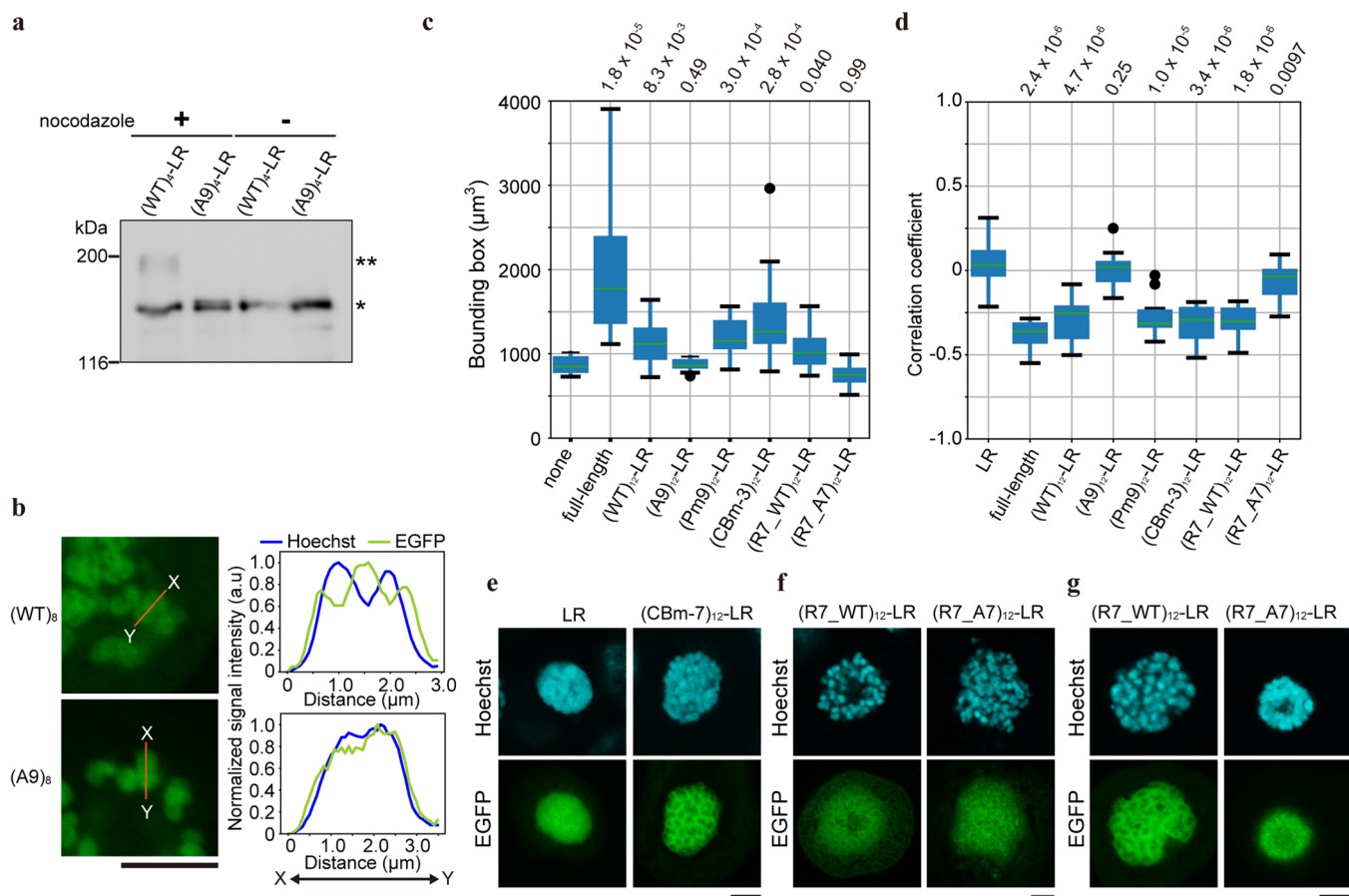

**Extended Data Fig. 7 | Mitotic phosphorylation, phosphomimetic mutants, and a charge-block mimetic mutant of Ki-67 R12 induce the formation of the mitotic chromosome periphery. a**, Mitotic phosphorylation of Ki-67 is abolished by alanine mutations. WT and A9 mutant of R12 (EGFP-(WT)$_4$-LR and EGFP-(A9)$_4$-LR) were expressed in HeLa cells. Nocodazole-treated and non-treated cells were harvested. Whole cell lysates were separated by SDS-PAGE, transferred to a polyvinylidene fluoride membrane (GE Healthcare), and probed with anti-GFP antibody (rabbit polyclonal, MBL, Code No. 598) (x4,000 dilution), followed by HRP-linked anti-rabbit IgG (GE Healthcare, NA-943) (x5,000 dilution). Immunoreactive signal was detected by chemiluminescence reaction and imaged using a LAS-3000 mini (Fuji Film). * and ** represent dephosphorylated and phosphorylated forms, respectively. **b**, Fluorescence images of mitotic chromosomes of HeLa cells expressing EGFP-(WT)$_8$-LR and EGFP-(A9)$_8$-LR. Section analyses of EGFP and DNA signal on mitotic chromosomes are shown. Bar: 5 μm. **c**, Dispersion of mitotic chromosomes. A 3D image of mitotic chromosome was reconstituted from a series of z-stack fluorescence images (Δz = 0.5 or 0.35 μm) of Hoechst33342 and analysed using "3D Object Counter" in Fiji[47] to obtain a bounding box of mitotic chromosomes. The volume of the bounding box was plotted for the cells expressing indicated Ki-67 constructs. Boxplot shows the minimum and maximum value by whiskers, 25% and 75% percentile by box boundaries, median by bar inside the box and outlier by dot. Numbers above graph indicate *p*-values (comparison with none) by one-tailed Mann–Whitney U test (n = 13, 11 and 12 cells for EGFP-(CBm-3)$_{12}$-LR, EGFP-(R7_WT)$_{12}$-LR and the others, respectively). **d**, Localization of Ki-67 at the mitotic chromosome periphery was quantified based on the coefficient of correlation between Hoechst and EGFP signal in mitotic cells expressing EGFP-fused Ki-67. Pixel of Interest (POI) was defined as the pixel that had a signal above the threshold value (α) of either EGFP or Hoechst, where α was defined by the equation described in Supplementary Note. The Pearson correlation coefficient between Hoechst and EGFP signal in the POI was calculated. Boxplot shows the minimum and maximum value by whiskers, 25% and 75% percentile by box boundaries, median by bar inside the box and outlier by dot. Numbers above graph indicate *p*-values (comparison with LR) by one-tailed Mann–Whitney U test (n = 11, 12 and 10 cells for EGFP-(R7_WT)$_{12}$-LR, EGFP-(R7_A7)$_{12}$-LR and the others, respectively). **e**, Intracellular localization of EGFP-LR and EGFP-(CBm-7)$_{12}$-LR in Ki-67-KO cell. DNA was stained with Hoechst33342. Bar, 5 μm. **f, g**, Intracellular localization of homogeneous repeat of R7 (EGFP-(R7_WT)$_{12}$-LR and EGFP-(R7_A7)$_{12}$-LR) in mitotic HeLa cells (**f**) and Ki-67-KO cells (**g**). DNA was stained with Hoechst33342. Bar, 5 μm. Source numerical data and an unprocessed blot are available in Source Data.

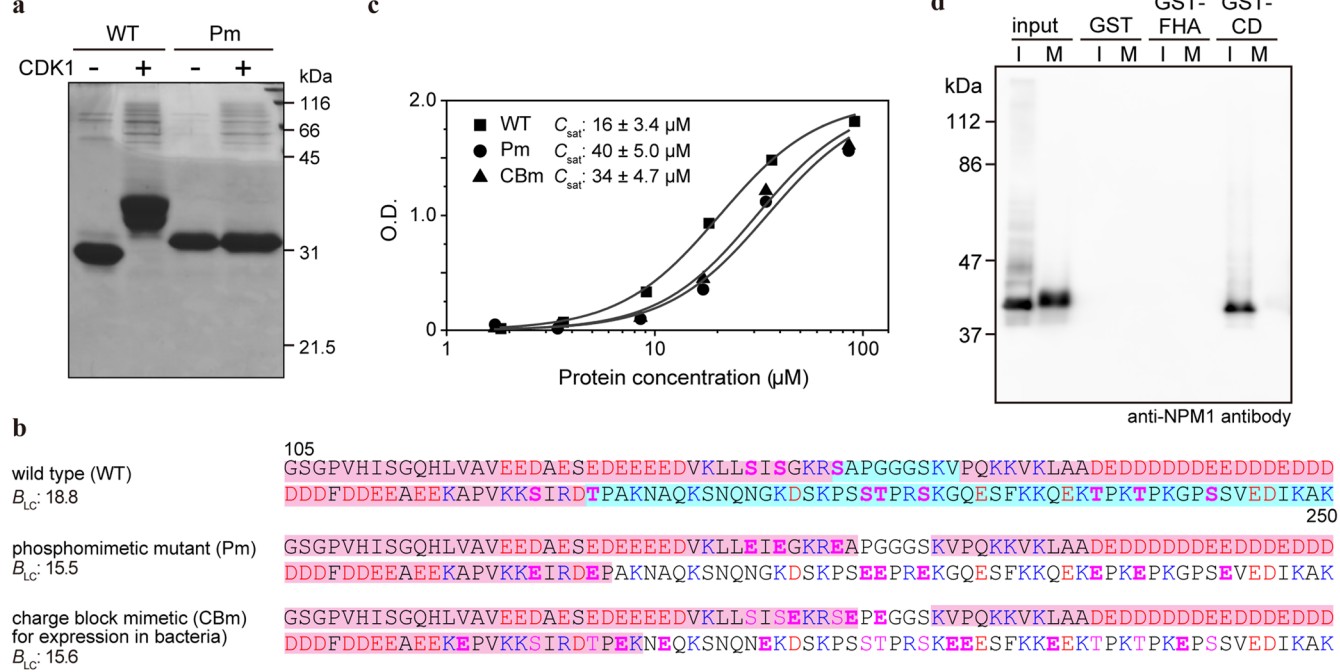

**a**

WT Pm
CDK1 − + − +  kDa
116
66
45
31
21.5

**b**

wild type (WT)
$B_{LC}$: 18.8

```
105
GSGPVHISGQHLVAVEEDAESEDEEEEDVKLLSISGKRSAPGGGSKVPQKKVKLAADEDDDDDDEEDDDEDDD
DDDFDDEEAEEKAPVKKSIRDTPAKNAQKSNQNGKDSKPSSTPRSKGQESFKKQEKTPKTPKGPSSVEDIKAK
                                                                   250
```

phosphomimetic mutant (Pm)
$B_{LC}$: 15.5

```
GSGPVHISGQHLVAVEEDAESEDEEEEDVKLLEIEGKREAPGGGSKVPQKKVKLAADEDDDDDDEEDDDEDDD
DDDFDDEEAEEKAPVKKEIRDEPAKNAQKSNQNGKDSKPSEEPREKGQESFKKQEKEPKEPKGPSEVEDIKAK
```

charge block mimetic (CBm)
for expression in bacteria)
$B_{LC}$: 15.6

```
GSGPVHISGQHLVAVEEDAESEDEEEEDVKLLSISEKRSEPEGGSKVPQKKVKLAADEDDDDDDEEDDDEDDD
DDDFDDEEAEEKEPVKKSIRDTPEKNEQKSNQNEKDSKPSSTPRSKEEESFKKEEKTPKTPKEPSSVEDIKAK
```

charge block mimetic (CBm)
for expression in HeLa)
$B_{LC}$: 15.6

```
GSGPVHISGQHLVAVEEDAESEDEEEEDVKLLAIAEKRAEPEGGSKVPQKKVKLAADEDDDDDDEEDDDEDDD
DDDFDDEEAEEKEPVKKAIRDAPEKNEQKSNQNEKDSKPSAAPRAKEEESFKKEEKAPKAPKEPASVEDIKAK
```

**c**

O.D.
2.0
1.0
0

■ WT  $C_{sat}$: 16 ± 3.4 µM
● Pm  $C_{sat}$: 40 ± 5.0 µM
▲ CBm  $C_{sat}$: 34 ± 4.7 µM

Protein concentration (µM)
1   10   100

**d**

input | GST | GST-FHA | GST-CD
I M | I M | I M | I M
kDa
112
86
47
37

anti-NPM1 antibody

**e**

|  | interphase | pro~metaphase | anaphase | telophase |
|---|---|---|---|---|
| localization | | | | |
| molecular orientation | | | | |

Ki67
N ——— LR— C
RD
NPM1
DNA

charge blocks
NPM1
Ki67

LLPS
NPM1
Ki67

**Extended Data Fig. 8 | See next page for caption.**

**Extended Data Fig. 8 | Mitotic hyper-phosphorylation reduces charge blockiness and the propensity for LLPS of NPM1. a**, *In vitro* phosphorylation of NPM1 IDR by CDK1/cyclin B. Purified NPM1-IDR (wild-type (WT) and phosphomimetic mutant (Pm)) was incubated with CDK1/cyclin B at room temperature for 60 min. Proteins were analysed by PhosTag gel electrophoresis and visualised by CBB staining. **b**, Amino-acid sequences of human NPM1 IDR (amino acids 105–250) and its mutants. Positively and negatively charged residues are labelled in blue and red, respectively. Mitotic phosphosites are labelled in bold magenta. Positive and negative charge blocks (see Methods (window size: 31 amino acids)) are highlighted in cyan and red, respectively. In the phosphomimetic mutant (Pm), 11 mitotic phosphosites were replaced with glutamic acid. In the charge-block mimetic mutant (CBm), non-charged residues, except mitotic phosphosites, were replaced with glutamic acid to mimic the charge blockiness of the phosphomimetic mutant. To express CBms in mammalian cells, 11 mitotic phosphosites were additionally replaced with alanine to avoid mitotic phosphorylation. The $B_{LC}$ values are also presented. **c**, Turbidity assay using WT, Pm, and CBm of NPM1 IDR. One representative result is shown (out of three). $C_{sat}$ values obtained from three independent measurements are presented as mean ± standard deviation. **d**, Interaction between the N-terminal domain of Ki-67 and NPM1. GST-fused FHA (amino acids 26–97) and conserved (amino acids 284–484, CD) domains of human Ki-67 were purified from bacteria and immobilized on glutathione-coupled beads. Interphase (I) and mitotic (M) HeLa cell lysates were prepared and incubated with the protein-bound beads. The bound protein was eluted by 50 mM glutathione and analysed by SDS-PAGE and subsequent immunoblotting using anti-NPM1 antibody (Invitrogen, FC-61991) (x2,000 dilution) and HRP-linked anti-mouse IgG (GE Healthcare, NA-931) (x6,000 dilution). **e**, Model of mitotic behaviour of Ki-67 and NPM1. Green: Ki-67, red: NPM1, blue: DNA. Fluorescence microscopic images of HeLa cells expressing EGFP-Ki-67 and mCherry-NPM1 are also shown. Bar, 2 µm. Ki-67 localises to the perinucleolar chromosome as it binds both to NPM1 (via its N-terminus) and chromosomes (via its C-terminal LR). Due to these spatial and orientational constrictions, the RD may be compacted via weak interactions among RDs (self-coacervation). Ki-67 and NPM1 are hyper-phosphorylated when the cell enters mitosis, and the nucleoli are disassembled because of a reduction in the propensity for LLPS of phosphorylated NPM1 (and other nucleolar proteins as well). In contrast, phosphorylated Ki-67 undergoes LLPS and forms a liquid-like phase at the chromosome surface while it is anchored to the chromosome surface via LR. In this 'free-end' state, the RD prefers to be more extended because of strong intermolecular interactions (Fig. 3c) and a high graft density on the chromosome surface resulting from mitotic chromosome condensation[37,48]. In anaphase, when NPM1 and Ki-67 are dephosphorylated and begin interacting with each other again, the chromosome periphery starts to disintegrate because of the reduced LLPS of Ki-67. Dephosphorylated NPM1 reassembles with Ki-67 and forms small liquid-like droplets in inter-chromosomal spaces, which eventually fuse with each other to form nucleoli in the early G1 phase as the chromosomes decondense. Source numerical data, an unprocessed gel and blot are available in Source Data.

Charge blockiness is enhanced by mitotic phosphorylation

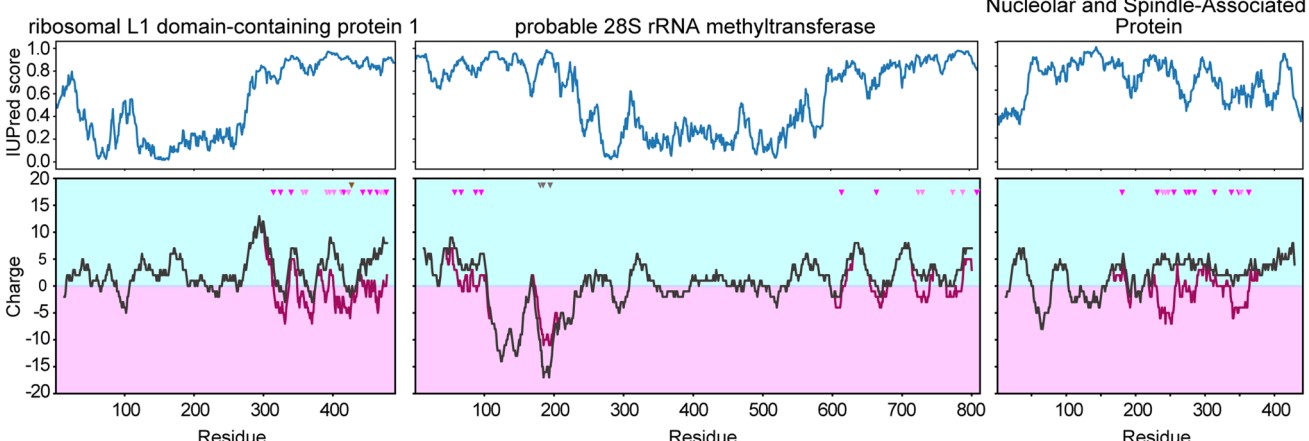

Charge blockiness is reduced by mitotic phosphorylation

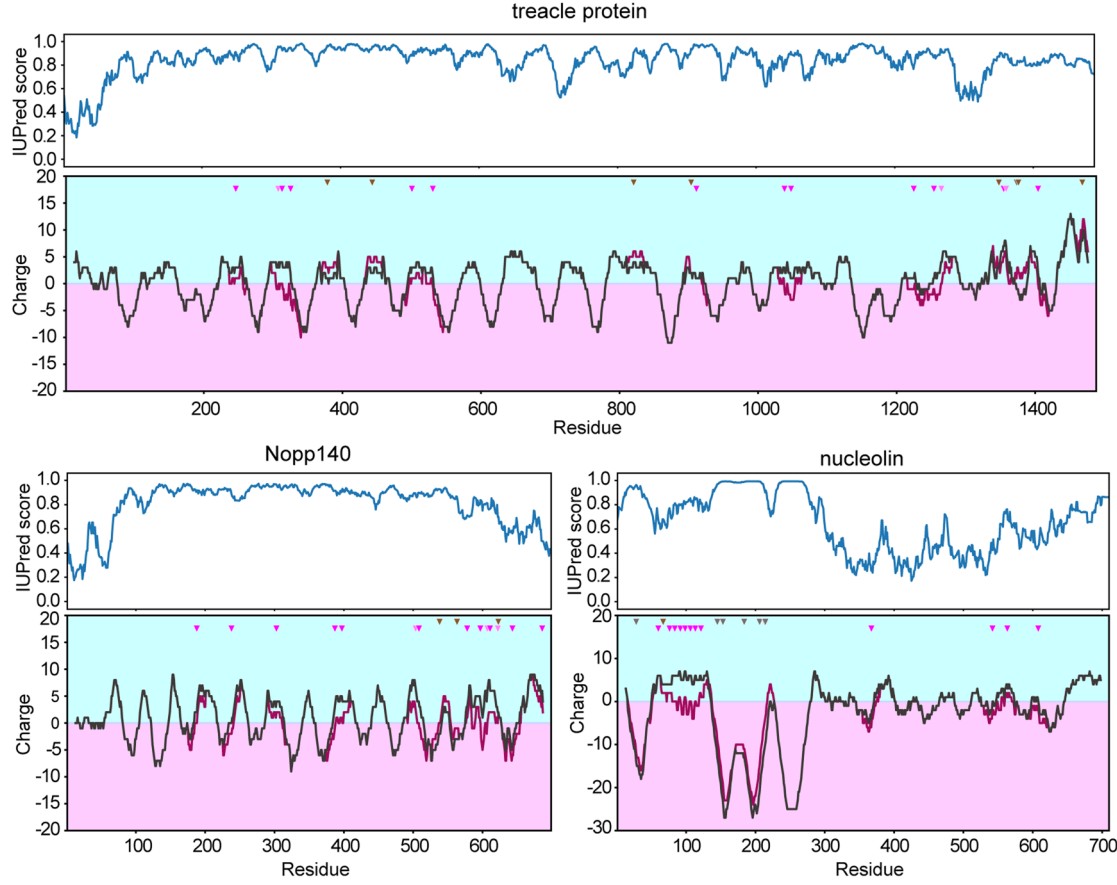

**Extended Data Fig. 9 | Nucleolar proteins that undergo mitotic hyperphosphorylation in IDRs.** The disorder score predicted by IUPred2A[49,50] and the charge plots (window size: 25 amino acids) of interphase (grey) and mitotic (purple) forms are shown. Phosphosites that are increased (magenta and light magenta) and decreased (brown and grey) upon entry into mitosis are indicated by triangles. Magenta and brown phosphosites were identified as a single site in trypsin peptide, whereas light magenta and grey sites were identified with nearby sites in the same peptide. Mitotic phosphorylation enhances the alternating charge blocks in ribosomal L1 domain-containing protein 1, probable 28 S rRNA methyltransferase, and nucleolar and spindle-associated protein (NuSAP), whereas it diminishes in treacle protein, nucleolin, and Nopp140. The sequences were obtained from UniProt database (UniProt accession numbers: ribosomal L1 domain-containing protein 1; O76021, probable 28 S rRNA methyltransferase; P46087, NuSAP; Q9BXS6, treacle protein; Q13428, nucleolin; P19338, Nopp140; Q14978).

# Reporting Summary

## Statistics

For all statistical analyses, confirm that the following items are present in the figure legend, table legend, main text, or Methods section.

| n/a | Confirmed | |
|---|---|---|
| ☐ | ☒ | The exact sample size ($n$) for each experimental group/condition, given as a discrete number and unit of measurement |
| ☐ | ☒ | A statement on whether measurements were taken from distinct samples or whether the same sample was measured repeatedly |
| ☐ | ☒ | The statistical test(s) used AND whether they are one- or two-sided<br>*Only common tests should be described solely by name; describe more complex techniques in the Methods section.* |
| ☒ | ☐ | A description of all covariates tested |
| ☒ | ☐ | A description of any assumptions or corrections, such as tests of normality and adjustment for multiple comparisons |
| ☐ | ☒ | A full description of the statistical parameters including central tendency (e.g. means) or other basic estimates (e.g. regression coefficient) AND variation (e.g. standard deviation) or associated estimates of uncertainty (e.g. confidence intervals) |
| ☐ | ☒ | For null hypothesis testing, the test statistic (e.g. $F$, $t$, $r$) with confidence intervals, effect sizes, degrees of freedom and $P$ value noted<br>*Give P values as exact values whenever suitable.* |
| ☒ | ☐ | For Bayesian analysis, information on the choice of priors and Markov chain Monte Carlo settings |
| ☒ | ☐ | For hierarchical and complex designs, identification of the appropriate level for tests and full reporting of outcomes |
| ☐ | ☒ | Estimates of effect sizes (e.g. Cohen's $d$, Pearson's $r$), indicating how they were calculated |

*Our web collection on statistics for biologists contains articles on many of the points above.*

## Software and code

Policy information about availability of computer code

| Data collection | Fluorescence images were obtained by a confocal laser scanning microscope from Olympus (FV-3000) with an accompanying software, FV-31S (2.3.1.163).<br>Optical density was measured by spectrophotometer from JASCO (V-630) with an accompanying software (Spectra Manager 2.10.1).<br>Circular dichroism spectra were measured by a spectropolarimeter from JASCO (J-805) with an accompanying software (Spectra Manager 1.55.0) |
|---|---|
| Data analysis | Python 3.7.6, Numpy 1.18.1, Pandas 1.0.1, Matplotlib 3.1.3, Scipy 1.4.1, opencv-python 4.4.0, seaborn 0.10.0, Fiji 2.1.0, MetaMorph 7.8.0.0, OriginPro 9.8.0.200, ClustalOmega 1.2.4 |

For manuscripts utilizing custom algorithms or software that are central to the research but not yet described in published literature, software must be made available to editors and reviewers. We strongly encourage code deposition in a community repository (e.g. GitHub). See the Nature Portfolio guidelines for submitting code & software for further information.

## Data

Policy information about availability of data

All manuscripts must include a data availability statement. This statement should provide the following information, where applicable:
- Accession codes, unique identifiers, or web links for publicly available datasets
- A description of any restrictions on data availability
- For clinical datasets or third party data, please ensure that the statement adheres to our policy

The data that support the findings of this study are available in Source Data file and Supporting Information. All other data supporting the findings of this study are available from the corresponding author on reasonable request.

Amino acid sequences of human Ki-67 and human NPM1 can be obtained from UniProt database (accession number P46013 for Ki-67 and P06748 for NPM1)

# Field-specific reporting

Please select the one below that is the best fit for your research. If you are not sure, read the appropriate sections before making your selection.

☒ Life sciences    ☐ Behavioural & social sciences    ☐ Ecological, evolutionary & environmental sciences

For a reference copy of the document with all sections, see nature.com/documents/nr-reporting-summary-flat.pdf

# Life sciences study design

All studies must disclose on these points even when the disclosure is negative.

| | |
|---|---|
| Sample size | No statistical methods were used to predetermine sample size. Sample sizes were estimated empirically on the basis of pilot experiments and previously performed experiments with similar setup to provide sufficient sample sizes for statistical analysis. |
| Data exclusions | No data was removed from the analysis with the exception of the image analysis of droplet. For the quantification of droplet, ones on the edge of images and ones with higher eccentricity than criterion (0.7) (to exclude the aggregation) were excluded. |
| Replication | Experiments were repeated multiple times. Turbidity assay has been performed 3 times, and microscopic observation, gel electrophoresis, western blotting and EMSA have been performed at least twice to check data reproducibility. |
| Randomization | For droplet assay, sample and measurement order was randomized. The area of microscopic observation was randomly determined. |
| Blinding | Experiments performed in this article were not blinded and this is consistent with what is published in the field. |

# Reporting for specific materials, systems and methods

We require information from authors about some types of materials, experimental systems and methods used in many studies. Here, indicate whether each material, system or method listed is relevant to your study. If you are not sure if a list item applies to your research, read the appropriate section before selecting a response.

## Materials & experimental systems

| n/a | Involved in the study |
|---|---|
| ☐ | ☒ Antibodies |
| ☐ | ☒ Eukaryotic cell lines |
| ☒ | ☐ Palaeontology and archaeology |
| ☒ | ☐ Animals and other organisms |
| ☒ | ☐ Human research participants |
| ☒ | ☐ Clinical data |
| ☒ | ☐ Dual use research of concern |

## Methods

| n/a | Involved in the study |
|---|---|
| ☒ | ☐ ChIP-seq |
| ☒ | ☐ Flow cytometry |
| ☒ | ☐ MRI-based neuroimaging |

## Antibodies

| | |
|---|---|
| Antibodies used | anti-NPM1 antibody (Invitrogen, FC-61991)<br>anti-GFP antibody (MBL, Code No.598)<br>HRP-linked anti-rabbit IgG (GE Healthcare, NA-943)<br>HRP-linked anti-mouse IgG (GE Healthcare, NA-931) |
| Validation | 1. anti-NPM1 antibody<br>According to the manufacturer's data, the anti-NPM1 antibody recognizes human, mouse and rat NPM1 in western blotting, Immunohistochemistry, immunoprecipitation, and ELISA. The specificity to human NPM1 in western blot was also confirmed by using the HeLa (human) cell lysate containing mCherry-tagged human NPM1. This antibody recognized both endogenous and exogenous human NPM1 in western blot analysis.<br><br>2. anti-GFP antibody (MBL, Code No. 598)<br>According to the manufacturers data, the polyclonal anti-GFP antibody can detect GFP from Aequorea Victria and its variants (EBFP, SEBFP, ECFP, SECFP, EGFP, SEGFP, cpSEGFP, EYFP, Venus, cpVenus, R-pericam, and Sapphire) on Western blotting, Immunoprecipitation, Immunocytochemistry and Immunohistochemistry. The specificity to EGFP was also confirmed by using the cell lysate from HeLa (human) cells expressing EGFP. |

# Eukaryotic cell lines

Policy information about cell lines

| | |
|---|---|
| Cell line source(s) | HeLa cells were purchased from ATCC. Knock-out line of Ki-67 in HCT116 was established in the previous study (Takagi, M. Biochemistry and Biophysics Reports 22, 100720 (2020) by one of the co-authors, M. Takagi (RIKEN, Japan). |
| Authentication | Knock-out of Ki-67 in HCT116 cells were authenticated by PCR and DNA sequencing. HeLa and HCT116 cells were authenticated by ATCC. |
| Mycoplasma contamination | All cell lines were tested negative for mycoplasma contamination. |
| Commonly misidentified lines (See ICLAC register) | No commonly misidentified lines were used in this study. |

