## [Peer Review File · Nature Cell Biology]

Peer Review Information

Journal: Nature Cell Biology

Manuscript Title: Cell cycle-specific phase separation regulated by protein charge blockiness

Corresponding author name(s): Shige Yoshimura

Reviewer Comments & Decisions:

Decision Letter, initial version:

Dear Professor Yoshimura,

Thank you for submitting your manuscript, "Cell cycle-specific phase separation regulated by protein charge blockiness" to Nature Cell Biology. It has now been seen by 3 referees, who are experts in Ki-67 (referee 1); LLPS/IDR (referee 2); and LLPS/IDR (referee 3). As you will see from their comments (attached below), they find this work of potential interest, but have raised substantial concerns, which in our view would need to be addressed with considerable revisions before we can consider publication in Nature Cell Biology.

As you may know, Nature Cell Biology editors discuss the referee reports in detail within the editorial team, including the chief editor, to identify key referee points that should be addressed with priority, and requests that are overruled as being beyond the scope of the current study. To guide the scope of the revisions, I have listed these points below. We are committed to providing a fair and constructive peer-review process, so please feel free to contact me if you would like to discuss any of the referee comments further. Our typical revision period is six months; however, please let me know if you anticipate any issues or delays addressing the reviews, we'd be happy to discuss resubmission to the journal and the revisions as needed.

In particular, for reconsideration at the journal, it would be essential to:

A) Provide definitive evidence resolving whether charge patterning or a change in the net charge is key to LLPS and associated cellular functions (Rev#3), providing controls for charge blockiness for both Ki-67 and B23 lines of investigations. Please also address Rev#3 and Rev#1's questions about kappa calculations and inconsistencies with experimental outcomes as well as the concern whether the R12 repeats can recapitulate the behavior of full-length Ki-67 (Rev#2 point #3).

B) We feel it would be important to strengthen the analyses of Ki-67 function in KO cells expressing WT or mutant proteins as per Rev#2 point #1 - providing an additional readout to assess Ki-67 KO

phenotype as suggested by the reviewer would help address the reviewer's concern on whether Ki-67 LLPS is essential for its cellular function.

C) Rev#2 pt #2: please strengthen the DNA-beads analyses, noting that further mechanistic investigation into how Ki-67 LLPS enables chromosomal periphery organization may be beyond the scope of the current manuscript.

D) All other referee concerns pertaining to strengthening existing data, providing controls, methodological details, clarifications and textual changes, should also be addressed. Please especially pay close attention to the technical issues raised the referees, including Rev#2's "technical issues" and Rev#3's worry that samples may be contaminated.

E) Finally please pay close attention to our guidelines on statistical and methodological reporting (listed below) as failure to do so may delay the reconsideration of the revised manuscript. In particular please provide:

We would be happy to consider a revised manuscript that would satisfactorily address these points, unless a similar paper is published elsewhere, or is accepted for publication in Nature Cell Biology in the meantime, as per journal policy.

In contrast, although we agree with Reviewer #2 that points #4 and #5 are valid, we consider these points to be beyond the scope of the present study. Thus, addressing them experimentally will not be necessary for reconsideration of the manuscript at this journal. Please adjust the model accordingly (point #4) and discuss point #5.

- ensure that it conforms to our format instructions and publication policies (see below and <https://www.nature.com/nature/for-authors>).

- provide a point-by-point rebuttal to the full referee reports verbatim, as provided at the end of this letter.

- provide the completed Reporting Summary (found here <https://www.nature.com/documents/nr->

2reporting-summary.pdf). This is essential for reconsideration of the manuscript will be available to editors and referees in the event of peer review. For more information see <http://www.nature.com/authors/policies/availability.html> or contact me.

When submitting the revised version of your manuscript, please pay close attention to our [Digital Image Integrity Guidelines](https://www.nature.com/nature-research/editorial-policies/image-integrity). and to the following points below:

Nature Cell Biology is committed to improving transparency in authorship. As part of our efforts in this direction, we are now requesting that all authors identified as 'corresponding author' on published papers create and link their Open Researcher and Contributor Identifier (ORCID) with their account on the Manuscript Tracking System (MTS), prior to acceptance. ORCID helps the scientific community achieve unambiguous attribution of all scholarly contributions. You can create and link your ORCID from the home page of the MTS by clicking on 'Modify my Springer Nature account'. For more information please visit www.springernature.com/orcid.

This journal strongly supports public availability of data. Please place the data used in your paper into a public data repository, or alternatively, present the data as Supplementary Information. If data can only be shared on request, please explain why in your Data Availability Statement, and also in the correspondence with your editor. Please note that for some data types, deposition in a public repository is mandatory - more information on our data deposition policies and available repositories appears below.

[REDACTED]

We hope that you will find our referees' comments and editorial guidance helpful. Please do not

hesitate to contact me if there is anything you would like to discuss. Thank you again for considering NCB for your work.

Best wishes,

Melina

Melina Casadio, PhD
Senior Editor, Nature Cell Biology
ORCID ID: <https://orcid.org/0000-0003-2389-2243>

Reviewers' Comments:

Reviewer #1:

Remarks to the Author:

The authors present detailed analysis of prominent mitotic protein phosphorylations that regulate cellular localization and liquid-like protein interactions. One set of experiments focuses on the internal repeats within the Ki-67 protein. Ki-67 has gained much interest recently as the foundation of the perichromosomal layer that coats mitotic chromosomes, and because Ki-67 ensures individualization of chromosome arms during mitosis.

Here, the authors show that the mitotic hyperphosphorylation of the Ki-67 internal repeats generates an alternating pattern of "charge blockiness", in which there are alternating regions of high positive charge (due to basic residues), followed by regions of high negative charge (resulting from mitotic phosphorylations). The definition of the minimal region of the protein necessary and sufficient for forming the PCL and individualizing chromosomes (Fig. 3), and the demonstration that these principles work in reverse for the B23 protein (Fig. 4) make this a particularly impactful study that will be of interest to all cell biologists, especially given the current level of interest in LLPS. I recommend publication once the minor issues I describe are addressed.

Minor points:

A particularly compelling set of experiments is the biochemical reconstitution of protein-protein interactions among the Ki-67 repeats (e.g. Fig. 2c). Proper interpretation of this experiment requires Ext. Fig. 2e, showing that the Pm9 domain lacking LR requires unlabeled, LR-anchored protein to bring it to the surface of the bead (otherwise a reader might wonder if phase separation by the repeats was affecting DNA binding by the LR domain). Would it be possible to move Extended Fig. 2e into the main Figure 2?

It's not fully clear to me how the "blocks" are defined for each Ki-67 repeat. Specifically, are the 2nd, 3rd and 4th "E" mutations in the R12-Pm9 mutant within the first block or the second block? The same question applies to the R7 repeat in Ext Fig. 2b. The block boundaries should be indicated.

4Figure 1b- where is the charge analysis and LLPS assay of the A9 mutant? Also, I was surprised to see that the value of "K" measuring "charge blockiness" is higher for the Pm6 mutant than for wt. This makes me wonder how frequently the mathematical calculation for this value deviates from the expectation one has based on visual inspection of the charge differences. For example, based on the charge distribution graphs and K values, one would predict that the CBm-4 mutant would have the strongest effect in the LLPS assay, but it doesn't appear so. Do the authors have a hypothesis about this? Also, it would be helpful to show the values of K for each of the repeats analyzed in Extended Figure 1a.

Fig. 1d- these are GFP-tagged, right? That should be indicated in the schematic.

Figure 1e- It would be more accurate to label the panels as WT-(R12)1 than as WT.

Extended Fig. 1- the graphs of the mitotic vs interphase charge in panels a and b would be easier to analyze if they were graphed in more visually distinct color and thickness.

Extended Fig. 1e- is this the correct Figure and Legend? What is shown looks like fluorescence intensity analysis of R12 multimers of different lengths, but the legend described this as quantitation of the kinetic mixing experiment in panel Extended 1d.

Extended Fig. 2a- Why are values > 104 dark-shaded here and not in the other measurements of drop intensity (Ext. Figs. 1c and 1e)? And what are the values shown in these panels?

Extended Fig. 3d- add the label "LR-free constructs"

Figure 4c- The meaning of the arrowheads and asterisks should also be in the Legend.

Extended Fig. 4c- label with "anti-B23 antibody detection"

Extended Fig. 5d. Next to the cartoons of Ki-67 and B23 at the bottom, there should be an indication that blue = DNA.

Dr. Paul Kaufman
U. Massachusetts Medical School

Reviewer #2:

Remarks to the Author:

The manuscript by Yamazaki et al. describes that alternating charge blocks of Ki-67 and B23 tuned by phosphorylation regulate their liquid-liquid phase separation (LLPS) involved in chromosome periphery formation and nucleoli dissolution during cell-cycle. The authors reported that mitotic hyper-phosphorylation of multiple residues throughout the entire repeat domain (RD) of Ki-67 generates multivalent charge blocks which promotes LLPS of Ki-67 LLPS and is essential for Ki-67-associated mitotic chromosome periphery formation in cells. In addition, the authors presented interesting pieces

5of data that mitotic phosphorylation reduces charge blocks of B23, thereby inhibiting its LLPS and assisting the disassembly of nucleolus in mitosis.

Overall, this work describes a couple of interesting observations, and comes up with two major findings including (1) new mechanism for regulation protein LLPS (charge blockiness by phosphorylation); (2) discovery of Ki-67 and B23 LLPS in mediating their physiological function in cell cycle. However, several previously studies have reported similar phenomena and LLPS mechanisms, which great reduce the novelty of this work. For instance, Zhou et al. (Angew Chem Int 2019) reported that charge blockiness is important in mediating protein LLPS. Cuylen et al. (Nature, 2016) suggested that Ki-67 forms a steric and electrostatic charge barrier, similar to surface-active agents (surfactants) that disperse particles or phase-separated liquid droplets in solvents. Booth et al. (eLife, 2014) reported the function of Ki-67 in organizing the mitotic chromosome periphery. This work claims that Ki-67 LLPS is important in mediating its function. However, the data only supports in part that Ki-67 can undergo LLPS in vitro and in cell involved in periphery organization. Whether Ki-67 LLPS is essential for the chromosome periphery organization and why LLPS of Ki-67 is capable of inducing periphery organization remain unanswered. Moreover, the overall quality of data is relatively low. Some were overinterpreted and cannot be used to support the author's claims. Please see my detailed concerns below.

Major concerns :

1. Despite that Ki-67 can undergo LLPS in vitro and in cell, the authors didn't provide convincing evidence to support Ki-67 LLPS is essential for its cellular function. Whether knock-in of Ki-67 variants (LLPS mimetic or LLPS deficient) can rescue Ki-67 knockout (KO)-phenotype can provide some clues on this. However, the data (Fig.3 and Fig.S3) is not convincing. The readout of the "KO-phenotype" is too simply (only calculate the convex hull area of mitotic chromosomes). Moreover, the difference of convex hull area between different variants are not obvious, and the variation within the same sample is huge (Fig. S3i). The authors need additional quantitative assay to study this.

2. The author studied the LLPS behavior of Ki-67 WT and its variants mainly in aqueous solution. However, in vivo, Ki-67 is attached on chromosome surface. The Ki-67 LLPS properties drawn from solution study cannot be applied to that on the chromosome surface. Based on the data in this study, it's hard to know why Ki-67 LLPS is essential in mediating periphery organization. The authors attempted to use DNA decorated beads to mimic an "artificial chromosome surface". Despite that Ki-67 variants can form a ring-like structure on the bead (Fig.2C), it CAN'T support that Ki-67 undergoes LLPS on the surface (as the authors claimed, Page7, line 125). This data can only tell Ki-67 variants can attach on the bead. The authors need to establish a proper model to firstly directly observe that Ki-67 WT indeed undergoes LLPS on the DNA surface, and then examine the behavior of different variants.

3. To investigate LLPS mechanism of Ki-67 in vitro and in cell, the authors mainly focused on R12, the first repeat domain of Ki-67. (R12)_n and its variants were used in the in vivo assay. However, Ki-67 contains 16 repeats. R1-R16 display different charge patterns and their changes of charge patterns in mitosis also show significant difference as shown in Figure S1. It's questionable whether the R12 tandem repeats can fully recapitulate the behavior of Ki-67 full-length (FL) in the in vitro or cellular assay (Figure 2 and Figure 3). The authors need to prepare Ki-67 constructs with different RDs (e.g.

6R1-R7 and R8-R16) and examine them in the in vitro and cell model.

4. The authors showed a working model of interplay between Ki-67 and B23 (Fig.S4) without experimental evidences. The authors may need to explore their potential co-localization in different steps of cell cycle in their model.

5. Previous article (Sara Cuylen-Haering et al. Nature 2020) showed that Ki-67 exhibits extended conformation at the chromosome periphery in the metaphase, but collapse during mitotic exit. How does conformational change between extended and collapse states influence Ki-67 LLPS behavior?

Technical issues:

1. Protein LLPS conditions (e.g. protein concentration) are not indicated. Does Ki-67 undergo LLPS in vitro at the concentration similar to its physiological in vivo concentration?

2. In fluorescence-labeling assay, what's the molar ratio of the protein and fluorescence dyes? It's unknown whether the authors removed the free fluorescent molecules (from spontaneous hydrolysis of maleimide or the dye reacted with DTT) existing in labeled-protein solution. Does the free dye affect the fluorescent signal in Figure 2c? The authors need to ensure that (WT)²-LR and its variants in Figure 2c have same fluorescence-labeling efficiency before analyzing their fluorescent signal around the beads?

3. In Figure 2b, the phase separation of WT and Pm1 seems like irregular aggregates and the droplets of Pm6 and Pm9 contain many holes. Did those pictures represent the general morphology of proteins under this condition? What's more, the author did not mention the incubated time before the microscopic observation in method. Does the long incubation or surface material properties of 96-well plate induce the formation of irregular aggregates?

4. To quantify the phase separation ability of different proteins, fluorescent intensity of individual droplet in the images was quantified and summarized in the histogram. However, the droplet surface properties of different proteins and the stack of different collected pictures may affect fluorescent signal and is hard to be controlled. The macroscopical measurement (turbidity like OD_{600nm} or OD_{350nm}) (centrifugation for detecting components of droplets) or other assay may need to be supplied to verify their LLPS ability.

5. The authors assessed the charge blockiness along the polypeptide by using k value and claim the k is positively correlated with protein LLPS. However, in Figure 2b, CBm-4 has highest k value (8.1 *10⁻³), but its droplets are smaller than Pm6 (k value is 1.3 *10⁻³).

Minor issues:

1. The author didn't show localization of homogeneous repeat construct of Ki-67 in interphase in Figure 4d, but they mentioned this in Page13, 204-206.

2. A control of CDK1+ATP should be added in Figure 1e.

3. Some of the images are of poor quality, as shown in Fig. 1d and 1e.

Reviewer #3:

Remarks to the Author:

7The authors present an extremely clear and well-written study examining the effect of charge blockiness of cellular localization and chromosomal dynamics. They focus on two proteins and provide evidence that supports charge patterning as a possible determinant of function, recruitment, and interaction. I thoroughly enjoyed the paper and found it easy to follow, compelling and interesting.

The insights are novel, although there are quite a few citations to key previous papers that are missing (raised below). I have several suggestions below, and while there are a number of absolutely essential components that need to be addressed, but my biggest concern is that the authors do not - in my opinion - fully delineate between a model in which charge patterning matters *in vivo* vs. simply a change in the net charge. This absolutely can be addressed with a variant that distributes the charged residues uniformly while holding the other residues fixed, and/or by defining a minimally perturbative variant that breaks the charge blocks and showing there is a loss of function *in vivo*. I want to be clear that I actually do think the authors are right, but, I worry a critic may look at this and conclude that they have shown phosphorylation is essential (a well-established result) and move on.

My specific concerns are outlined below. I consider everything I have raised as requiring attention, but I do want to draw attention to my concern about *in vitro* purify of samples re: RNA contamination. I don't think this concern alters the authors' conclusions at all, but, for the sake of maintaining rigorous and high-caliber science is essential to address.

Major comments

The authors claim that the repeat number showing a similar tendency to phase separate demonstrates that repeat number rather than amino acid sequence determines phase separation. This sets up a potentially false dichotomy - both the repeat number and specific amino acid sequence could matter, or repeat number and amino acid composition could be the key determinant. However, as it stands, the results do not show that "the repeat number (multivalency), rather than the amino-acid sequence of individual repeats, is important for LLPS". There are a couple of ways to address this. At a minimum, the dependence of the saturation on repeat number should be shown and qualitatively (or ideally quantitatively) compared to expectation from polymer theory. Additionally, IF the authors wish to propose that a specific amino acid sequence is unimportant they must show that a shuffle of R12 (s-R12) shows the same length-dependence as the number of repeats is altered. Given the authors later show that charge patterning matters, it seems unlikely to me that a random shuffle would recapitulate wildtype behavior, invalidating the authors' statement, so my suggestion would simply be to remove that implication that amino acid sequence is not important.

I do not understand how the authors are calculating kappa, but it seems to be incorrect - this can be verified by passing the sequence into the CIDER webserver (<http://pappulab.wustl.edu/CIDER/analysis/>). The red flag for me is that a value of 0.0013 is incredibly small. Based on the correct definition of kappa, the wildtype R12 sequence has a kappa of 0.264, while the phosphorylated version has a value of 0.211 (see figure below). This is somewhat problematic in that the phosphomimetic variant (which phase separates more strongly than wildtype) actually has a lower (less blocky) sequence than the wildtype sequence. This sort of undermines the authors' model - in fact their data sort of shows the opposite, at least for WT and the Pm9 sequence. I didn't have the energy to manually type out the other sequences, but I think if the authors are going

8to argue that charge patterning influences phase separation an essential figure needs to show the saturation concentration vs. κ for the set of permutations described in Fig. 2 (and the Csat vs. net charge as a supplementary figure to confirm this is not the determinant of assembly). Without a clear quantification of this trend it's impossible for me to really evaluate if this model is correct or not (although to be clear, it could well be and it does make sense, so, don't take these comments as skepticism with respect to the model!). Again though, the key issue is that fluorescence images do not provide any means to really quantify the driving forces for phase separation.

Are the authors CERTAIN their sample is pure and free of RNA? The presence of vacuoles in Fig. 2 is something I've only ever seen when multi-component systems undergo maturation, notably when RNA is pulled down during a purification. I'd have to imagine these proteins can bind RNA robustly (regardless of if that is physiologically relevant). We have first-hand experience of this exact issue causing problems, and always purify with a high-salt wash and even RNase treatment. I would encourage the authors to identify a protocol develop for purifying RNA binding proteins without RNA and consider the steps taken there to assess if they are truly getting pure sample. The 260/280 nm absorbance ratio is also a good diagnostic check. I am not trying to be overly critical or cautious but we have been burned by this EXACT issue before; it is not easy!!!

Ammonium acetate seems like a pretty extreme chemical perturbant - how can the authors delineate the impact of ammonium acetate on KI-67 vs. it's global impact on cellular health and physiology? This may be well established but I am unaware of the associated literature.

The phospho-dependent localization to the periphery is interesting, but, in my opinion without a control that shows a uniformly shuffled sequence with the same charge does NOT localize to the periphery I'm not sure it's possible to distinguish between a charge-block effect vs. a change in overall net charge effect. Given the (actually) modest changes in κ , upon phosphorylation, I'm sort of bias to think maybe this is a net charge effect and not charge patterning effect. Again, this can be confirmed using charge shuffle variant, or at least showing that CBm1 does not rescue function. Showing CBm1 does not rescue function would be a really interesting result given it has the same charge as CBm3.

Similar concerns raised re: κ and the interpretation of charge blockiness apply to the B23 results - i.e. we need true controls of charge blockiness

Missing references include

Importance of charge blockiness as determinant of cellular assembly

Nott, T. J., Petsalaki, E., Farber, P., Jervis, D., Fussner, E., Plochowietz, A., Craggs, T. D., Bazett-Jones, D. P., Pawson, T., Forman-Kay, J. D., & Baldwin, A. J. (2015). Phase transition of a disordered nuage protein generates environmentally responsive membraneless organelles. *Molecular Cell*, 57(5), 936–947. (SEE FIG 6A)

Pak, C. W., Kosno, M., Holehouse, A. S., Padrick, S. B., Mittal, A., Ali, R., Yunus, A. A., Liu, D. R., Pappu, R. V., & Rosen, M. K. (2016). Sequence determinants of intracellular phase separation by complex coacervation of a disordered protein. *Molecular Cell*, 63(1), 72–85. (SEE FIG 5)

Bishof, I., Dammer, E. B., Duong, D. M., Kunding, S., Gearing, M., Lah, J. J., Levey, A. I., & Seyfried, N. T. (2018). RNA-binding proteins with basic-acidic dipeptide (BAD) domains self-assemble and aggregate in Alzheimer's disease. *The Journal of Biological Chemistry*.

<https://doi.org/10.1074/jbc.RA118.001747>

FINANCIAL AND NON-FINANCIAL COMPETING INTERESTS – the authors must include one of three declarations: (1) that they have no financial and non-financial competing interests; (2) that they have

10financial and non-financial competing interests; or (3) that they decline to respond, after the Author Contributions section. This statement will be published with the article, and in cases where financial and non-financial competing interests are declared, these will be itemized in a web supplement to the article. For further details please see <https://www.nature.com/licenceforms/nrg/competing-interests.pdf>.

Methods should be written concisely, but should contain all elements necessary to allow interpretation and replication of the results. As a guideline, Methods sections typically do not exceed 3,000 words. The Methods should be divided into subsections listing reagents and techniques. When citing previous methods, accurate references should be provided and any alterations should be noted. Information must be provided about: antibody dilutions, company names, catalogue numbers and clone numbers for monoclonal antibodies; sequences of RNAi and cDNA probes/primers or company names and catalogue numbers if reagents are commercial; cell line names, sources and information on cell line identity and authentication. Animal studies and experiments involving human subjects must be reported in detail, identifying the committees approving the protocols. For studies involving human subjects/samples, a statement must be included confirming that informed consent was obtained. Statistical analyses and information on the reproducibility of experimental results should be provided in a section titled "Statistics and Reproducibility".

All Nature Cell Biology manuscripts submitted on or after March 21 2016 must include a Data availability statement as a separate section after Methods but before references, under the heading "Data Availability". For Springer Nature policies on data availability see <http://www.nature.com/authors/policies/availability.html>; for more information on this particular policy see <http://www.nature.com/authors/policies/data/data-availability-statements-data-citations.pdf>. The Data availability statement should include:

- Accession codes for primary datasets (generated during the study under consideration and designated as "primary accessions") and secondary datasets (published datasets reanalysed during the study under consideration, designated as "referenced accessions"). For primary accessions data should be made public to coincide with publication of the manuscript. A list of data types for which submission to community-endorsed public repositories is mandated (including sequence, structure,

11microarray, deep sequencing data) can be found here
<http://www.nature.com/authors/policies/availability.html#data>.

- Unique identifiers (accession codes, DOIs or other unique persistent identifier) and hyperlinks for datasets deposited in an approved repository, but for which data deposition is not mandated (see here for details <http://www.nature.com/sdata/data-policies/repositories>).
- At a minimum, please include a statement confirming that all relevant data are available from the authors, and/or are included with the manuscript (e.g. as source data or supplementary information), listing which data are included (e.g. by figure panels and data types) and mentioning any restrictions on availability.
- If a dataset has a Digital Object Identifier (DOI) as its unique identifier, we strongly encourage including this in the Reference list and citing the dataset in the Methods.

We recommend that you upload the step-by-step protocols used in this manuscript to the Protocol Exchange. More details can found at www.nature.com/protocolexchange/about.

All imaging data should be accompanied by scale bars, which should be defined in the legend. Cropped images of gels/blots are acceptable, but need to be accompanied by size markers, and to retain visible background signal within the linear range (i.e. should not be saturated). The boundaries of panels with low background have to be demarked with black lines. Splicing of panels should only be considered if unavoidable, and must be clearly marked on the figure, and noted in the legend with a statement on whether the samples were obtained and processed simultaneously. Quantitative comparisons between samples on different gels/blots are discouraged; if this is unavoidable, it should only be performed for samples derived from the same experiment with gels/blots were processed in parallel, which needs to be stated in the legend.

Figures should be provided at approximately the size that they are to be printed at (single column is 86 mm, double column is 170 mm) and should not exceed an A4 page (8.5 x 11"). Reduction to the scale that will be used on the page is not necessary, but multi-panel figures should be sized so that the whole figure can be reduced by the same amount at the smallest size at which essential details in each panel are visible. In the interest of our colour-blind readers we ask that you avoid using red and green for contrast in figures. Replacing red with magenta and green with turquoise are two possible colour-safe alternatives. Lines with widths of less than 1 point should be avoided. Sans serif typefaces,

12such as Helvetica (preferred) or Arial should be used. All text that forms part of a figure should be rewritable and removable.

SUPPLEMENTARY INFORMATION – Supplementary information is material directly relevant to the

13conclusion of a paper, but which cannot be included in the printed version in order to keep the manuscript concise and accessible to the general reader. Supplementary information is an integral part of a Nature Cell Biology publication and should be prepared and presented with as much care as the main display item, but it must not include non-essential data or text, which may be removed at the editor's discretion. All supplementary material is fully peer-reviewed and published online as part of the HTML version of the manuscript. Supplementary Figures and Supplementary Notes are appended at the end of the main PDF of the published manuscript.

The total number of Supplementary Figures (not including the "unprocessed scans" Supplementary Figure) should not exceed the number of main display items (figures and/or tables (see our Guide to Authors and March 2012 editorial <http://www.nature.com/ncb/authors/submit/index.html#suppinfo>; <http://www.nature.com/ncb/journal/v14/n3/index.html#ed>). No restrictions apply to Supplementary Tables or Videos, but we advise authors to be selective in including supplemental data.

GUIDELINES FOR EXPERIMENTAL AND STATISTICAL REPORTING

REPORTING REQUIREMENTS – We are trying to improve the quality of methods and statistics reporting in our papers. To that end, we are now asking authors to complete a reporting summary that collects information on experimental design and reagents. The Reporting Summary can be found here <https://www.nature.com/documents/nr-reporting-summary.pdf> If you would like to reference the guidance text as you complete the template, please access these flattened versions at <http://www.nature.com/authors/policies/availability.html>.

STATISTICS – Wherever statistics have been derived the legend needs to provide the n number (i.e. the sample size used to derive statistics) as a precise value (not a range), and define what this value

14represents. Error bars need to be defined in the legends (e.g. SD, SEM) together with a measure of centre (e.g. mean, median). Box plots need to be defined in terms of minima, maxima, centre, and percentiles. Ranges are more appropriate than standard errors for small data sets. Wherever statistical significance has been derived, precise p values need to be provided and the statistical test used needs to be stated in the legend. Statistics such as error bars must not be derived from $n < 3$. For sample sizes of $n < 5$ please plot the individual data points rather than providing bar graphs. Deriving statistics from technical replicate samples, rather than biological replicates is strongly discouraged. Wherever statistical significance has been derived, precise p values need to be provided and the statistical test stated in the legend.

Author Rebuttal to Initial comments

Comments from the editor

A) Provide definitive evidence resolving whether charge patterning or a change in the net charge is key to LLPS and associated cellular functions (Rev#3), providing controls for charge blockiness for both Ki-67 and B23 lines of investigations. Please also address Rev#3 and Rev#1's questions about kappa calculations and inconsistencies with experimental outcomes as well as the concern whether the R12 repeats can recapitulate the behavior of full-length Ki-67 (Rev#2 point #3).

Response: We highly appreciate these comments. All three reviewers were concerned about the relationship between the charge blockiness and LLPS in the original manuscript. To improve this, we

15performed turbidity assays to quantify LLPS with the saturation concentration (C_{sat}) (suggested by Rev #3) and re-evaluated charge blockiness with several parameters different from kappa. The details are described in the individual responses (Rev#1 point #3, Rev#2 points #9 and #10, and Rev#3 points #2, #5, and #6). Thanks to these comments and additional experiments, we could show more convincing evidence for the relationship between charge blockiness and LLPS.

In addition, we performed the same set of *in vitro* and *in vivo* experiments that we had performed for R12 using another repeat, R7, to demonstrate that R12 recapitulates the behaviour of RD. As described in the response to another comment (Rev#2 point #3), R7 behaved similarly to R12. Now we strongly believe that the revised manuscript presents more convincing experimental evidence regarding the charge blockiness of RD and LLPS. We once again wish to express our appreciation for this useful comment.

B) We feel it would be important to strengthen the analyses of Ki-67 function in KO cells expressing WT or mutant proteins as per Rev#2 point #1 - providing an additional readout to assess Ki-67 KO phenotype as suggested by the reviewer would help address the reviewer's concern on whether Ki-67 LLPS is essential for its cellular function.

Response: We also appreciate this comment. To improve the analysis of the phenotype rescue by Ki-67 mutants, we performed two additional morphological analyses on mitotic chromosomes, namely the bounding box, which indicates the dispersion of chromosomes in a cell, and correlation coefficient, which indicates the segregation of DNA and Ki-67 signals. As described in the response to another comment (Rev#2, point #1), both analyses could demonstrate a significant effect of R12 mutants on the morphology of the chromosomes and the formation of the chromosome periphery in mitotic cells. We strongly believe that the revised manuscript now demonstrates the importance of charge blockiness for the formation of a mitotic chromosome periphery *in vivo*.

C) Rev#2 pt #2: please strengthen the DNA-beads analyses, noting that further mechanistic investigation into how Ki-67 LLPS enables chromosomal periphery organization may be beyond the scope of the current manuscript.

Response: We thank the reviewer for raising this issue. Rev#1 also suggested the rearrangement of the figures pertaining to the bead assay. According to these comments, we performed additional

16experiments, including fluorescence recovery after photobleaching analysis, to demonstrate LLPS on the DNA bead surface. All of the results are arranged in new Fig. 3 of the revised manuscript. Please refer to the detailed response to the reviewer's comment (Rev#2 point #2).

D) All other referee concerns pertaining to strengthening existing data, providing controls, methodological details, clarifications and textual changes, should also be addressed. Please especially pay close attention to the technical issues raised the referees, including Rev#2's "technical issues" and Rev#3's worry that samples may be contaminated.

Response: We highly appreciate many of the comments from the reviewers. As described in our point-by-point responses, we have addressed all of these concerns in the revised manuscript. The purity of our protein preparation was examined using an RNA-specific probe, the results of which turned out to be negative for an unusual droplet shape. We thus strongly believe that their concerns have been addressed.

E) Finally please pay close attention to our guidelines on statistical and methodological reporting (listed below) as failure to do so may delay the reconsideration of the revised manuscript. In particular please provide:

Response: Thank you for your kind instruction. We prepared a Supplementary Table that includes all numerical source data in Excel format, as well as a Supplementary Figure that contains unprocessed gel/blot images in pdf format. These files have been uploaded together with the revised manuscript and figures.

We would be happy to consider a revised manuscript that would satisfactorily address these points, unless a similar paper is published elsewhere, or is accepted for publication in Nature Cell Biology in the meantime, as per journal policy.

In contrast, although we agree with Reviewer #2 that points #4 and #5 are valid, we consider these points to be beyond the scope of the present study. Thus, addressing them experimentally will not be necessary for reconsideration of the manuscript at this journal. Please adjust the model accordingly (point #4) and discuss point #5.

Comments from reviewers:

Reviewer #1:

Remarks to the Author:

The authors present detailed analysis of prominent mitotic protein phosphorylations that regulate cellular localization and liquid-like protein interactions. One set of experiments focuses on the internal repeats within the Ki-67 protein. Ki-67 has gained much interest recently as the foundation of the perichromosomal layer that coats mitotic chromosomes, and because Ki-67 ensures individualization of chromosome arms during mitosis.

Here, the authors show that the mitotic hyperphosphorylation of the Ki-67 internal repeats generates an alternating pattern of “charge blockiness”, in which there are alternating regions of high positive charge (due to basic residues), followed by regions of high negative charge (resulting from mitotic phosphorylations). The definition of the minimal region of the protein necessary and sufficient for forming the PCL and individualizing chromosomes (Fig. 3), and the demonstration that these principles work in reverse for the B23 protein (Fig. 4) make this a particularly impactful study that will be of interest

18to all cell biologists, especially given the current level of interest in LLPS. I recommend publication once the minor issues I describe are addressed.

Minor points:

1. A particularly compelling set of experiments is the biochemical reconstitution of protein-protein interactions among the Ki-67 repeats (e.g. Fig. 2c). Proper interpretation of this experiment requires Ext. Fig. 2e, showing that the Pm9 domain lacking LR requires unlabeled, LR-anchored protein to bring it to the surface of the bead (otherwise a reader might wonder if phase separation by the repeats was affecting DNA binding by the LR domain). Would it be possible to move Extended Fig. 2e into the main Figure 2?

Response: We thank the referee for this useful comment. We agree that Extended Fig. 2e is necessary for the interpretation of the bead assay shown in Fig. 2c. As Reviewer 2 also raised some issues regarding the bead assay, we performed additional experiments to demonstrate LLPS on the bead surfaces. The results are as follows:

- i) LR is required for binding to the DNA beads (Fig. 3a, b, and Extended Fig. 3c)
- ii) LR-free R12 assembles to the layer of LR-R12 on the bead via intermolecular interactions (Extended Fig. 3c)
- iii) The assembly on the bead is stronger for Pm and CBm-3 than for WT (Fig. 3c, d)
- iv) The protein layer on the beads has a liquid-like behaviour (Fig. 3e)

All of these results have been presented in new Fig. 3 and Extended Data Fig. 3 of the revised manuscript. We believe that LLPS of phosphorylated R12 on the DNA-coated beads is clearly demonstrated. We have accordingly revised the main text and figure legend (page 7, line 135 – 141 and page 9, line 156 – page 10, line 168).

2. It's not fully clear to me how the “blocks” are defined for each Ki-67 repeat. Specifically, are the 2nd, 3rd and 4th “E” mutations in the R12-Pm9 mutant within the first block or the second block? The same question applies to the R7 repeat in Ext Fig. 2b. The block boundaries should be indicated.

Response: According to this suggestion, we have defined ‘charge block’ according to the following procedure:

- i) Obtain the charge plot with a window size 35 (for Ki-67 RDs) or 31 (for B23)

19

- ii) Calculate the areas above and below zero
- iii) Define the ‘charge block’ as the area larger than 20

The positive and negative charge blocks are now indicated on the amino-acid sequences in Fig. 2a (R12) and Extended Data Fig. 2b (R7), 2f (CBMs), and 5b (B23). The description has also been added to the Methods section (page 19, line 316 – 317).

In addition to these modifications, we have improved the procedure to quantify ‘charge blockiness’ according to the reviewers’ suggestions (please refer to comment #3). We believe that the charge block is now clearly defined and quantified.

3. Figure 1b- where is the charge analysis and LLPS assay of the A9 mutant? Also, I was surprised to see that the value of “K” measuring “charge blockiness” is higher for the Pm6 mutant than for wt. This makes me wonder how frequently the mathematical calculation for this value deviates from the expectation one has based on visual inspection of the charge differences. For example, based on the charge distribution graphs and K values, one would predict that the CBm-4 mutant would have the strongest effect in the LLPS assay, but it doesn’t appear so. Do the authors have a hypothesis about this? Also, it would be helpful to show the values of K for each of the repeats analyzed in Extended Figure 1a.

Response: We highly appreciate this comment. Other reviewers raised a similar concern about the quantification of LLPS based on the microscopic images and the relationship between LLPS and charge blockiness evaluated based on the kappa value. To solve this problem, we first examined several new parameters for charge blockiness and then conducted a more quantitative LLPS assay utilising a turbidity assay. The details of the new analyses are given below.

i) Turbidity assay for the quantification of LLPS

To quantify LLPS, we used a turbidity assay in which the OD_{600} of the droplet solution was measured and plotted against the protein concentration (Larson et al., *Nature*, 2017). By fitting a sigmoidal curve, the saturation concentration (C_{sat}), which represents the protein concentration with half-maximal turbidity, was obtained. This enabled us to quantify the LLPS propensity by comparing C_{sat} values obtained under the same experimental condition. We performed the turbidity assay in all droplet assays presented in the manuscript (Fig. 1d, e, 2b, 5b, and Extended Data Fig. 2b) and the data are shown in the revised figures (Fig. 1f, 2c, and Extended Data Fig. 2c, 5c). The original data have been provided in the Supplementary Table. We believe that the quantification of LLPS is a significant improvement from fluorescence image analysis.

ii) Evaluation of charge blockiness

The Kappa value has been used to evaluate charge segregation in synthetic polymers composed of only positively and negatively charged monomers (Das and Pappu, *P.N.A.S.*, 2013). However, this value might not be suitable for actual polypeptides, which comprise only 10–20% charged residues and polypeptides with different net charges (e.g. phosphorylated and de-phosphorylated proteins). Several other methods are available to evaluate charge segregation along a polymer chain, including sequence charge decoration (SCD) (Sawle and Ghosh, *J. Chem. Phys.*, 2015), the degree of segregation (D_{seg}), and the blockiness of like charge (B_{LC}). As SCD has been demonstrated to show a significant correlation with kappa (Sawle and Ghosh, *J. Chem. Phys.*, 2015), we investigated D_{seg} and B_{LC} for evaluating the charge blockiness of Ki-67.

D_{seg} has been used to evaluate the statistical difference in the positions of two different groups and is defined as per the following equation:

$$D_{\text{seg}} = -\log_{10}(\text{p-value}_{\text{U-test}(p+, p-)}),$$

where $p+$ and $p-$ represent the position of positively and negatively charged amino acids along the polypeptide, respectively.

B_{LC} evaluates the distances among the residues within the same charge group (like charges) and is defined as per the following equation:

$$B_{\text{LC}} = (C_{\text{max}(+)} + C_{\text{max}(-)}) * \frac{\sum_{k=1}^{Nd(+,-)} d_k(+,-)}{Nd(+,-)} / \left(\frac{\sum_{k=1}^{Nd(+,+)} d_k(+,+)}{Nd(+,+)} + \frac{\sum_{k=1}^{Nd(-,-)} d_k(-,-)}{Nd(-,-)} \right),$$

where $C_{\text{max}(+)}$ and $C_{\text{max}(-)}$ represents the absolute maximum positive and negative values in the charge plot, respectively, $d(+, +)$, $d(-, -)$, and $d(+, -)$ represent the distance of a pair of charged residues (+: positive charge, -: negative charge) and $Nd(+, +)$, $Nd(-, -)$, and $Nd(+, -)$ represent the numbers of these pairs, respectively.

As shown in the panels below, the kappa value is less sensitive to small charge segregations than D_{seg} , whereas B_{LC} and D_{seg} show good correlation over a wide range of charge segregations. Therefore, we decided to use B_{LC} and D_{seg} values instead of kappa in this study. B_{LC} values are

now indicated in all charge plots in the revised figures (Fig. 1b, 2b, 5c and Extended Data Fig. 1a, 1b, 2b, 2f).

(A) Correlation between κ and D_{seg} of R12 mutants used in this study. (B) Dotted square in (A) is enlarged. (C) Correlation between B_{LC} and D_{seg} .

iii) Correlation analysis between charge blockiness and LLPS propensity

We combined the results from i) and ii) described above and analysed the relationship between LLPS propensity and charge blockiness. C_{sat} exhibited a strong correlation with both B_{LC} (Fig. 2c) and D_{seg} (Extended Data Fig. 2e). To confirm this, we constructed a dozen additional mutants with different charge blockiness and net charges (Extended Data Fig. 2f; this was pointed out by Reviewer #3). As shown in revised Fig. 2d and Extended Data Fig. 2g, C_{sat} strongly correlated with charge blockiness (B_{LC} and D_{seg}) but not the net charge (Extended Data Fig. 2h).

As a concern of the reviewer, some fluorescence images of the droplets do not seem to match the charge blockiness. This is because the size and brightness of the droplets vary widely, and small droplets are sometimes difficult to observe. Generally, the results of the image analysis and turbidity assay showed good agreement, and the turbidity assay is more quantitative (compare Fig. 2c and Extended Data Fig. 2a).

Based on all results described above, we strongly believe that the correlation between charge blockiness and LLPS is convincing. We once again wish to express our appreciation of this useful comment. We have described the new results in the main text (page 7, line 117 – 124), as well as Methods section (page 19, line 318 – 330).

4. Fig. 1d- these are GFP-tagged, right? That should be indicated in the schematic.

Response: We thank the referee for pointing this out. The fluorescence signal shown in Fig. 1d is not from EGFP but from ATTO488. We apologise our insufficient explanation of the experimental procedure. We have added the following description in the legend of Fig. 1 (page 5, line 83 – 84): “*In vitro* LLPS assay using ATTO488-labelled recombinant proteins.”

5. Figure 1e- It would be more accurate to label the panels as WT-(R12)1 than as WT.

Response: We thank the referee for pointing this out. We have made the change accordingly in Fig. 1e (Fig. 1h in the revised manuscript).

6. Extended Fig. 1- the graphs of the mitotic vs interphase charge in panels a and b would be easier to analyze if they were graphed in more visually distinct color and thickness.

Response: We thank the referee for pointing this out. We have ensured consistent colours in Extended Data Fig. 1a, as well as in other charge plots (Extended Data Fig. 5f). The legend has also been revised accordingly (page 28, line 560 – 561 and page 40, line 679 – 680).

7. Extended Fig. 1e- is this the correct Figure and Legend? What is shown looks like fluorescence intensity analysis of R12 multimers of different lengths, but the legend described this as quantitation of the kinetic mixing experiment in panel Extended 1d.

Response: We apologise if our description caused some confusion. Extended Fig. 1e is the statistical analysis of the protein droplet shown in Fig. 1d in the main text (fluorescence intensity analysis of R12 multimers of different lengths), not Extended Data Fig. 1d. We have clarified this in the revised manuscript (page 28, line 565 – 567).

8. Extended Fig. 2a- Why are values > 104 dark-shaded here and not in the other measurements of drop intensity (Ext. Figs. 1c and 1e)? And what are the values shown in these panels?

Response: We apologise for this insufficient description. The shaded bars in the histograms represent the samples above a threshold value. To distinguish protein droplets from background small particles, we set a threshold value for the fluorescence intensity. As we additionally conducted a turbidity assay to quantify LLPS (described in our response to comment #3), we believe that the threshold in the histogram is no longer necessary and thus it was deleted in the revised manuscript.

9. Extended Fig. 3d- add the label “LR-free constructs”

Response: We have added the label “LR-free constructs” in Extended Data Fig. 3d (Extended Data Fig. 4d in the revised manuscript).

10. Figure 4c- The meaning of the arrowheads and asterisks should also be in the Legend.

Response: We have described what the arrowheads and asterisks indicate in the legend of Fig. 4c (Fig. 5c in the revised manuscript) (page 16, line 262 – 263).

11. Extended Fig. 4c- label with “anti-B23 antibody detection”

Response: We have added the label “anti-B23 antibody” in Extended Data Fig. 4c (Extended Data Fig. 5d in the revised manuscript).

12. Extended Fig. 5d. Next to the cartoons of Ki-67 and B23 at the bottom, there should be an indication that blue = DNA.

Response: We have added “blue: DNA” in Extended Data Fig. 5d (Extended Data Fig. 5e in the revised manuscript) to address this (page 40, line 676).

Dr. Paul Kaufman

U. Massachusetts Medical School

Reviewer #2:

Remarks to the Author:

The manuscript by Yamazaki et al. describes that alternating charge blocks of Ki-67 and B23 tuned by phosphorylation regulate their liquid-liquid phase separation (LLPS) involved in chromosome periphery formation and nucleoli dissolution during cell-cycle. The authors reported that mitotic hyper-phosphorylation of multiple residues throughout the entire repeat domain (RD) of Ki-67 generates multivalent charge blocks which promotes LLPS of Ki-67 LLPS and is essential for Ki-67-associated mitotic chromosome periphery formation in cells. In addition, the authors presented interesting pieces of data that mitotic phosphorylation reduces charge blocks of B23, thereby inhibiting its LLPS and assisting the disassembly of nucleolus in mitosis.

Overall, this work describes a couple of interesting observations, and comes up with two major findings including (1) new mechanism for regulation protein LLPS (charge blockiness by phosphorylation); (2) discovery of Ki-67 and B23 LLPS in mediating their physiological function in cell cycle. However, several previously studies have reported similar phenomena and LLPS mechanisms, which great reduce the novelty of this work. For instance, Zhou et al. (Angew Chem Int 2019) reported that charge blockiness is important in mediating protein LLPS. Cuylen et al. (Nature, 2016) suggested that Ki-67 forms a steric and electrostatic charge barrier, similar to surface-active agents (surfactants) that disperse particles or phase-separated liquid droplets in solvents. Booth et al. (eLife, 2014) reported the function of Ki-67 in organizing the mitotic chromosome periphery. This work claims that Ki-67 LLPS is important in mediating its function. However, the data only supports in part that Ki-67 can undergo LLPS in vitro and in cell involved in periphery organization. Whether Ki-67 LLPS is essential for the chromosome periphery organization and why LLPS of Ki-67 is capable of inducing periphery organization remain unanswered. Moreover, the overall quality of data is relatively low. Some were over-interpreted and cannot be used to support the author's claims. Please see my detailed concerns below.

Major concerns:

1. Despite that Ki-67 can undergo LLPS in vitro and in cell, the authors didn't provide convincing evidence to support Ki-67 LLPS is essential for its cellular function. Whether knock-in of Ki-67 variants (LLPS mimetic or LLPS deficient) can rescue Ki-67 knockout (KO)-phenotype can provide some clues on this. However, the data (Fig.3 and Fig.S3) is not convincing. The readout of the "KO-phenotype" is too simply (only calculate the convex hull area of mitotic chromosomes). Moreover, the

25difference of convex hull area between different variants are not obvious, and the variation within the same sample is huge (Fig. S3i). The authors need additional quantitative assay to study this.

Response: We appreciate this suggestion. Ki-67 KO cells have several distinct phenotypes, including the aggregation of mitotic chromosomes and lack of chromosome peripheries. The convex hull, which represents how the mitotic chromosomes are dispersed in a 2D microscopic image (Cuylen et al., 2016), was used in our study to evaluate whether Ki-67 mutants can rescue the KO phenotype. As the reviewer pointed out, some differences were small, but they were statistically significant (Extended Data Fig. 3i in the original manuscript). Therefore, we conducted the following additional experiments and analyses.

i) Chromosome dispersion evaluated by the bounding box

We obtained 3D stack images of mitotic chromosomes and calculated a ‘bounding box’, which is a minimum box to accommodate all mitotic chromosomes in a cell. The volume of this box was obtained and plotted in individual mutants. As shown in the revised Extended Data Fig. 4h, cells expressing the WT, phosphomimetic mutant (Pm9), or charge-block mutant (CBm-3) showed a larger bounding box than non-expressing cells. As a result, the difference in median values was larger (27–78% increased) and the variation was smaller (36% reduction in width between the maximum and minimum) in the bounding box analysis relative to that in the convex hull analysis.

ii) Segregation of Ki-67 and DNA signal on mitotic chromosomes

In addition to chromosome dispersion, we analysed the localisation of Ki-67 mutants around chromosomal DNA, which directly indicates the formation of the chromosome periphery (Booth et al., *eLife*, 2014; Hayashi et al., *Biochem. Biophys. Res. Commun.*, 2017). If the Ki-67 mutant forms the chromosome periphery, it should localise outside the chromosomal DNA, which would result in the segregation of the fluorescence signal of Ki-67 (EGFP) and DNA (Hoechst). As shown in Extended Data Fig. 4i in the revised manuscript, the correlation coefficient between EGFP and the Hoechst signal was significantly reduced with full-length Ki-67, (WT)₁₂-LR, (Pm9)₁₂-LR, (CBm-3)₁₂-LR, and (R7_WT)₁₂-LR but not with (A9)₁₂-LR and (R7_A7)₁₂-LR.

The results of these two additional analyses have been added in Extended Data Fig. 4h, i. We believe that it is now clear that the phosphorylated, phosphomimetic and charge-block mimetic forms of R12-LR can rescue the phenotype Ki-67 KO cells. We have also described these findings in the main text (page 11, line 189 – 192).

2. The author studied the LLPS behavior of Ki-67 WT and its variants mainly in aqueous solution. However, *in vivo*, Ki-67 is attached on chromosome surface. The Ki-67 LLPS properties drawn from solution study cannot be applied to that on the chromosome surface. Based on the data in this study, it's hard to know why Ki-67 LLPS is essential in mediating periphery organization. The authors attempted to use DNA decorated beads to mimic an “artificial chromosome surface”. Despite that Ki-67 variants can form a ring-like structure on the bead (Fig.2C), it CAN'T support that Ki-67 undergoes LLPS on the surface (as the authors claimed, Page7, line 125). This data can only tell Ki-67 variants can attach on the bead. The authors need to establish a proper model to firstly directly observe that Ki-67 WT indeed undergoes LLPS on the DNA surface, and then examine the behavior of different variants.

Response: We appreciate this comment. To demonstrate the LLPS of Ki-67 R12 on the DNA-coated beads, we conducted the following additional experiments.

i) Examining inter-molecular interactions among RDs

As shown in Fig. 2c in the original manuscript, the protein layer on the DNA-coated beads was stronger for Pm9 and CBm-3 than for WT. This suggested that not only LR-dependent DNA binding, but also intermolecular interactions between RDs, occurred in the protein layer. To confirm this, we performed an experiment in which LR-free R12 (fluorescently labelled) was added to the beads coated with DNA and LR-fused R12. As shown in Fig. 3c in the revised manuscript, LR-free Pm9 and CBm-3 were more strongly incorporated into the protein layer than WT. These results demonstrated that intermolecular interactions between RDs contribute to the formation of the protein layer on the DNA beads.

ii) FRAP analysis of the protein layer

Based on the above results, we conducted fluorescence recovery after photobleaching (FRAP) analysis of the fluorescence signal on the beads. As shown in Fig. 3e in the revised manuscript, Pm9 and CBm-3 showed recovery of the signal, which was similar to that observed in the droplet in solution (Extended Data Fig. 1d). These results indicated that RDs form a liquid-like phase on the bead.

In addition, simulation studies on the phase separation of the grafting polymer brush (polymer with one end attached to a solid substrate) demonstrated that the grafting polymer undergoes phase separation under a phase diagram similar to that in free solution (Norizoe et al., *Europhys. Lett.*, 2013; Murakami, *Macromol.*, 2016).

We have rearranged all of the original and newly obtained results into new Fig. 3 in the revised manuscript. The main text (page 7, line 134 – 141) and the figure legend (page 9, line 156 – page 10, line 168) have also been revised accordingly. Based on all of the above results, we strongly believe that Ki-67 RD forms a liquid-like layer on the DNA beads in a phosphorylation-dependent manner.

3. To investigate LLPS mechanism of Ki-67 in vitro and in cell, the authors mainly focused on R12, the first repeat domain of Ki-67. (R12)_n and its variants were used in the in vivo assay. However, Ki-67 contains 16 repeats. R1-R16 display different charge patterns and their changes of charge patterns in mitosis also show significant difference as shown in Figure S1. It's questionable whether the R12 tandem repeats can fully recapitulate the behavior of Ki-67 full-length (FL) in the in vitro or cellular assay (Figure 2 and Figure 3). The authors need to prepare Ki-67 constructs with different RDs (e.g. R1-R7 and R8-R16) and examine them in the in vitro and cell model.

Response: We highly appreciate this useful suggestion. To confirm that R12 represents other repeats in RD, we constructed the following mutants and performed both *in vitro* and *in vivo* experiments.

i) The homogeneous repeat of R7 can also rescue the KO phenotype

As shown in Extended Data Fig. 2b in the original manuscript, phosphomimetic mutations enhanced the LLPS of R7 in an *in vitro* droplet assay. Based on this result, we constructed a homogeneous repeat of R7 ((R7_WT)₁₂-LR, (R7_A7)₁₂-LR) and examined whether it could rescue the Ki-67 KO phenotype, as we demonstrated for R12. As shown in revised Extended Data Fig. 4h, i, k, l, (R7_WT)₁₂-LR localised to the chromosome periphery in HeLa cells and rescued the phenotype of Ki-67 KO cells, whereas the non-phosphorylatable mutant (R7_A7)₁₂-LR could not. These results demonstrated that not only R12, but also R7, can form a functional chromosome periphery when tandemly repeated and fused with LR.

ii) A stretch of RD between R8–16

In addition to the homogenous repeats of R7 and R12, we investigated a heterogeneous repeat. As shown in Extended Data Fig. 1c of the original manuscript, an RD fragment encompassing R8 to R16 (R8–16) was phase-separated *in vitro*. Based on this result, we newly constructed (R8–16)-LR and examined whether it can rescue the Ki-67 KO phenotype. As shown in the panels below, this construct could form a chromosome periphery in KO cells. However, the dispersion of the mitotic chromosome was not statistically significant, probably due to the smaller repeat number (nine repeats between R8 and R16 compared to 12 repeats for the other homogeneous constructs).

28

Left: Localisation of R8-16-LR in mitotic HeLa and Ki-67 KO cells. DNA was stained with Hoechst.

Right: Statistical analysis of the morphology of mitotic chromosomes. The bounding box and correlation coefficient were measured as described in Extended Data Fig. 4h, i.

We believe that the result of R7 described in (i) is sufficient to support the contention that not only R12, but also other repeats, can form a functional chromosome periphery when tandemly repeated. The *in vitro* (Extended Data Fig. 2b, c) and *in vivo* (Extended Data Fig. 4h, i, k, l) analyses of R7 have been added in the revised manuscript. The main text has also been revised accordingly (page 6, line 105 – 106 and page 12, line 194 – 198).

- The authors showed a working model of interplay between Ki-67 and B23 (Fig.S4) without experimental evidences. The authors may need to explore their potential co-localization in different steps of cell cycle in their model.

Response: We appreciate this comment. The model presented in Extended Data Fig. 4 in the original manuscript was constructed from our experimental results of homogeneous tandem repeats of Ki-67

RD and B23. To provide experimental evidence of their behaviours, we simultaneously observed full-length Ki-67 and B23 during mitosis under a fluorescence microscope. The images have been added in revised Extended Data Fig. 5e. Together with the results from the biochemical interaction assay of Ki67 and B23 (Extended Data Fig. 5d), these data have better clarified the behaviour of Ki-67 and B23.

We believe that the dynamic disassembly/reassembly of nucleoli and the chromosome periphery is governed not only by Ki-67 and B23 but also by other nucleolar proteins listed in Extended Data Fig. 5f (as well as by RNAs). It would be highly interesting and necessary to investigate how mitotic phosphorylation regulates the interactions among them and controls the macroscopic behaviours of the entire nucleolus. We believe this is another large topic of study that warrants investigation in the future.

5. Previous article (Sara Cuylen-Haering et al. Nature 2020) showed that Ki-67 exhibits extended conformation at the chromosome periphery in the metaphase, but collapse during mitotic exit. How does conformational change between extended and collapse states influence Ki-67 LLPS behavior?

Response: By using dual-tagging (EGFP and mCherry), the group of Cuylen-Haering showed that Ki-67 exists in a partially extended form at the chromosome periphery during mitosis (Cuylen et al., *Nature*, 2016, Cuylen-Haering et al., *Nature*, 2020). They proposed an extended polymer brush model of Ki-67, in which Ki-67 has a preferential partly extended form owing to its dense grafting and electrostatic repulsions among different Ki-67 molecules on the surface of mitotic chromosomes. Our results from the *in vitro* droplet assay demonstrated that the mitotic phosphorylated form of RD has stronger molecular interactions and higher propensity for LLPS than the non-phosphorylated form. Although we understand the reviewer's concern, we believe that our results are compatible with the partially extended model based on the following considerations:

- i) Density of Ki-67 on the mitotic chromosome surface

The conformation of a grafting polymer on a solid substrate depends on the density of the polymer (Milner, *Science*, 1991); as the density increases, the polymer prefers to be extended, due to the mutual exclusion effect. Considering that Ki-67 binds to chromosomes via LR, it can be expected that the density of Ki-67 increases as the chromosomes condense at early mitosis. This high density of Ki-67 on the mitotic chromosome surface shifts (drives) the conformation of the RDs from a compact 'mushroom' form towards the extended form.

ii) Intermolecular interactions between RDs

In addition to the effect described above, the intermolecular interactions between RDs contribute to the extended conformation. As demonstrated in Fig. 3, R12 of Ki-67 bound to the bead surface via intermolecular interactions. Alternating charge blocks would greatly contribute to this interaction. Even ‘free’ R12 (Pm9 and CBm-3) was incorporated into the liquid protein layer on the bead, indicating that intermolecular interactions among RDs occur even on the solid surface. This result strongly supports the possibility that intermolecular interactions among RDs occur on the mitotic chromosome surface. Cuylen et al. (*Nature*, 2016) expected electrostatic repulsion among Ki-67 molecules based on the positive net charge simply calculated from the amino acid composition (+122 in the whole molecule and +81 in the RD). However, we expect that the electrostatic repulsion becomes weaker owing to hyperphosphorylation and that intermolecular interactions among RDs become stronger via enhanced charge blocks during mitosis. This effect would also help Ki-67 adapt to the proposed extended conformation.

iii) Interaction with B23

The conformation of Ki-67 (RD) is determined not only by interactions among RDs but also by interactions with B23 via its N-terminus. In interphase cells, Ki-67 binds to both chromosomes and B23 (Fig. 5d, Extended Data Fig. 3a, 5d). Since B23 exists exclusively in nucleoli and does not co-localise with chromosomes in interphase cells (Fig. 5c), Ki-67 mainly exists at the border between nucleoli and chromosomes (Extended Data Fig. 5e). When cells enter mitosis, however, Ki-67 dissociates from B23 (Extended Data Fig. 5d, 5e) and binds to the chromosome surface (Fig. 5d, Extended Data Fig. 3a, 5e). As B23 is dispersed in the cytoplasm owing to mitotic phosphorylation (Fig. 5c, Extended Data Fig. 5e), Ki-67 is anchored to the chromosome surface only at its C-terminus, and the N-terminus is facing the cytoplasm. Such a ‘free-end’ situation would also help Ki-67 adapt to the proposed extended conformation.

Thus, we believe it reasonable to assume that the Ki-67 RD exists in partially extended forms by interacting with other molecules. In the revised manuscript, we have explained this in the main text (page 17, line 285 – page 18 line 293). We fully appreciate that an understanding of the exact conformations of Ki-67 at the chromosome periphery is an important problem to be addressed in the future.

Technical issues:

6. Protein LLPS conditions (e.g. protein concentration) are not indicated. Does Ki-67 undergo LLPS in vitro at the concentration similar to its physiological in vivo concentration?

Response: We apologise for our insufficient description of the experimental procedures. We have mentioned how we measured protein concentrations for the droplet assay in the Methods section (page 21, line 385 – 401). Normally, the final protein concentration in the droplet assay was 40 μM , unless otherwise indicated. In the turbidity assay, we used a wide range of protein concentrations (0.01–100 μM), which is indicated in the individual graphs of the revised manuscript.

In the LLPS assay with a single repeat, the final protein concentration was $\sim 100 \mu\text{M}$, which is higher than the physiological concentration. However, as shown in revised Fig. 1g and f, the propensity for LLPS largely depends on the repeat number; a protein with eight or nine repeats undergoes LLPS at the sub- μM range, which is reasonable in a cellular environment. Therefore, it is highly probable that full-length Ki67 (16 repeats) undergoes LLPS under physiological conditions.

7. In fluorescence-labeling assay, what's the molar ratio of the protein and fluorescence dyes? It's unknown whether the authors removed the free fluorescent molecules (from spontaneous hydrolysis of maleimide or the dye reacted with DTT) existing in labeled-protein solution. Does the free dye affect the fluorescent signal in Figure 2c? The authors need to ensure that (WT)₂-LR and its variants in Figure 2c have same fluorescence-labeling efficiency before analyzing their fluorescent signal around the beads?

Response: The molecular ratio of the protein to the fluorescent maleimide dye was 400:1. This low ratio was i) because not all protein is necessarily labelled in the LLPS assay and ii) to reduce the effect of the fluorescent moiety on protein behaviour. Therefore, we expected that only a small amount of free (unreacted) dye remained in the protein solution, which was even further quenched by additional DTT. We have added this information to the Methods section (page 21, line 385 – 388).

In addition, as suggested by the reviewer, we have examined the labelling efficiency of proteins used in the beads assay by fluorescence imaging of the SDS gel. As shown in revised Extended Data Fig. 3b, the (WT)₂-LR, as well as (Pm9)₂-LR and (CBm-3)₂-LR, showed comparable fluorescence signals on the SDS gel. We have described the labelling efficiency in the main text (page 7, line 131 – 133).

8. In Figure 2b, the phase separation of WT and Pm1 seems like irregular aggregates and the droplets of

32

Pm6 and Pm9 contain many holes. Did those pictures represent the general morphology of proteins under this condition? What's more, the author did not mention the incubated time before the microscopic observation in method. Does the long incubation or surface material properties of 96-well plate induce the formation of irregular aggregates?

Response: We thank the referee for pointing out the morphology of the droplet. As the reviewer highlighted, some droplets shown in Fig. 2b contained small holes inside. The droplet was observed under a fluorescence microscope 30 min after mixing with the droplet buffer. We examined whether the holes inside the droplet increased during this incubation time, as suggested by the reviewer. However, we did not see such a tendency. We also examined the effect of the surface of the 96-well plate by pre-coating the inner walls of the wells with bovine serum albumin (1 mg/mL in PBS). However, we did not see any effect on droplet morphology and dynamics.

Alternatively, as pointed out by Reviewer 3, this type of droplet morphology could have been caused by RNA contamination during protein purification from bacterial cells. Therefore, we quantified RNA in the purified protein solutions by using a high-sensitive RNA-specific fluorescent probe. The amount of RNA in the protein solution was 0.1–0.7 % (w/w). Therefore, we concluded that RNA contamination was not a major reason for the holes in the droplet. The description on the purity of protein sample has been added to Method section of the revised manuscript (page 21, line 380 – 382).

The morphology of liquid droplets provides important information regarding the mechanism of micro-phase separation of polyampholytes. It is tightly correlated with both the properties of the polymer (e.g. length, charge distribution, density) and environmental factors (e.g. temperature, solvent, salt). Although we strongly believe that mechanistic insights would be obtained if droplet morphology would be more carefully investigated, we are of the opinion that this is beyond the scope of this study and will be thoroughly investigated in the future.

9. To quantify the phase separation ability of different proteins, fluorescent intensity of individual droplet in the images was quantified and summarized in the histogram. However, the droplet surface properties of different proteins and the stack of different collected pictures may affect fluorescent signal and is hard to be controlled. The macroscopical measurement (turbidity like OD_{600nm} or OD_{350nm}) (centrifugation for detecting components of droplets) or other assay may need to be supplied to verify their LLPS ability.

Response: We highly appreciate this useful suggestion. Accordingly, we have quantified the propensity for LLPS by measuring the turbidity of the protein solution at OD₆₀₀. By fitting with a sigmoidal curve, the saturation concentration (C_{sat}), which represents the protein concentration with

33

the half-maximal turbidity, was obtained. This enabled us to quantify the LLPS propensity by comparing the C_{sat} values obtained under the same experimental condition. We performed this turbidity assay for all droplet assays presented in the manuscript and summarised the results in revised Fig. 1f, 1g, 2c, 2d and Extended Data Fig. 2c, e, g, h, and 5c. This analysis enabled us to not only compare the LLPS propensity among the RD mutants but also to show a correlation between LLPS (C_{sat}) and charge blockiness (D_{seg}) as described in the following response (to comment #10).

10. The authors assessed the charge blockiness along the polypeptide by using k value and claim the k is positively correlated with protein LLPS. However, in Figure 2b, CBm-4 has highest k value (8.1×10^{-3}), but its droplets are smaller than Pm6 (k value is 1.3×10^{-3}).

Response: We appreciate this comment. In the original manuscript, we used the kappa value to evaluate the charge blockiness of the polypeptide. This value is useful to evaluate charge segregation in a polyelectrolyte that is composed of only positively and negatively charged monomers, as demonstrated in a previous study (Das and Pappu, *P.N.A.S.*, 2013). However, as other reviewers also pointed out, this value might not be suitable for actual polypeptides, which comprise only 10–20% charged residues, and the net charge can be changed by phosphorylation. There are several other methods to evaluate charge segregation along a polymer chain, including sequence charge decoration (*SCD*) (Sawle and Ghosh, *J. Chem. Phys.*, 2015), the degree of segregation (D_{seg}), and the blockiness of like charge (B_{LC}). As *SCD* has been demonstrated to show a significant correlation with kappa (Sawle and Ghosh, *J. Chem. Phys.*, 2015), we investigated D_{seg} and B_{LC} for evaluating the charge blockiness of Ki-67.

D_{seg} has been used to evaluate a statistical difference in the positions of two different groups and is defined as per the following equation:

$$D_{\text{seg}} = -\log_{10}(\text{p-value}_{\text{U-test}[p+, p-]}),$$

where $p+$ and $p-$ represent the position of positively and negatively charged amino acids along the polypeptide, respectively.

B_{LC} evaluates the distances among the residues within the same charge group (like charges) and is defined as per the following equation:

$$B_{LC} = (C_{max(+)} + C_{max(-)}) * \frac{\sum_{k=1}^{Nd(+,-)} d_k(+,-)}{Nd(+,-)} / \left(\frac{\sum_{k=1}^{Nd(+,+)} d_k(+,+)}{Nd(+,+)} + \frac{\sum_{k=1}^{Nd(-,-)} d_k(-,-)}{Nd(-,-)} \right),$$

where $C_{max(+)}$ and $C_{max(-)}$ represents the absolute maximum positive and negative values in the charge plot, respectively, $d(+, +)$, $d(-, -)$, and $d(+, -)$ represent the distance of a pair of charged residues (+: positive charge, -: negative charge) and $Nd(+, +)$, $Nd(-, -)$, and $Nd(+, -)$ represent the numbers of these pairs, respectively.

As shown in the panels below, the kappa value is less sensitive to small charge segregations than D_{seg} , whereas B_{LC} and D_{seg} show good correlation over a wide range of charge segregations. Therefore, we decided to use B_{LC} and D_{seg} values instead of kappa in this study. B_{LC} values are now indicated in all charge plots in the revised figures (Fig. 1b, 2b, 5c and Extended Data Fig. 1a, 1b, 2b, 2f).

(A) Correlation between kappa and D_{seg} of R12 mutants used in this study. (B) Dotted square in (A) is enlarged. (C) Correlation between B_{LC} and D_{seg} .

Using B_{LC} and D_{seg} values together with C_{sat} described in our response to the previous comment (comment #9), we analysed the relationship between LLPS and charge blockiness. As shown in the revised manuscript, C_{sat} strongly correlated with B_{LC} (Fig. 2c) and D_{seg} (Extended Data Fig.

2e). In addition, we constructed a dozen R12 mutants that had different charge blockiness and/or net charges (Extended Data Fig. 2f) and performed the same analysis. The results are summarised in Fig. 2d and Extended Data Fig. 2g. From all of these results, it is clear that LLPS is closely related to the charge blockiness of the polypeptide.

As a concern of the reviewer, some fluorescence images of the droplets do not seem to match the charge blockiness. This is because the size and brightness of the droplets vary widely, and small droplets are sometimes difficult to observe. Generally, the results of the image analysis and turbidity assay showed good agreement, although the turbidity assay is more quantitative (compare Fig. 2c and Extended Data Fig. 2a).

Minor issues:

11. The author didn't show localization of homogeneous repeat construct of Ki-67 in interphase in Figure 4d, but they mentioned this in Page13, 204-206.

Response: We apologise for this discrepancy between the main text and the figure. We have added microscopic images of interphase cells in Fig. 4d (Fig. 5d in the revised manuscript).

12. A control of CDK1+ATP should be added in Figure 1e.

Response: We have performed an additional experiment using ADP. The revised Fig. 1h now contains three conditions (+ATP, +ATP/CDK1, +ADP/CDK1).

13. Some of the images are of poor quality, as shown in Fig. 1d and 1e.

Response: We thank the referee for pointing this out. We have replaced the images in Fig. 1d and 1e with high-quality ones (Fig. 1e and h in the revised manuscript).

Reviewer #3:

Remarks to the Author:

The authors present an extremely clear and well-written study examining the effect of charge blockiness on cellular localization and chromosomal dynamics. They focus on two proteins and provide evidence that

supports charge patterning as a possible determinant of function, recruitment, and interaction. I thoroughly enjoyed the paper and found it easy to follow, compelling and interesting.

The insights are novel, although there are quite a few citations to key previous papers are missing (raised below). I have several suggestions below, and while there are a number of absolutely essential components that need to be addressed, but my biggest concern is that the authors do not - in my opinion - fully delineate between a model in which charge patterning matters in vivo vs. simply a change in the net charge. This absolutely can be addressed with a variant that distributes the charged residues uniformly while holding the other residues fixed, and/or by defining a minimally perturbative variant that breaks the charge blocks and showing there is a loss of function in vivo. I want to be clear that I actually do think the authors are right, but, I worry a critic may look at this and conclude that they have shown phosphorylation is essential (a well-established result) and move on.

My specific concerns are outlined below. I consider everything I have raised as requiring attention, but I do want to draw attention to my concern about in vitro purify of samples re: RNA contamination. I don't think this concern alters the authors' conclusions at all, but, for the sake of maintaining rigorous and high-caliber science is essential to address.

Major comments

1. The authors claim that the repeat number showing a similar tendency to phase separate demonstrates that repeat number rather than amino acid sequence determines phase separation. This sets up a potentially false dichotomy - both the repeat number and specific amino acid sequence could matter, or repeat number and amino acid composition could be the key determinant. However, as it stands, the results do not show that "the repeat number (multi-valency), rather than the amino-acid sequence of individual repeats, is important for LLPS". There are a couple of ways to address this

At a minimum, the dependence of the saturation on repeat number should be shown and qualitatively (or ideally quantitatively) compared to expectation from polymer theory.

Additionally, IF the authors wish to propose that a specific amino acid sequence is unimportant they must show that a shuffle of R12 (s-R12) shows the same length-dependence as the number of repeats is altered. Given the authors later show that charge patterning matters, it seems unlikely to me that a random shuffle would recapitulate wild-type behavior, invalidating the authors' statement, so my suggestion would simply be to remove that implication that amino acid sequence is not important.

Response: We highly appreciate these comments. As per the reviewer's suggestion, we have performed a turbidity assay to clarify the relationship between the repeat number and LLPS. As shown in revised Fig. 1f and g, the saturation concentrations (C_{sat}) of (R12)₁, (R12)₂, (R12)₄, and

(R12)₈, which were obtained from the turbidity assay, sharply decreased as the repeat number increased. This implies that the propensity for LLPS increases as multi-valency (copy number) increases, which corresponds well with the theory of polymer phase separation. However, we agree that our claim that the ‘repeat number rather than amino acid sequence determines phase separation’ was an overstatement. Therefore, we revised this sentence as follows:

(page 6, line 98 – 100)

“A clear inverse correlation between the number of repeats and the propensity for LLPS (quantified by the saturation concentration, C_{sat}) was observed; C_{sat} sharply decreased as the repeat number increased (Fig. 1f, g).”

2. I do not understand how the authors are calculating kappa, but it seems to be incorrect - this can be verified by passing the sequence into the CIDER webserver (<http://pappulab.wustl.edu/CIDER/analysis/>). The red flag for me is that a value of 0.0013 is incredibly small. Based on the correct definition of kappa, the wild-type R12 sequence has a kappa of 0.264, while the phosphorylated version has a value of 0.211 (see figure below). This is somewhat problematic in that the phosphomimetic variant (which phase separates more strongly than wild-type) actually has a lower (less blocky) sequence than the wild-type sequence. This sort of undermines the authors' model - in fact their data sort of shows the opposite, at least for WT and the Pm9 sequence. I didn't have the energy to manually type out the other sequences, but I think if the authors are going to argue that charge patterning influences phase separation an essential figure needs to show the saturation concentration vs. kappa for the set of permutations described in Fig. 2 (and the C_{sat} vs. net charge as a supplementary figure to confirm this is not the determinant of assembly). Without a clear quantification of this trend it's impossible for me to really evaluate if this model is correct or not (although to be clear, it could well be and it does make sense, so, don't take these comments as skepticism with respect to the model!). Again though, the key issue is that fluorescence images do not provide any means to really quantify the driving forces for phase separation.

Response: We highly appreciate this comment. In the original manuscript, we used the kappa value to evaluate the charge blockiness of the polypeptide (Das and Pappu, *P.N.A.S.*, 2013). The Kappa value contains a parameter that represents the size of the charge block (g). In this paper, the g value was set as 5 and 6, because the polymer length was fixed at 30. However, in actual polypeptide chains, the block size varies from protein to protein and should be defined individually. After considering these issues, we found that $g = 40$ was reasonable to evaluate the charge blockiness of the Ki-67 repeat domain. As it was not clear which g value the CIDER web server uses to calculate the kappa value, and we cannot change it, we calculated the g value ourselves, with $g = 40$. This is why our kappa values are different from those returned by the CIDER web server.

Though the kappa value is useful for evaluating the charge segregation in a synthetic polyelectrolyte that is composed of only positively and negatively charged monomers, we found that it might not be suitable for actual polypeptides, which comprise only 10–20% charged residues. There are several other methods to evaluate charge segregation along a polymer chain, including sequence charge decoration (*SCD*) (Sawle and Ghosh, *J. Chem. Phys.*, 2015), the degree of segregation (D_{seg}), and the blockiness of like charge (B_{LC}). As *SCD* has been demonstrated to show a significant correlation with kappa (Sawle and Ghosh, *J. Chem. Phys.*, 2015), we investigated D_{seg} and D_{like} for evaluating the charge blockiness of Ki-67.

D_{seg} has been used to evaluate a statistical difference in the positions of two different groups and is defined as per the following equation:

$$D_{\text{seg}} = -\log_{10}(\text{p-value}_{\text{U-test}(p+, p-)}),$$

where $p+$ and $p-$ represent the position of positively and negatively charged amino acids along the polypeptide, respectively.

B_{LC} evaluates the distances among the residues within the same charge group (like charges) and is defined as per the following equation:

$$B_{\text{LC}} = (C_{\text{max}(+)} + C_{\text{max}(-)}) * \frac{\sum_{k=1}^{Nd(+,-)} d_k(+,-)}{Nd(+,-)} / \left(\frac{\sum_{k=1}^{Nd(+,+)} d_k(+,+)}{Nd(+,+)} + \frac{\sum_{k=1}^{Nd(-,-)} d_k(-,-)}{Nd(-,-)} \right),$$

where $C_{\text{max}(+)}$ and $C_{\text{max}(-)}$ represents the absolute maximum positive and negative values in the charge plot, respectively, $d(+, +)$, $d(-, -)$, and $d(+, -)$ represent the distance of a pair of charged residues (+: positive charge, -: negative charge) and $Nd(+, +)$, $Nd(-, -)$, and $Nd(+, -)$ represent the numbers of these pairs, respectively.

As shown in the panels below, the kappa value is less sensitive to small charge segregations than D_{seg} , whereas B_{LC} and D_{seg} show good correlation over a wide range of charge segregations. Therefore, we decided to use B_{LC} and D_{seg} values instead of kappa in this study. B_{LC} values are now indicated in all charge plots in the revised figures (Fig. 1b, 2b, 5c and Extended Data Fig. 1a, 1b, 2b, 2f).

(A) Correlation between κ and D_{seg} of R12 mutants used in this study. (B) Dotted square in (A) is enlarged. (C) Correlation between B_{LC} and D_{seg} .

As the reviewer suggested, we plotted C_{sat} against B_{LC} (or D_{seg}) and found a strong correlation between these values (Fig. 2c and Extended Data Fig. 2e). As described in our response to the following comment (comment #5), we performed the same analysis for additional R12 mutants that carried different charge blockiness and net charges and obtained the same tendency. Based on all the findings combined, we strongly believe that LLPS is strongly correlated with charge blockiness. We have described this in the main text (page 7, line 117 – 124).

- Are the authors CERTAIN their sample is pure and free of RNA? The presence of vacuoles in Fig. 2 is something I've only ever seen when multi-component systems undergo maturation, notably when RNA is pulled down during a purification. I'd have to imagine these proteins can bind RNA robustly (regardless of if that is physiologically relevant). We have first-hand experience of this exact issue causing problems, and always purify with a high-salt wash and even RNase treatment. I would encourage the authors to identify a protocol develop for purifying RNA binding proteins without RNA and consider the steps taken there to assess if they are truly getting pure sample. The 260/280

40nm absorbance ratio is also a good diagnostic check. I am not trying to be overly critical or cautious but we have been burned by this EXACT issue before; it is not easy!!!

Response: The morphology of the droplet was also pointed out by Reviewer 2. We fully agree that we should examine the possibility of RNA contamination in the purified protein sample. Therefore, we quantified RNA in the purified protein solutions using a high-sensitive RNA-specific fluorescent probe. The amount of RNA in the protein solution was 0.1–0.7% (w/w). We also examined RNA-binding activity of the repeat domain of Ki-67 using the same method but could not detect any activity (<0.1% w/w). Therefore, we concluded that RNA contamination was not a major reason for the holes in protein droplets. The description on the purity of protein sample has been added to Method section of the revised manuscript (page 21, line 380 – 382).

The morphology of the liquid droplet (size, holes, etc.) provides important information regarding the mechanism of micro-phase separation of a polyampholyte. It is tightly correlated with both the properties of the polymer (e.g. length, charge distribution, density) and environmental factors (e.g. temperature, solvent, salt) and could be an important target of investigation in this research field. We are currently collaborating with theoretical polymer physicists to solve this problem. Although we believe that these investigations will provide more mechanistic insights into the morphology of the liquid droplet, we are of the opinion that this is beyond the scope of this study, and this will be reported in the future.

4. Ammonium acetate seems like a pretty extreme chemical perturbant - how can the authors delineate the impact of ammonium acetate on KI-67 vs. it's global impact on cellular health and physiology? This may be well established but I am unaware of the associated literature.

Response: Ammonium acetate has been used in live cells in other studies, including those of Jane and Vale, *Nature*, 2017; Trivedi et al., *Nature Cell Biology*, 2019; and Yasuda et al., *Nature*, 2020, to elucidate the property of phase separation. The cells were treated with 90 or 100 mM ammonium acetate for 2–30 min. We treated Ki-67-KO cells expressing EGFP-Ki-67 for 10 min with 100 mM ammonium acetate, and the dissociation of Ki-67 from chromosomes was observed soon after adding ammonium acetate. We consider that our treatment has a limited impact on cellular health and physiology.

5. The phospho-dependent localization to the periphery is interesting, but, in my opinion without a control that shows a uniformly shuffled sequence with the same charge does NOT localize to the periphery I'm not sure it's possible to distinguish between a charge-block effect vs. a change in

41overall net charge effect. Given the (actually) modest changes in κ , upon phosphorylation, I'm sort of bias to think maybe this is a net charge effect and not charge patterning effect. Again, this can be confirmed using charge shuffle variant, or at least showing that CBm1 does not rescue function. Showing CBm1 does not rescue function would be a really interesting result given it has the same charge as CBm3.

Response: We fully agree with this point. It could be that a negative shift in the total charge could promote the LLPS of Ki-67 (from +5 to -13 by mitotic phosphorylation and to -4 in the phosphomimetic mutant Pm9). To clarify this, we constructed another series of mutants (CBm-5-16, Extended Data Fig. 2f) that carried different charge blockiness and/or net charges and quantified their LLPS propensity using the turbidity assay and fluorescence microscopy. The saturation concentration (C_{sat}) was more strongly correlated with charge blockiness (B_{LC} and D_{seg} , Fig. 2d and Extended Data Fig. 2g, respectively) than with the net charge (Extended Data Fig. 2h). This result clearly demonstrated that the charge blockiness, rather than the net charge, affects LLPS. We have added these descriptions to the main text (page 7, line 117 – 124).

In addition to the *in vitro* experiments described above, we examined the function of CBms, which carries the same net charge but different charge blockiness (A9, B_{LC} : 5.54, net charge: +5 and CBm-7, B_{LC} : 35, net charge: +5) in the formation of the mitotic chromosome periphery *in vivo*. As shown in Extended Data Fig. 4j, (CBm-7)₁₂-LR, but not (A9)₁₂-LR, could rescue the Ki-67 KO phenotype. We have added this description to the main text (page 12, line 200 – 203). We strongly believe that the enhanced LLPS and the periphery formation do not result from a negative shift of the net charge.

6. Similar concerns raised re: κ and the interpretation of charge blockiness apply to the B23 results - i.e. we need true controls of charge blockiness

Response: According to the comment, we have calculated B_{LC} (and D_{seg}) for all B23 constructs (WT, Pm, and CBm) and performed the turbidity assay (Extended Data Fig. 5c). As demonstrated for Ki-67 RD, the LLPS of B23 (quantified by the C_{sat} value) correlated with the charge blockiness (B_{LC}).

7. Missing references include

Importance of charge blockiness as determinant of cellular assembly

Nott, T. J., Petsalaki, E., Farber, P., Jervis, D., Fussner, E., Plochowietz, A., Craggs, T. D., Bazett-Jones, D. P., Pawson, T., Forman-Kay, J. D., & Baldwin, A. J. (2015). Phase transition of a

disordered nuage protein generates environmentally responsive membraneless organelles. *Molecular Cell*, 57(5), 936–947. (SEE FIG 6A)

Pak, C. W., Kosno, M., Holehouse, A. S., Padrick, S. B., Mittal, A., Ali, R., Yunus, A. A., Liu, D. R., Pappu, R. V., & Rosen, M. K. (2016). Sequence determinants of intracellular phase separation by complex coacervation of a disordered protein. *Molecular Cell*, 63(1), 72–85. (SEE FIG 5)

Bishof, I., Dammer, E. B., Duong, D. M., Kundinger, S., Gearing, M., Lah, J. J., Levey, A. I., & Seyfried, N. T. (2018). RNA-binding proteins with basic–acidic dipeptide (BAD) domains self-assemble and aggregate in Alzheimer’s disease. *The Journal of Biological Chemistry*.
<https://doi.org/10.1074/jbc.RA118.001747>

Response: We thank the referee for the useful suggestion. We have cited these papers in the revised manuscript (page 3, line 52 – 53).

Decision Letter, first revision:

11th February 2022

Dear Dr. Yoshimura,

Thank you for submitting your revised manuscript "Cell cycle-specific phase separation regulated by protein charge blockiness" (NCB-Y46023A). It has now been seen by the original referees and their comments are below. The reviewers find that the paper has improved in revision, and therefore we'll be happy in principle to publish it in *Nature Cell Biology*, pending minor revisions to satisfy the referees' final requests and to comply with our editorial and formatting guidelines.

****The current version of your manuscript is in a PDF format. Please email us a copy of the file in an editable format (Microsoft Word or LaTeX)-- we can not proceed with PDFs at this stage.****

After we receive the Word file, we are to perform detailed checks on your paper and will send you a checklist detailing our editorial and formatting requirements in about a week. Please do not upload the final materials and make any revisions until you receive this additional information from us.

43Thank you again for your interest in Nature Cell Biology. Please do not hesitate to contact me if you have any questions.

Sincerely,

Melina

Melina Casadio, PhD
Senior Editor, Nature Cell Biology
ORCID ID: <https://orcid.org/0000-0003-2389-2243>

Reviewer #1 (Remarks to the Author):

The authors have done a thorough job of responding to the reviews and I now recommend publication of this important study.

Reviewer #2 (Remarks to the Author):

The authors did additional experiments to address my concerns with satisfaction. I don't have further question.

Reviewer #3 (Remarks to the Author):

I want to really thank the authors for a well-executed set of revisions and I strongly support publication. I have a set of comments and suggestions below which I would encourage the authors to consider.

In terms of revision responses:

The quantification of csat vs. length is excellent and strengthens the manuscript

The new extended data comparing net charge and csat is also a major strength

I appreciate the authors entertaining my question re: RNA contamination – I realize this may have been seen as unnecessary but the RNA binding potential of these proteins is so substantial I am very pleased with their thorough assessment of this potential issue. I also look forward to the author's future work in this area, as per their comment!

In vitro PEG

I do have a concern I had somehow missed: are all LLPS experiments done at 15% PEG3350? This is pretty high PEG concentrations - I recognize I should have raised this concern before now, but assuming this is true I would certainly require that the authors clearly state this in the main text (right now the word PEG appears only once in this context, buried in the methods). I may have misunderstood, in which case please accept my apologies. I want to be clear, I don't think this affects the conclusions of the paper at all, nor are additional experiments required but I do think it's important to clearly state this. I think a critical reviewer could worry that the charge block effect in vitro actually reflects the increased propensity of negatively charged stretches to interact with PEG; while this is possible, given the strong in cell data it seems a much less parsimonious explanation than the one provided by the authors, and I would at this stage not consider explicitly testing this to be required nor even a good use of time.

Charge patterning discussion

One additional set of comments - the authors have switched away from using kappa to Bcl and Dseg; I must raise a couple of points with respect to the author's interpretation of aspects of the Das et al 2013 paper (if only for their own education). Note here I use the word 'kappa' to refer to the Greek lowercase 'k'.

The authors state that In this paper, the g value was set as 5 and 6, because the polymer length was fixed at 30. This is not correct. The value g is set at 5 and 6 because this matches the lengthscale at which a polypeptide behaves as a Gaussian chain (the so-called blob-length, see Fig. S12 in Das & Pappu 2013) [also, the polymer length is 50, not 30]. g is not dependent on the sequence length nor composition, but an intrinsic property of the polypeptide chain (in the limit of sequences where the fraction of proline $< \sim 20\%$). As such, again, varying this value is not appropriate; kappa is defined with $g=5$ and 6 (average); g is not a free parameter. Secondly, kappa is a normalized value (i.e. Δ/Δ_{\max}). The authors offer no explanation as to how Δ_{\max} is calculated, but this is numerically not straightforward. As such, I am left wondering if the authors ever actually calculated kappa correctly? This is largely irrelevant given kappa has now been removed, but I raise this in the hope that it will aid the authors in the future.

Though the kappa value is useful for evaluating the charge segregation in a synthetic polyelectrolyte that is composed of only positively and negatively charged monomers, we found that it might not be suitable for actual polypeptides, which comprise only 10–20% charged residues.

Again, while the authors may state this, there is ample evidence that kappa is fully applicable to sequences that are not strong polyampholytes, as evidenced by literally every paper except Das & Pappu 2013 where kappa is shown to correlate with global dimensions. In fact, the entire approach for kappa was developed using strong polyampholytes only as a convenient model system. As such, while I recognize the authors have decided to switch their metric to an alternative one that offers better agreement, it remains unclear if kappa would be appropriate (or not). All this said,

All this said; the revised data strongly supports the authors' intuitions, and I agree charge patterning absolutely seems to correlate with propensity to phase separate in a very compelling way - the above comments are simply meant to clarify confusions here.

I would suggest that it may be useful to justify the use of BCL over kappa. I agree with the authors is an appropriate measure here, but perhaps by saying something like "We chose to use BCL over other charge patterning parameters such as SCD and kappa because BCL employs a larger block size, more appropriately capturing the longer charge tracts we believe are important in KI-67". I suggest this only to provide some explanation to a reader who might wonder why kappa was not used.

A quick final point; while I recognize that the authors no-longer use the kappa parameter, in the name of scholarship I would suggest they retain the Das & Pappu 2013 reference because it is the original paper that proposed that charge patterning in IDPs is an important determinant of IDP conformation and function. For full transparency; I am neither Das nor Pappu!

Missing reference

The authors should really cite Cuylen-Haering et al. Nature 2020 where they discuss the question raised by R2 re: KI-67 extensions. This paper is highly relevant and I am very surprised it was not cited.

Minor points:

In Fig 1b - it would be helpful to delineate the boundary of the two blocks to help the reader

The use of B23 to name NPM1 is somewhat confusing; having worked in the nucleolar space for a number of years I have never heard the name B23, and while I completely recognize that it is a valid alternative name it would I think aid the paper substantially in readability by a more modern audience to use NPM1 (starting early on that NPM1 is also known as B23). I direct the authors to the recent literature, where B23 appears in virtually no publications, whereas NPM1 is ubiquitous used. This is especially relevant in this paper, given the author's naming scheme for the various mutants

When the authors state “Recombinant proteins of human Ki-67 RD formed liquid-like droplets in vitro” I would strongly suggest they (at this point) specify the solution conditions [at a minimum salt and PEG]. This is essential information.

Please ensure all sequences used in this study are provided in a text format that are clearly labeled in the supplementary information

References in this review

Cuylen-Haering, S., Petrovic, M., Hernandez-Armendariz, A., Schneider, M. W. G., Samwer, M., Blaukopf, C., Holt, L. J., & Gerlich, D. W. (2020). Chromosome clustering by Ki-67 excludes cytoplasm during nuclear assembly. *Nature*. <https://doi.org/10.1038/s41586-020-2672-3>

Das, R. K., & Pappu, R. V. (2013). Conformations of intrinsically disordered proteins are influenced by linear sequence distributions of oppositely charged residues. *Proceedings of the National Academy of Sciences of the United States of America*, 110(33), 13392–13397.

28th February 2022

Dear Dr. Yoshimura,

Thank you for your patience as we've prepared the guidelines for final submission of your Nature Cell Biology manuscript, "Cell cycle-specific phase separation regulated by protein charge blockiness" (NCB-Y46023A). Please carefully follow the step-by-step instructions provided in the attached file, and add a response in each row of the table to indicate the changes that you have made. Ensuring that each point is addressed will help to ensure that your revised manuscript can be swiftly handed over to our production team.

We would like to start working on your revised paper, with all of the requested files and forms, as soon as possible (preferably within one week). Please get in contact with us if you anticipate delays.

47If you have not done so already, please alert us to any related manuscripts from your group that are under consideration or in press at other journals, or are being written up for submission to other journals (see: <https://www.nature.com/nature-research/editorial-policies/plagiarism#policy-on-duplicate-publication> for details).

In recognition of the time and expertise our reviewers provide to Nature Cell Biology's editorial process, we would like to formally acknowledge their contribution to the external peer review of your manuscript entitled "Cell cycle-specific phase separation regulated by protein charge blockiness". For those reviewers who give their assent, we will be publishing their names alongside the published article.

Nature Cell Biology offers a Transparent Peer Review option for new original research manuscripts submitted after December 1st, 2019. As part of this initiative, we encourage our authors to support increased transparency into the peer review process by agreeing to have the reviewer comments, author rebuttal letters, and editorial decision letters published as a Supplementary item. When you submit your final files please clearly state in your cover letter whether or not you would like to participate in this initiative. Please note that failure to state your preference will result in delays in accepting your manuscript for publication.

Cover suggestions

As you prepare your final files we encourage you to consider whether you have any images or illustrations that may be appropriate for use on the cover of Nature Cell Biology.

Nature Cell Biology has now transitioned to a unified Rights Collection system which will allow our Author Services team to quickly and easily collect the rights and permissions required to publish your work. Approximately 10 days after your paper is formally accepted, you will receive an email in providing you with a link to complete the grant of rights. If your paper is eligible for Open Access, our Author Services team will also be in touch regarding any additional information that may be required to arrange payment for your article.

Please note that Nature Cell Biology is a Transformative Journal (TJ). Authors may publish their research with us through the traditional subscription access route or make their paper immediately open access through payment of an article-processing charge (APC). Authors will not be required to make a final decision about access to their article until it has been accepted. Find out more about Transformative Journals

Authors may need to take specific actions to achieve compliance with funder and institutional open access mandates. If your research is supported by a funder that requires immediate open access (e.g. according to Plan S principles) then you should select the gold OA route, and we will direct you to the compliant route where possible. For authors selecting the subscription publication route, the journal's standard licensing terms will need to be accepted, including self-archiving policies. Those licensing terms will supersede any other terms that the author or any third party may assert apply to any version of the manuscript.

For information regarding our different publishing models please see our Transformative Journals page. If you have any questions about costs, Open Access requirements, or our legal forms, please contact ASJournals@springernature.com.

Please use the following link for uploading these materials:
[REDACTED]

Best regards,

Editorial Assistant
Nature Cell Biology

On behalf of

Melina Casadio, PhD
Senior Editor, Nature Cell Biology
ORCID ID: <https://orcid.org/0000-0003-2389-2243>

Reviewer #1:

Remarks to the Author:

The authors have done a thorough job of responding to the reviews and I now recommend publication of this important study.

Reviewer #2:

Remarks to the Author:

The authors did additional experiments to address my concerns with satisfaction. I don't have further question.

Reviewer #3:

Remarks to the Author:

I want to really thank the authors for a well-executed set of revisions and I strongly support publication. I have a set of comments and suggestions below which I would encourage the authors to consider.

In terms of revision responses:

The quantification of csat vs. length is excellent and strengthens the manuscript

The new extended data comparing net charge and csat is also a major strength

I appreciate the authors entertaining my question re: RNA contamination – I realize this may have been seen as unnecessary but the RNA binding potential of these proteins is so substantial I am very pleased

50with their thorough assessment of this potential issue. I also look forward to the author's future work in this area, as per their comment!

In vitro PEG

I do have a concern I had somehow missed: are all LLPS experiments done at 15% PEG3350? This is pretty high PEG concentrations - I recognize I should have raised this concern before now, but assuming this is true I would certainly require that the authors clearly state this in the main text (right now the word PEG appears only once in this context, buried in the methods). I may have misunderstood, in which case please accept my apologies. I want to be clear, I don't think this affects the conclusions of the paper at all, nor are additional experiments required but I do think it's important to clearly state this. I think a critical reviewer could worry that the charge block effect in vitro actually reflects the increased propensity of negatively charged stretches to interact with PEG; while this is possible, given the strong in cell data it seems a much less parsimonious explanation than the one provided by the authors, and I would at this stage not consider explicitly testing this to be required nor even a good use of time.

Charge patterning discussion

One additional set of comments - the authors have switched away from using kappa to Bcl and Dseg; I must raise a couple of points with respect to the author's interpretation of aspects of the Das et al 2013 paper (if only for only their own education). Note here I use the word 'kappa' to refer to the Greek lowercase 'k'.

The authors state that In this paper, the g value was set as 5 and 6, because the polymer length was fixed at 30. This is not correct. The value g is set at 5 and 6 because this matches the lengthscale at which a polypeptide behaves as a Gaussian chain (the so-called blob-length, see Fig. S12 in Das & Pappu 2013) [also, the polymer length is 50, not 30]. g is not dependent on the sequence length nor composition, but an intrinsic property of the polypeptide chain (in the limit of sequences where the fraction of proline $< \sim 20\%$). As such, again, varying this value is not appropriate; kappa is defined with $g=5$ and 6 (average); g is not a free parameter. Secondly, kappa is a normalized value (i.e. Δ/Δ_{max}). The authors offer no explanation as to how Δ_{max} is calculated, but this is numerically not straightforward. As such, I am left wondering if the authors ever actually calculated kappa correctly? This is largely irrelevant given kappa has now been removed, but I raise this in the hope that it will aid the authors in the future.

Though the kappa value is useful for evaluating the charge segregation in a synthetic polyelectrolyte that is composed of only positively and negatively charged monomers, we found that it might not be suitable for actual polypeptides, which comprise only 10–20% charged residues.

Again, while the authors may state this, there is ample evidence that kappa is fully applicable to sequences that are not strong polyampholytes, as evidenced by literally every paper except Das & Pappu 2013 where kappa is shown to correlate with global dimensions. In fact, the entire approach for kappa was developed using strong polyampholytes only as a convenient model system. As such, while I recognize the authors have decided to switch their metric to an alternative one that offers better agreement, it remains unclear if kappa would be appropriate (or not). All this said,

All this said; the revised data strongly supports the authors' intuitions, and I agree charge patterning absolutely seems to correlate with propensity to phase separate in a very compelling way - the above comments are simply meant to clarify confusions here.

I would suggest that it may be useful to justify the use of BCL over kappa. I agree with the authors is an appropriate measure here, but perhaps by saying something like “We chose to use BCL over other charge patterning parameters such as SCD and kappa because BCL employs a larger block size, more appropriately capturing the longer charge tracts we believe are important in KI-67”. I suggest this only to provide some explanation to a reader who might wonder why kappa was not used.

A quick final point; while I recognize that the authors no-longer use the kappa parameter, in the name of scholarship I would suggest they retain the Das & Pappu 2013 reference because it is the original paper that proposed that charge patterning in IDPs is an important determinant of IDP conformation and function. For full transparency; I am neither Das nor Pappu!

Missing reference

The authors should really cite Cuylen-Haering et al. Nature 2020 where they discuss the question raised by R2 re: KI-67 extensions. This paper is highly relevant and I am very surprised it was not cited.

Minor points:

In Fig 1b - it would be helpful to delineate the boundary of the two blocks to help the reader

The use of B23 to name NPM1 is somewhat confusing; having worked in the nucleolar space for a number of years I have never heard the name B23, and while I completely recognize that it is a valid alternative name it would I think aid the paper substantially in readability by a more modern audience

to use NPM1 (starting early on that NPM1 is also known as B23). I direct the authors to the recent literature, where B23 appears in virtually no publications, whereas NPM1 is ubiquitous used. This is especially relevant in this paper, given the author's naming scheme for the various mutants

When the authors state "Recombinant proteins of human Ki-67 RD formed liquid-like droplets in vitro" I would strongly suggest they (at this point) specify the solution conditions [at a minimum salt and PEG]. This is essential information.

Please ensure all sequences used in this study are provided in a text format that are clearly labeled in the supplementary information

References in this review

Cuylen-Haering, S., Petrovic, M., Hernandez-Armendariz, A., Schneider, M. W. G., Samwer, M., Blaukopf, C., Holt, L. J., & Gerlich, D. W. (2020). Chromosome clustering by Ki-67 excludes cytoplasm during nuclear assembly. *Nature*. <https://doi.org/10.1038/s41586-020-2672-3>

Das, R. K., & Pappu, R. V. (2013). Conformations of intrinsically disordered proteins are influenced by linear sequence distributions of oppositely charged residues. *Proceedings of the National Academy of Sciences of the United States of America*, 110(33), 13392–13397.

Author Rebuttal, first revision:

Reviewer #1 (Remarks to the Author):

The authors have done a thorough job of responding the the reviews and I now recommend publication of this important study.

Response: We highly appreciate all of the Reviewers' comments, and their time and effort for improving our manuscript. We are very pleased to know that all of their concerns have been addressed in the revised manuscript.

Reviewer #2 (Remarks to the Author):

The authors did additional experiments to address my concerns with satisfaction. I don't have further question.

53Response: We highly appreciate all of the Reviewers' comments, and their time and effort for improving our manuscript. We are very pleased to know that all of their concerns have been addressed in the revised manuscript.

Reviewer #3 (Remarks to the Author):

I want to really thank the authors for a well-executed set of revisions and I strongly support publication. I have a set of comments and suggestions below which I would encourage the authors to consider.

In terms of revision responses:

The quantification of csat vs. length is excellent and strengthens the manuscript

The new extended data comparing net charge and csat is also a major strength

I appreciate the authors entertaining my question re: RNA contamination – I realize this may have been seen as unnecessary but the RNA binding potential of these proteins is so substantial I am very pleased with their thorough assessment of this potential issue. I also look forward to the author's future work in this area, as per their comment!

Response: We highly appreciate all of the comments, and great support to our manuscript. They are really useful to improve our manuscript.

In vitro PEG

I do have a concern I had somehow missed: are all LLPS experiments done at 15% PEG3350? This is pretty high PEG concentrations - I recognize I should have raised this concern before now, but assuming this is true I would certainly require that the authors clearly state this in the main text (right now the word PEG appears only once in this context, buried in the methods). I may have misunderstood, in which case please accept my apologies. I want to be clear, I don't think this affects the conclusions of the paper at all, nor are additional experiments required but I do think it's important to clearly state this. I think a critical reviewer could worry that the charge block effect in vitro actually reflects the increased propensity of negatively charged stretches to interact with PEG; while this is possible, given the strong in cell data it seems a much less parsimonious explanation than the one provided by the authors, and I would at this stage not consider explicitly testing this to be required nor even a good use of time.

Response: We highly appreciate the comment. We totally agree that the experimental conditions of *in vitro* LLPS assay, especially concentration of PEG, which is important for the interpretation of the result, should be clearly described in the manuscript. According to the reviewer's suggestion, we have added the description of PEG concentration in the main text of the revised manuscript (page 4, line 78-79). Although there is a possibility that the enhanced charge blocks interact with PEG in solution to promote LLPS, our results from *in vivo* observation do not support this, as the reviewer kindly mentioned. PEG has been widely used as a crowder in various *in vitro* condensation assays using charged polymers and polypeptides. Examining possible interaction between PEG and charge blocks in a polypeptide and its effect on LLPS may provide additional insights into the mechanism of LLPS. However, we are of the opinion that it is beyond

54the scope of our present study and will be thoroughly investigated in the future, as the reviewer mentioned.

Charge patterning discussion

One additional set of comments - the authors have switched away from using kappa to Bcl and Dseg; I must raise a couple of points with respect to the author's interpretation of aspects of the Das et al 2013 paper (if only for only their own education). Note here I use the word 'kappa' to refer to the Greek lowercase 'k'.

The authors state that In this paper, the g value was set as 5 and 6, because the polymer length was fixed at 30. This is not correct. The value g is set at 5 and 6 because this matches the lengthscale at which a polypeptide behaves as a Gaussian chain (the so-called blob-length, see Fig. S12 in Das & Pappu 2013) [also, the polymer length is 50, not 30]. g is not dependent on the sequence length nor composition, but an intrinsic property of the polypeptide chain (in the limit of sequences where the fraction of proline < ~20%). As such, again, varying this value is not appropriate; kappa is defined with g=5 and 6 (average); g is not a free parameter. Secondly, kappa is a normalized value (i.e. delta/deltamax). The authors offer no explanation as to how deltamax is calculated, but this is numerically not straightforward. As such, I am left wondering if the authors ever actually calculated kappa correctly? This is largely irrelevant given kappa has now been removed, but I raise this in the hope that it will aid the authors in the future.

Though the kappa value is useful for evaluating the charge segregation in a synthetic polyelectrolyte that is composed of only positively and negatively charged monomers, we found that it might not be suitable for actual polypeptides, which comprise only 10–20% charged residues.

Again, while the authors may state this, there is ample evidence that kappa is fully applicable to sequences that are not strong polyampholytes, as evidenced by literally every paper except Das & Pappu 2013 where kappa is shown to correlate with global dimensions. In fact, the entire approach for kappa was developed using strong polyampholytes only as a convenient model system. As such, while I recognize the authors have decided to switch their metric to an alternative one that offers better agreement, it remains unclear if kappa would be appropriate (or not). All this said,

All this said; the revised data strongly supports the authors' intuitions, and I agree charge patterning absolutely seems to correlate with propensity to phase separate in a very compelling way - the above comments are simply meant to clarify confusions here.

I would suggest that it may be useful to justify the use of BCL over kappa. I agree with the authors is an appropriate measure here, but perhaps by saying something like “We chose to use BCL over other charge patterning parameters such as SCD and kappa because BCL employees a larger block size, more appropriately capturing the longer charge tracts we believe are important in KI-67”. I suggest this only to provide some explanation to a reader who might wonder why kappa was not used.

Response: We thank the referee for the useful suggestion. We agree that we should explain why we used BLC over kappa, and have added those sentences in the revised manuscript (page 18, line 459-462).

A quick final point; while I recognize that the authors no-longer use the kappa parameter, in the name of

55scholarship I would suggest they retain the Das & Pappu 2013 reference because it is the original paper that proposed that charge patterning in IDPs is an important determinant of IDP conformation and function. For full transparency; I am neither Das nor Pappu!

Response: We totally agree that we should cite Das and Pappu, 2013. It has been added in the revised manuscript.

Missing reference

The authors should really cite Cuylen-Haering et al. Nature 2020 where they discuss the question raised by R2 re: KI-67 extensions. This paper is highly relevant and I am very surprised it was not cited.

Response: We appreciate the referee for the useful suggestion. We have cited this paper in the revised manuscript (page 3, line 59).

Minor points:

In Fig 1b - it would be helpful to delineate the boundary of the two blocks to help the reader

Response: We also thank this useful suggestion. We added the boundary of the charge blocks in Fig. 1b.

The use of B23 to name NPM1 is somewhat confusing; having worked in the nucleolar space for a number of years I have never heard the name B23, and while I completely recognize that it is a valid alternative name it would I think aid the paper substantially in readability by a more modern audience to use NPM1 (starting early on that NPM1 is also known as B23). I direct the authors to the recent literature, where B23 appears in virtually no publications, whereas NPM1 is ubiquitous used. This is especially relevant in this paper, given the author's naming scheme for the various mutants

Response: We agree that NPM1 is more widely used in the recent studies. We have replaced B23 by NPM1 in the text, figures and figure legends.

When the authors state "Recombinant proteins of human Ki-67 RD formed liquid-like droplets in vitro" I would strongly suggest they (at this point) specify the solution conditions [at a minimum salt and PEG]. This is essential information.

Response: We appreciate this comment. We agree that it is important to describe the experimental condition of *in vitro* droplet assay. We have added such information in the main text of the revised manuscript (page 4, line 78-79).

Please ensure all sequences used in this study are provided in a text format that are clearly labeled in the supplementary information

56Response: We highly appreciate this comment, and have presented the amino-acid sequences of all constructs that we used in this study in Supplementary Table.

References in this review

Cuylen-Haering, S., Petrovic, M., Hernandez-Armendariz, A., Schneider, M. W. G., Samwer, M., Blaukopf, C., Holt, L. J., & Gerlich, D. W. (2020). Chromosome clustering by Ki-67 excludes cytoplasm during nuclear assembly. *Nature*. <https://doi.org/10.1038/s41586-020-2672-3>

Das, R. K., & Pappu, R. V. (2013). Conformations of intrinsically disordered proteins are influenced by linear sequence distributions of oppositely charged residues. *Proceedings of the National Academy of Sciences of the United States of America*, 110(33), 13392–13397.

Final Decision Letter:

Dear Dr Yoshimura,

I am pleased to inform you that your manuscript, "Cell cycle-specific phase separation regulated by protein charge blockiness", has now been accepted for publication in *Nature Cell Biology*.

Over the next few weeks, your paper will be copyedited to ensure that it conforms to *Nature Cell Biology* style. Once your paper is typeset, you will receive an email with a link to choose the appropriate publishing options for your paper and our Author Services team will be in touch regarding any additional information that may be required.

57Due to the importance of these deadlines, we ask that you please let us know now whether you will be difficult to contact over the next month. If this is the case, we ask you provide us with the contact information (email, phone and fax) of someone who will be able to check the proofs on your behalf, and who will be available to address any last-minute problems.

Please note that Nature Cell Biology is a Transformative Journal (TJ). Authors may publish their research with us through the traditional subscription access route or make their paper immediately open access through payment of an article-processing charge (APC). Authors will not be required to make a final decision about access to their article until it has been accepted. Find out more about Transformative Journals

Authors may need to take specific actions to achieve compliance with funder and institutional open access mandates. If your research is supported by a funder that requires immediate open access (e.g. according to Plan S principles) then you should select the gold OA route, and we will direct you to the compliant route where possible. For authors selecting the subscription publication route, the journal's standard licensing terms will need to be accepted, including self-archiving policies. Those licensing terms will supersede any other terms that the author or any third party may assert apply to any version of the manuscript.

If your paper includes color figures, please be aware that in order to help cover some of the additional cost of four-color reproduction, Nature Research charges our authors a fee for the printing of their color figures. Please contact our offices for exact pricing and details.

If you have not already done so, we strongly recommend that you upload the step-by-step protocols used in this manuscript to the Protocol Exchange (www.nature.com/protocolexchange), an open online resource established by Nature Protocols that allows researchers to share their detailed experimental know-how. All uploaded protocols are made freely available, assigned DOIs for ease of citation and are fully searchable through nature.com. Protocols and the Nature and Nature research journal papers in which they are used can be linked to one another, and this link is clearly and prominently visible in the online versions of both papers. Authors who performed the specific experiments can act as primary authors for the Protocol as they will be best placed to share the methodology details, but the Corresponding Author of the present research paper should be included as one of the authors. By uploading your Protocols to Protocol Exchange, you are enabling researchers to more readily reproduce or adapt the methodology you use, as well as increasing the visibility of your protocols and papers. You can also establish a dedicated page to collect your lab Protocols. Further information can be found at www.nature.com/protocolexchange/about

You can use a single sign-on for all your accounts, view the status of all your manuscript submissions and reviews, access usage statistics for your published articles and download a record of your refereeing activity for the Nature journals.

With kind regards,

Melina

Melina Casadio, PhD
Senior Editor, Nature Cell Biology
ORCID ID: <https://orcid.org/0000-0003-2389-2243>

** Visit the Springer Nature Editorial and Publishing website at www.springernature.com/editorial-and-publishing-jobs for more information about our career opportunities. If you have any questions please click here.**